# Clonal dynamics of haematopoiesis across the human lifespan

Emily Mitchell[1,2,3], Michael Spencer Chapman[1,10], Nicholas Williams[1,10], Kevin J. Dawson[1,10], Nicole Mende[2], Emily F. Calderbank[2], Hyunchul Jung[1], Thomas Mitchell[1], Tim H. H. Coorens[1], David H. Spencer[4], Heather Machado[1], Henry Lee-Six[1], Megan Davies[5], Daniel Hayler[2], Margarete A. Fabre[1,2,3], Krishnaa Mahbubani[6,7], Federico Abascal[1], Alex Cagan[1], George S. Vassiliou[1,2,3], Joanna Baxter[3], Inigo Martincorena[1], Michael R. Stratton[1], David G. Kent[8], Krishna Chatterjee[9], Kourosh Saeb Parsy[6,7], Anthony R. Green[2,3], Jyoti Nangalia[1,2,3,11✉], Elisa Laurenti[2,3,11✉] & Peter J. Campbell[1,2,11✉]

Age-related change in human haematopoiesis causes reduced regenerative capacity[1], cytopenias[2], immune dysfunction[3] and increased risk of blood cancer[4–6], but the reason for such abrupt functional decline after 70 years of age remains unclear. Here we sequenced 3,579 genomes from single cell-derived colonies of haematopoietic cells across 10 human subjects from 0 to 81 years of age. Haematopoietic stem cells or multipotent progenitors (HSC/MPPs) accumulated a mean of 17 mutations per year after birth and lost 30 base pairs per year of telomere length. Haematopoiesis in adults less than 65 years of age was massively polyclonal, with high clonal diversity and a stable population of 20,000–200,000 HSC/MPPs contributing evenly to blood production. By contrast, haematopoiesis in individuals aged over 75 showed profoundly decreased clonal diversity. In each of the older subjects, 30–60% of haematopoiesis was accounted for by 12–18 independent clones, each contributing 1–34% of blood production. Most clones had begun their expansion before the subject was 40 years old, but only 22% had known driver mutations. Genome-wide selection analysis estimated that between 1 in 34 and 1 in 12 non-synonymous mutations were drivers, accruing at constant rates throughout life, affecting more genes than identified in blood cancers. Loss of the Y chromosome conferred selective benefits in males. Simulations of haematopoiesis, with constant stem cell population size and constant acquisition of driver mutations conferring moderate fitness benefits, entirely explained the abrupt change in clonal structure in the elderly. Rapidly decreasing clonal diversity is a universal feature of haematopoiesis in aged humans, underpinned by pervasive positive selection acting on many more genes than currently identified.

The age-related mortality curve for modern humans is an outlier across the tree of life, with an abrupt increase in mortality after the average lifespan[7], leading to surprisingly low variance in age at death[8]. Studies of ageing at the cellular level have demonstrated that accumulation of molecular damage across the lifespan is gradual and lifelong, including telomere attrition, somatic mutation, epigenetic change and oxidative or replicative stress[9]. It remains unresolved how such gradual accumulation of molecular damage can translate into an abrupt increase in mortality after 70 years of age.

Sequencing of blood samples from population cohorts has revealed an age-related increase in acquired mutations in genes that cause myeloid neoplasms[4,5,10–12], known as driver mutations. This phenomenon is called clonal haematopoiesis, and reaches 10–20% prevalence[4,5,10–12] or higher[13] after 70 years of age, but the driver mutations typically account for only a small fraction of haematopoiesis (less than 5% of cells). Some elderly individuals show evidence of clonal expansions even in the absence of known driver mutations[14–16]. Deep sequencing of bulk blood samples, as used in these studies, struggles to elucidate clonal relationships among cells and is insensitive to mutations that are present in a low proportion (below 1–5%) of blood cells—as a result, we lack an unbiased, high-resolution model for the clonal dynamics of human haematopoiesis with ageing. Whole-genome

[1]Wellcome Sanger Institute, Hinxton, UK. [2]Wellcome-MRC Cambridge Stem Cell Institute, Cambridge Biomedical Campus, Cambridge, UK. [3]Department of Haematology, University of Cambridge, Cambridge, UK. [4]Department of Medicine, McDonnell Genome Institute, Washington University, St Louis, MO, USA. [5]Cambridge Molecular Diagnostics, Milton Road, Cambridge, UK. [6]Department of Surgery, University of Cambridge, Cambridge, UK. [7]Cambridge Biorepository for Translational Medicine, NIHR Cambridge Biomedical Research Centre, University of Cambridge, Cambridge, UK. [8]York Biomedical Research Institute, Department of Biology, University of York, York, UK. [9]Wellcome Trust-MRC Institute of Metabolic Science, University of Cambridge, Cambridge, UK. [10]These authors contributed equally: Michael Spencer Chapman, Nicholas Williams, Kevin J. Dawson. [11]These authors jointly supervised this work: Jyoti Nangalia, Elisa Laurenti, Peter J. Campbell. ✉e-mail: jn5@sanger.ac.uk; el422@cam.ac.uk; pc8@sanger.ac.uk

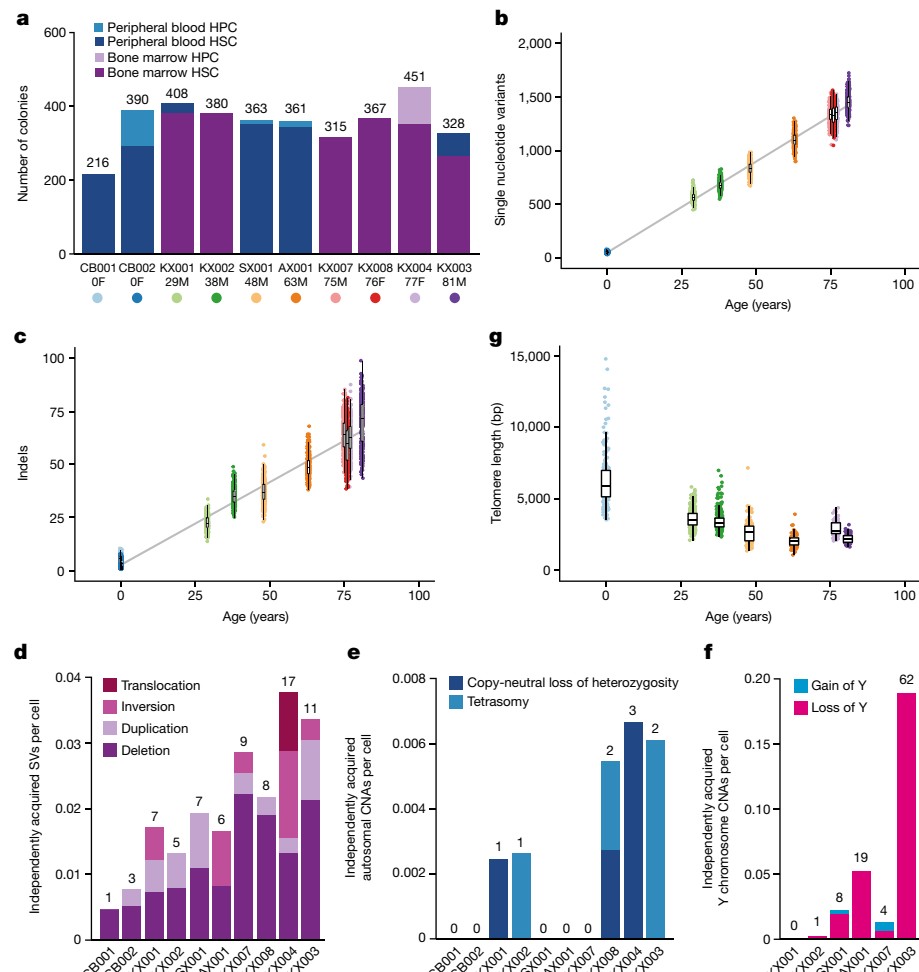

**Fig. 1 | Mutational burden in normal HSC/MPPs. a**, Bar plot showing the numbers of colonies sequenced from each tissue and cell type for each donor in the study. Age and sex (F, female; M, male) are indicated below the donor ID. **b**, Burden of single nucleotide variants across the donor cohort. The points represent individual HSC/MPP colonies (*n* = 3,361) and are coloured by donor as indicated in **a**. The grey line represents a regression of age with mutation burden, with shading indicating the 95% confidence interval. **c**, Burden of small indels across the donor cohort. **d**, Bar plot showing the number of independently acquired structural variants (SVs) per colony sequenced in each donor. The absolute number of structural variants is shown at the top of each bar. **e**, Bar plot showing the number of independently acquired autosomal copy

number aberrations (CNAs) per colony sequenced in each donor. The absolute number of copy number aberrations is shown at the top of each bar. **f**, Bar plot showing the number of independently acquired Y chromosome copy number aberrations sequenced in each male donor. The absolute number of copy number aberrations is shown at the top of each bar. **g**, Telomere length across the donor cohort, including only those samples sequenced on the HiSeq X10 platform. Each point represents a single HSC/MPP colony. Two outlying points for CB001 are not shown (telomere lengths 16,037 bp and 21,155 bp). In **b**, **c**, **g**, boxes overlaid indicate the median and interquartile range and whiskers denote the minimum of either the range or 25th and 75th centile plus 1.5× interquartile range.

sequencing of colonies grown from single cells circumvents these limitations of bulk sequencing[17–19]. We sequenced whole genomes of 3,579 single cell-derived haematopoietic colonies from 10 healthy individuals spanning the human lifespan, revealing an abrupt and universal loss of clonal diversity after the age of 70 years.

## Whole-genome sequencing of HSC colonies

We obtained samples from 10 individuals, with no known haematological disease, aged between 0 and 81 years (Extended Data Fig. 1a, Supplementary Table 1). One subject (KX002, a 38-year-old male) had inflammatory bowel disease treated with azathioprine and another (SX001, a 48-year0old male) had selenoprotein deficiency[20], a genetic disorder not known to affect haemapoietic stem cell (HSC) dynamics. Stem cells were obtained from cord blood for the two neonates, and from bone marrow and/or peripheral blood for adult donors (Fig. 1a). Bone marrow samples were obtained peri-mortem, enabling sampling of large volumes (50–80 ml) from multiple vertebrae.

Single immunophenotypic HSC/MPPs (Lin⁻CD34⁺CD38⁻CD45RA−) were sorted using flow cytometry and cultured (Extended Data Fig. 1b). Overall, 42–89% of sorted HSC/MPPs produced colonies (Extended Data Fig. 1c), suggesting that the sequenced colonies were a representative sample of the HSC/MPP population in each individual, as supported by analysis of cell surface markers (Extended Data Fig. 1d). Haematopoietic progenitor cells (HPCs) (Lin⁻CD34⁺CD38⁺) were also flow-sorted for four individuals.

We performed whole-genome sequencing at average sequencing depths of 14× on 224–453 colonies per individual. We excluded 17 colonies with low coverage, 34 technical duplicates and 7 colonies derived from more than a single cell (Extended Data Fig. 2a–c). The final dataset comprised whole genomes from 3,579 colonies, of which 3,361 were derived from HSC/MPPs and 218 were derived from HPCs. Raw mutation burdens were corrected for sequencing depth using asymptotic regression (Extended Data Fig. 2d). Single-base substitution spectra were consistent with previous results[17,18,21] (Extended Data Fig. 2e, f). Phylogenetic trees for each adult individual were constructed—terminal branch lengths

were then corrected for sequencing depth, followed by normalization to identical total root-to-tip branch lengths and scaling to chronological time (Extended Data Fig. 3). Benchmarking of the phylogenies included assessments of internal consistency, stability across phylogenetic inference algorithms and robustness to bootstrapping (Supplementary Methods). All code for variant filtering, phylogenetic reconstruction, benchmarking and downstream analyses are available in the Supplementary Code.

## Mutation burden and telomere lengths

Consistent with previous data[18,22], point mutations accumulated linearly in HSC/MPPs throughout life, at a rate of 16.8 substitutions per cell per year (95% confidence interval 16.5–17.1; Fig. 1b) and 0.71 insertion–deletion mutations (indels) per cell per year (95% confidence interval 0.65–0.77; Fig. 1c). We found no significant difference in mutation burden between HSC/MPPs and HPCs (Extended Data Fig. 4a). Structural variants were rare, with only 1–17 events observed in each individual, mostly deletions, correlating with age (Fig. 1d, Extended Data Fig. 4b, Supplementary Table 3). Autosomal copy number aberrations were rare at all ages and comprised either copy-neutral loss-of-heterozygosity events or tetrasomies (Fig. 1e). By contrast, loss of the Y chromosome was frequent in males, increasing with age as previously shown[23] (Fig. 1f); no corresponding examples of loss of the inactive X chromosome were observed in older females.

We estimated telomere lengths for the 1,505 HSC/MPP colonies from 7 individuals sequenced on Hiseq X10 (Fig. 1g). As previously reported[24,25], telomere lengths decreased steadily with age, at an average attrition rate of 30.8 base pairs (bp) per year in adult life (95% confidence interval 13.2–48.4), close to published estimates of 39 bp per year from bulk granulocytes[25]. By sequencing single cell-derived colonies, we can estimate the variance and distribution in telomere lengths among cells with a greater resolution than is possible with bulk populations. In cord blood and young adults, a small proportion of HSC/MPPs had unexpectedly long telomeres, a proportion that decreased with age (Extended Data Fig. 4c). Given that telomeres shorten at cell division, these outlier cells have presumably undergone fewer historic cell divisions. A rare population of infrequently dividing dormant HSCs has been described in the mouse[26,27] and our telomere data are consistent with an analogous population in humans, especially early in life.

## HSC clonal dynamics after 70 years of age

The phylogenetic trees generated here depict the lineage relationships among ancestors of the sequenced stem and progenitor cells. Given the consistent, linear rate of mutation accumulation across the lifespan, we scaled the raw phylogenetic trees (Extended Data Fig. 3d) to chronological time to study clonal dynamics of HSC/MPPs across the lifespan (Figs. 2, 3, Extended Data Fig. 5). Branch points in the tree ('coalescences') define historic stem cell divisions. In taxonomy, a clade is defined as a group of organisms descended from a single common ancestor—in the context of somatic cells, this represents a clone, and its size can be estimated from the fraction of colonies derived from that ancestor. Here, we define an 'expanded clade' as a postnatal ancestral lineage whose descendants contributed more than 1% of colonies at the time of sampling.

Phylogenetic trees of the 4 adults aged below 65 years showed that healthy haematopoiesis in young and middle-aged individuals is highly polyclonal (Fig. 2, Extended Data Fig. 5a). Despite sequencing 361–408 colonies per individual, we found at most a single expanded clade in each sample, and the two that we observed contributed less than 2% of all haematopoiesis in those individuals. Only four known or possible driver point mutations were identified, none of which occurred in the two expanded clades.

Phylogenetic trees for the 4 adults aged over 70 years were qualitatively different (Fig. 3, Extended Data Fig. 5b), with an oligoclonal pattern of haematopoiesis. In each elderly individual, we found 12–18

independent clones established between birth and 40 years of age that each contributed between 1% and 34% of colonies sequenced, most in the 1–3% range. Collectively, these clones summed to a significant proportion of all blood production in our elderly research subjects—between 32% and 61% of all colonies sequenced derived from the expanded clades.

Only a minority of clonal expansions in the elderly individuals carried known driver mutations. Although we identified mutations in *DNMT3A*, *TET2* and *CBL*, mutations in the top 17 myeloid driver genes could explain only 10 out of 58 expanded clades with a clonal fraction above 1%. We identified only 3 additional mutations when we extended the analysis to a wider set of 92 genes implicated in myeloid neoplasms[28] (Supplementary Tables 4, 5), leaving 45 clonal expansions unexplained.

## HSC/MPP population size in young donors

The frequency of branch points in phylogenetic trees in a neutrally evolving, well-mixed population of somatic cells is determined primarily by the product of population size and time between symmetric self-renewal cell divisions ($N\tau$)—both smaller populations and more frequent symmetric divisions increase coalescences. In young adults, in whom clonal selection has had minimal impact on phylogenetic trees, we can exploit this property to estimate the lifelong trajectory of population size dynamics[29] (Fig. 4a). There is a rapid increase in estimated $N\tau$ in utero, reaching a plateau during infancy and early childhood. This is evidenced by a high density of coalescences in the first approximately 50 mutations along molecular time (Fig. 2, Extended Data Fig. 5a); the cord blood colonies we sequenced had a mean of 55 mutations.

Using both phylodynamic and approximate Bayesian computation (ABC) modelling, we estimate that steady-state haematopoiesis plateaus at an $N\tau$ of about 100,000 HSC-years (Fig. 4). This was consistent across all 4 young adults, with individual estimates and 95% confidence intervals in the range 50,000–250,000, matching published estimates[17,30]. To estimate $N$ (the HSC population size), we require an estimate of $\tau$, the time between symmetric HSC self-renewal divisions. Previous estimates[17,25,31] of $\tau$ for HSCs range between 0.6 and 6 years. Telomeres in HSCs shorten at mitosis by 30–100 bp per cell division[32], and therefore provide an independent means to estimate the number of historic cell divisions an HSC has undergone. We estimated that HSC/MPP telomeres shorten at a rate of 30 bp per year (Fig. 1g)—this bounds the number of symmetric cell divisions to at most 1–2 divisions per year. Two recent studies in mice have shown that symmetric divisions predominate within the HSC pool, accounting for 80–100% of all HSC divisions[33,34]. Together, these data are most consistent with adult haematopoiesis being maintained by a population of 20,000–200,000 long-term HSCs.

In the young individuals, and indeed in the four elderly individuals, we noted a sparsity of branch points towards the tips of the phylogenies compared to earlier ages—this emerged consistently about 10–15 years of molecular time before sampling, irrespective of the subject's age. We hypothesize that this pattern arises from a large, short-term HSC/MPP compartment that contributes to haematopoiesis for 10–15 years. Owing to its short-term contribution, the effects of this compartment would only be evident at the tips of the phylogeny, with earlier branches reflecting dynamics in the long-term HSC compartment. A substantially larger population size for these short-term HSC/MPPs would explain why the density of coalescences decreases towards the tips of the trees (Extended Data Fig. 6). The mouse short-term HSC/MPP compartment is known to sustain steady-state haematopoiesis without significant input from the long-term HSC compartment for many months[35,36], a similar proportion of a mouse's lifespan as 10–15 years is for a human.

## HSC dynamics in the elderly

We observed a qualitative change in the population structure of HSCs in the elderly, with an abrupt increase in the frequency of expanded clones and the total fraction of haematopoiesis they generate (Fig. 5a).

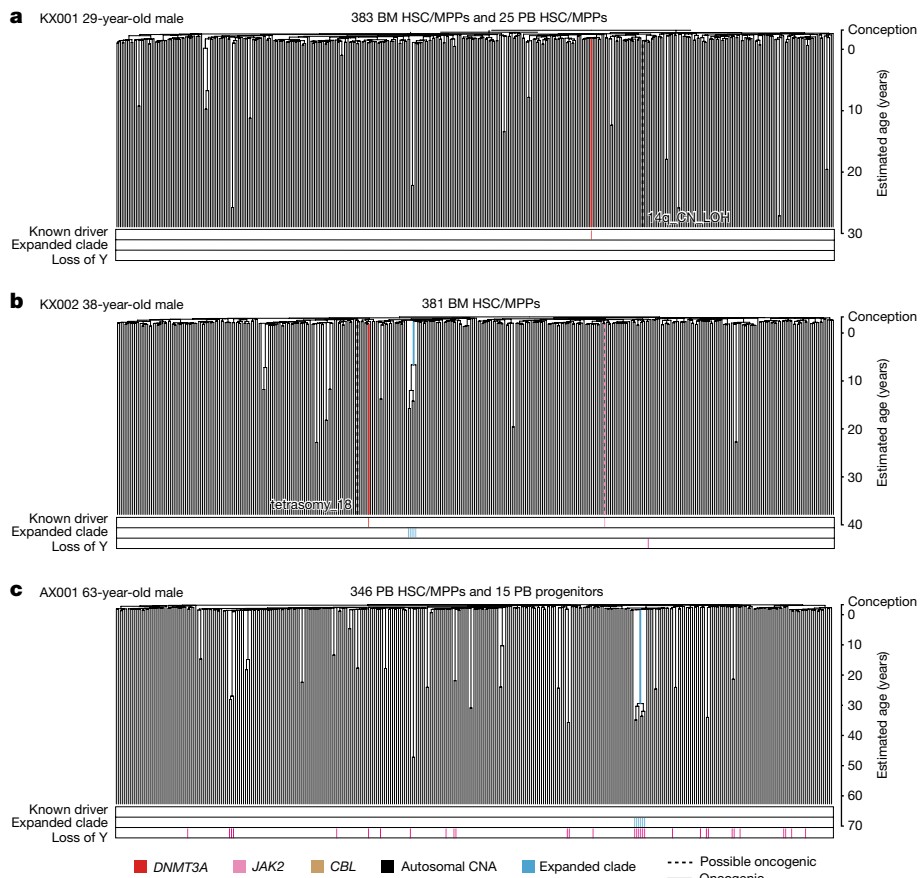

**Fig. 2 | HSPC phylogenies for three young adult donors. a–c**, Phylogenies were constructed for a 29-year-old (**a**), a 38-year-old (**b**) and a 63-year-old (**c**) male donor using shared mutation data and the algorithm MPBoot (Methods). Branch lengths are proportional to the number of mutations assigned to the branch—terminal branches have been corrected for sequence coverage, and overall root-to-tip branch lengths have been normalized to the same total length (because all colonies were collected from a single time point). The y-axis is scaled to chronological time using the somatic mutation rate as a molecular clock, with age 0 (representing birth) set at 55 mutations (as estimated from our cord blood colonies). Each tip on a phylogeny represents a single colony, with the respective numbers of colonies of each cell and tissue type recorded at the top. Onto these trees, we have layered clone and colony-specific phenotypic information. We have highlighted branches on which we have identified known oncogenic drivers (solid line) and possible oncogenic drivers (dashed line) in one of 17 clonal haematopoiesis genes (Supplementary Table 4), coloured by gene. Branches with autosomal copy number alterations are highlighted with a black dashed line. A heat map at the bottom of each phylogeny highlights colonies from known driver clades coloured by gene, expanded clades (defined as those with a clonal fraction above 1%) in blue and colonies with loss of the Y chromosome in pink (males only). BM, bone marrow; PB, peripheral blood; CN_LOH, copy-neutral loss of heterozygosity. The phylogeny of the fourth young adult donor is shown in Extended Data Fig. 5a.

The Shannon index, which measures clonal diversity, showed a precipitous decline in diversity after age 70 (Fig. 5b). Although this change was evident only in the elderly, the branch points in the phylogenetic tree that underpinned the clonal expansions occurred much earlier in life, between childhood and middle age (Extended Data Fig. 7). There are several possible explanations for these observations, including age-related changes in population size, changes in spatial patterning of HSCs and cell-autonomous variation in selective fitness.

Using ABC, we explored whether changes in population size, most probably a population bottleneck occurring in mid-life, could explain the observed trees (Extended Data Fig. 7b). However, although they accurately recapitulated trees observed in subjects over 65 years of age, even the 1% most closely matching phylogenies from these simulations poorly replicated those observed in the elderly (Extended Data Fig. 8). Essentially, observed trees in the elderly showed marked asymmetry among clones, with a few large clades but also numerous 'singleton' branches. By contrast, simulations of variable population sizes generated trees with more evenly distributed coalescences among clades.

Theoretically, altered spatial patterning of HSCs could also explain the changes in phylogeny—if, for example, HSCs circulated less frequently with ageing, the bone marrow population would be less well mixed and therefore show increased density of coalescences in a bone marrow sample obtained from just a few vertebrae. To address this, we sequenced both marrow and peripheral blood HSCs from the 81-year-old subject. Peripheral blood colonies recaptured most of the expanded clades evident in marrow and at similar clonal fractions, suggesting that spatial segregation is not sufficient to explain the changes with ageing (Extended Data Fig. 7c).

## Genetic evidence for positive selection

The asymmetry of population structure in the elderly suggests that clone-specific factors could lead to differential expansion among HSCs. Positively selected driver mutations are one, but not the only, possible cause of clone-specific variation in expansion rates; for example, epigenetic change, telomere shortening and microenvironment ageing could also contribute. Furthermore, it is unclear whether driver mutations acquired randomly throughout life could lead to such a qualitative change in population structure after the age of 70. To address these questions, we used two approaches: one based on genetic analysis of selection and one based on ABC models.

For the genetic analysis, we considered synonymous mutations as selectively neutral, and used their rate to quantify whether

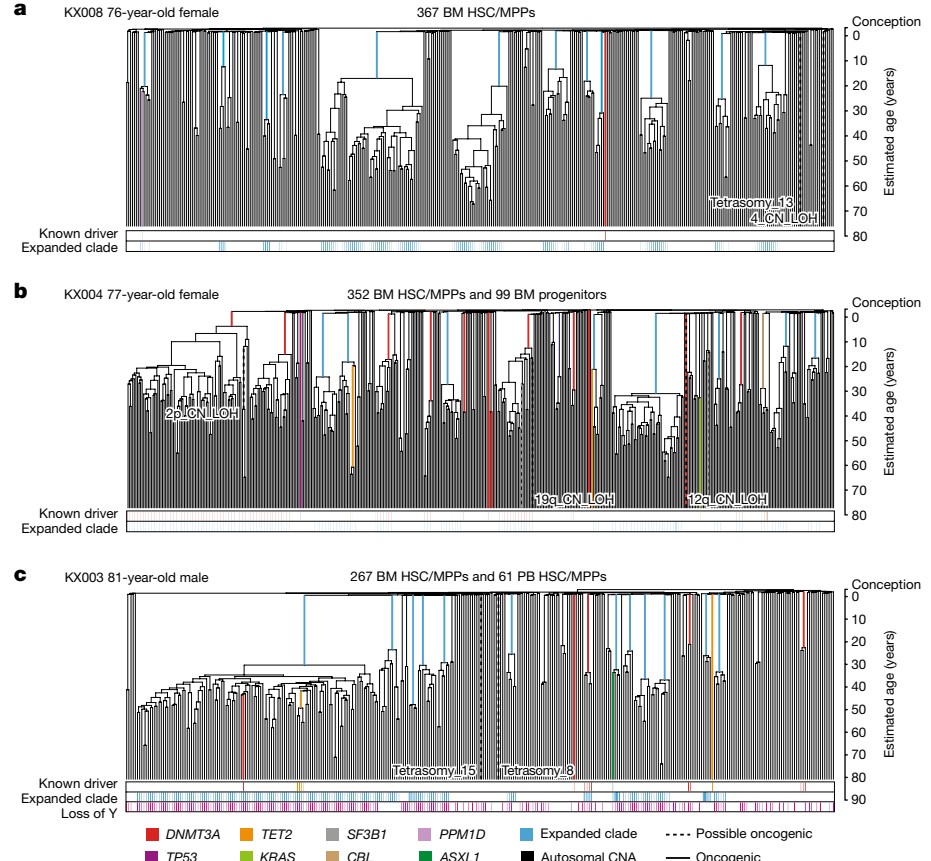

**Fig. 3 | HSPC phylogenies for three elderly adult donors. a–c**, Phylogenetic trees were constructed for a 76-year-old female donor (**a**), a 77-year-old female donor (**b**) and an 81-year-old male donor (**c**) and presented as described for

Fig. 2. The phylogeny of the fourth elderly adult donor included in the study is shown in Extended Data Fig. 5b.

non-synonymous mutations occurred at equivalent or elevated rates (d$N$/d$S$ ratio, with d$N$/d$S$ > 1.0 denoting a tilt towards positive selection)—this was undertaken gene-by-gene and across all coding genes collectively[37] (Supplementary Tables 6, 7). Estimating d$N$/d$S$ gene by gene identified three genes under positive selection: *DNMT3A* ($q$ = 2.7 × 10[-11]), *ZNF318* ($q$ = 1.2 × 10[-6]) and *HIST2H3D* ($q$ = 0.086) (Extended Data Fig. 9a, Supplementary Table 8). *DNMT3A* is a well-known myeloid cancer gene with 23 mutations in our dataset, of which 13 were in expanded clades. The age of occurrence of expanded *DNMT3A* mutations could be estimated from the trees: 2 were acquired before 10 years of age, 3 were acquired before 20 years of age, and the remainder were acquired before 40 years of age. Screening 534 acute myeloid leukaemia (AML) genomes[38,39] for variants in *ZNF318* and *HIST2H3D* identified only one possible oncogenic mutation in *ZNF318* (Supplementary Table 9), showing that although these variants are under selection in HSC/MPPs, they do not necessarily contribute to malignancy.

The genome-wide estimate of d$N$/d$S$ was significantly higher, at 1.06 (95% confidence interval 1.03–1.09; Extended Data Fig. 9b), a value that remained almost unchanged after evaluating potential biases (Supplementary Methods). This value equates to 1 in 18 (range: 1 in 34 to 1 in 12) non-synonymous coding mutations in the dataset being drivers (defined as any mutation under positive selection). Estimated d$N$/d$S$ ratios were almost identical in young and old individuals (1.06 and 1.05, respectively), suggesting that the fraction of non-synonymous mutations (approximately 5%) under positive selection does not change with age. Given that the overall mutation rate is constant, driver mutations therefore enter the HSC pool at a constant rate throughout life, and the expected number of driver mutations per cell increases linearly with age.

Converting d$N$/d$S$ ratios to estimated numbers of driver mutations revealed that each adult studied typically had more than 100 driver mutations among the colonies sequenced (Extended Data Fig. 9c). These numbers are considerably higher than the number of non-synonymous mutations identified in known cancer genes. This implies that mutations under positive selection in normal HSCs affect a wider set of genes than usually assessed, as suggested by other studies[15], and corroborates our observation that many clonal expansions in the elderly occur in the absence of mutations in known cancer genes.

Loss of chromosome Y (loss of Y) was a frequent occurrence in older males in our cohort. Loss of Y has been described in bulk blood samples from ageing men, correlating with all-cause mortality[23]. Notably, in our data, loss of Y showed frequent parallel evolution—the oldest male in our study, aged 81 years, had at least 62 independent loss of Y events across the phylogeny. Furthermore, loss of Y was significantly correlated with clonal expansion ($P$ < 0.001; Extended Data Fig. 9d)—several expanded clades in our dataset exhibited loss of Y, often without known point mutation drivers, even in younger males (Figs. 2, 3, Extended Data Fig. 5). Negative selection acting on loss of Y is likely to be less stringent than on loss of autosomes; nevertheless, that loss of Y generates significantly larger clones than euploid HSCs suggests that it is also under positive selection, corroborating mouse models showing Y-linked *KDM6C* suppresses leukaemogenesis[40].

## Modelling positive selection in HSCs

The genetic analysis suggests that positive selection is pervasive in HSCs, but whether positive selection can completely explain the observed phylogenies remains unclear. To address this, we used ABC modelling

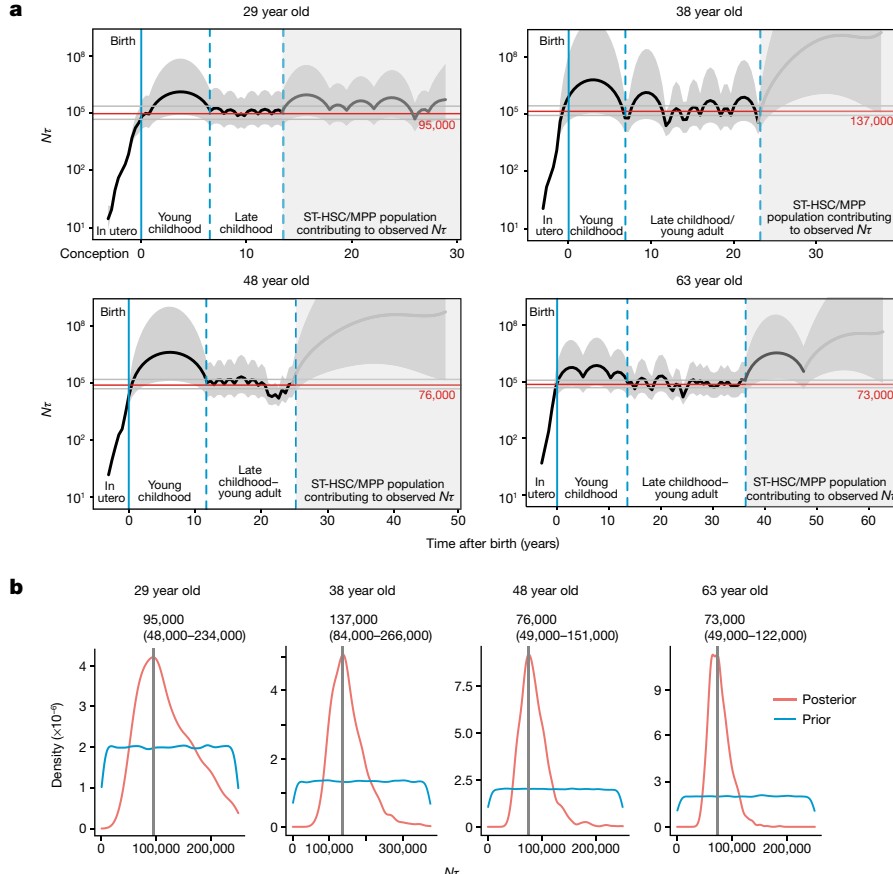

**Fig. 4 | Estimating $N\tau$ in the human long-term HSC compartment.**
**a**, Trajectory of $N\tau$ for human long-term (LT)-HSCs in the four adult donors aged over 65 years, estimated using Bayesian phylodynamics. The black line represents the estimated mean trajectory of LT-HSC $N\tau$, with the shaded grey area on either side representing the 95% credibility interval. The solid blue line is the time of birth. The dashed blue lines enclose the region of time in each individual where the trajectory is at the late childhood–young adult level. The shaded region of the plots represents the period of time before sampling over which it is likely that short-term (ST)-HSC/MPPs are contributing to the observed $N\tau$. The trajectory line is shaded dark grey in the time period where coalescent events are occurring and the trajectory probably represents the combined $N\tau$ of both long-term HSC and short-term HSC/MPP compartments. The trajectory line is shaded light grey where there is a complete absence of coalescent events and the estimates are therefore inaccurate. The red line shows the Bayesian (maximum posterior density) estimate of $N\tau$. **b**, Results from approximate Bayesian inference of population size over the first (non-shaded) part of life for each individual. The blue line represents the prior density of $N\tau$ and the red line represents the posterior density. The vertical grey line denotes the peak $N\tau$ for each donor; values are shown above each plot.

to infer HSC dynamics (Supplementary Note 1), with three aims: (1) to assess whether clone-specific positive selection can explain apparently abrupt loss in clonal diversity after age 70; (2) to estimate the rate at which driver mutations enter the HSC pool; and (3) to estimate the distribution of fitness effects, defined as the excess average growth rate per year of a clone with the driver over that of wild-type HSCs (denoted as $s$, with $s = 0$ indicating neutrality; Extended Data Fig. 10, Supplementary Methods). Simulations showed that mutations with fitness effects $s < 5\%$ are unlikely to expand to more than 1% of HSCs over a human lifespan (Extended Data Fig. 10d), so were not further considered.

ABC modelling showed that our observed trees could be closely emulated (Extended Data Figs. 11, 12) and carry sufficient information to estimate occurrence rate and fitness effects of driver mutations (Fig. 5c, Extended Data Fig. 13a). From the ABC modelling, we estimated that driver mutations enter the HSC compartment at $2.0 \times 10^{-3}$ per HSC per year. The estimate obtained from the genetic analysis, based on d$N$/d$S$, was of drivers accruing at $3.6–10.0 \times 10^{-3}$ per HSC per year, an estimate which includes drivers with $s < 5\%$ present in colonies. Thus, even though the ABC estimates derive only from branch structures of the phylogenies and the genetic analysis relies only on non-synonymous versus synonymous mutations, the two approaches concur in implying much higher rates of driver mutation occurrence than estimated from bulk sequencing studies.

The ABC modelling generates simulations of HSC clonal dynamics across a lifetime, which enables us to track how phylogenetic trees for a given avatar would appear when sampled at different ages. In these avatars, we found that there was typically a sharp change towards oligoclonal haematopoiesis after 60 years of age (Fig. 5d, Extended Data Figs. 12, 13b), demonstrating that lifelong accumulation of drivers triggering slow but inexorable exponential expansions causes abrupt loss of clonal diversity in the elderly.

For fitness effects, the distribution most consistent with the data had a preponderance of moderate-effect drivers, with $s$ in the range 5–10%, but a heavy tail of rare drivers conferring greater selective advantage ($s > 10\%$) (Fig. 5c). For the 46 largest observed clones, we could directly estimate their fitness effects from the patterns of coalescences within their clade, resulting in estimates of $s$ in the range of 10–30% (Fig. 5e, Extended Data Fig. 11b). This shows that clones without known drivers can evolve comparable selective advantages to those with classic driver mutations.

## Discussion

Our data suggest that age-associated loss of clonal diversity is orders of magnitude more pervasive than estimated from previous studies[4,5,10–12]:

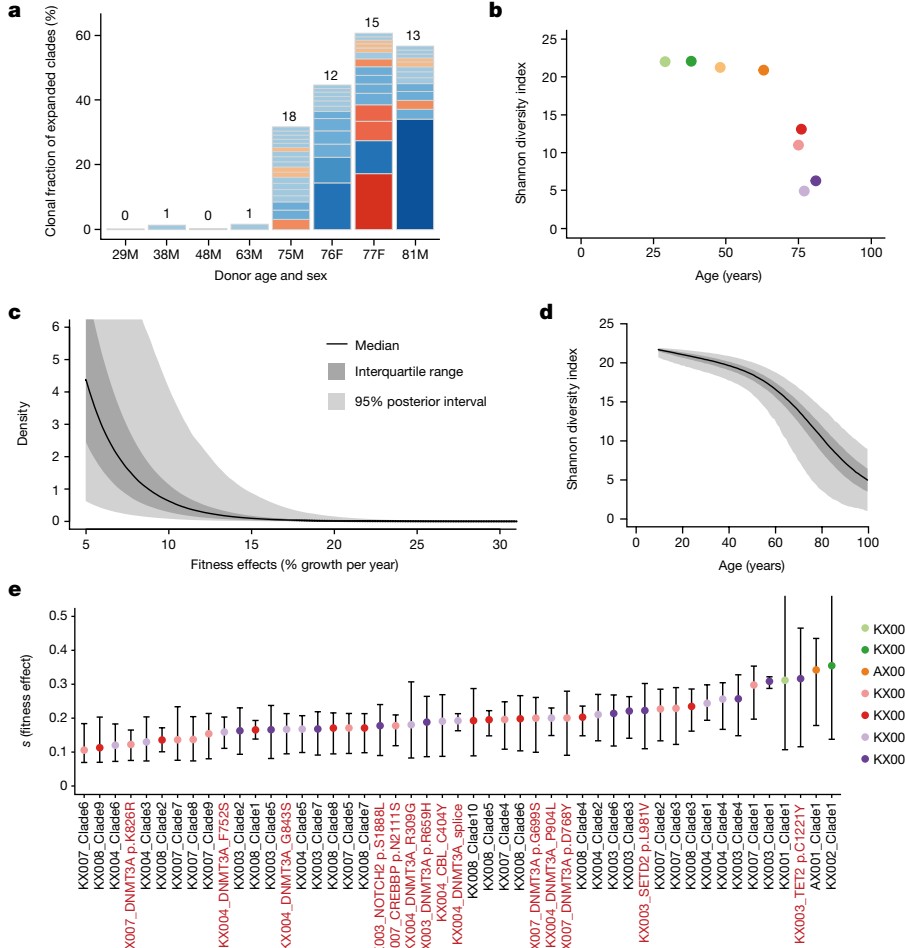

**Fig. 5 | Widespread positive selection in the HSC/MPP compartment of normal individuals. a**, Stacked bar plot of number and size of clades with a clonal fraction above 1% per individual. Blue segments denote clones with no known driver and red ones denote those with a known myeloid gene driver. Numbers above each stack denote the total number of expanded clades in that individual. **b**, Shannon diversity index calculated for each phylogeny from the number of lineages present at 100 mutations of molecular time (equivalent to the first few years after birth) and their abundance (number of colonies derived from that lineage). **c**, Distribution of fitness effects for the driver mutations entering the HSC population, as determined using the ABC modelling approach. The point estimate for the shape and rate parameters of the gamma distribution were shape = 0.47 and rate = 34 (univariate marginal maximum

posterior density estimates; Supplementary Fig. 14). The black line denotes the median, interquartile range is shown in dark grey and 95% posterior intervals are shown in light grey. **d**, Median, interquartile range, and 95% posterior intervals for Shannon diversity indices calculated yearly for 10,000 HSC population simulations run with the optimal parameter values from the ABC modelling. **e**, Fitness effects within the HSC/MPP compartment are estimated for clades with known driver mutations containing four or more HSC/MPP colonies (per cent additional growth per year). Fitness effects are also estimated for expanded clades without known drivers containing five or more HSC/MPP colonies. Error bars show the 95% credibility intervals of inferred fitness effects. Clade numbers are illustrated on the phylogenies in Extended Data Fig. 11b.

(1) the prevalence of clones at more than 1% variant allele fraction (VAF) is universal after age 70, not 10–20% prevalent as previously estimated; (2) the number of expanded clones per individual is 10–20, not 1–2; (3) the fraction of overall haematopoiesis accounted for by expanded clones is 30–60%, not 3–5%; and (4) clonal expansions have their origins in mutations that occurred decades earlier, and not in old age. Age-related phenotypes in blood include anaemia, loss of regenerative capacity, remodelling of the bone marrow microenvironment and increased risk of blood cancer—phenotypes that sharply increase in prevalence after 70 years of age. The link between known driver mutations and blood cancer risk is clear[4–6], with some evidence that expanded clones with undiscovered drivers also increase leukaemia transformation[14]. However, cancer need not be the ineluctable end-point of positive selection[41], and addressing whether expanded clones collectively driving a sizable fraction of haematopoiesis contribute to, for example, age-related loss of resilience to perturbation or microenvironment remodelling would be of interest in future studies.

From an evolutionary perspective, ageing arises because the force of natural selection rapidly reduces to zero at ages beyond which reproductive output is negligible—the 'selective shadow'[42]. Our data illustrate how an abrupt, qualitative change in tissue composition can occur within this shadow as the result of lifelong, gradual accumulation of molecular damage. HSCs acquire driver mutations with moderate fitness benefits at a steady rate throughout life, so the clonal expansions they trigger, although exponential, take decades to gather momentum. Evolutionary pressure on germline variants in genes controlling these dynamics—HSC population size and turnover, somatic mutation rates and distribution of fitness benefits—has historically acted to maintain robust, polyclonal haematopoiesis until 60–70 years of age, but no further. Thus, the distribution of fitness effects that we infer (Fig. 5c) has precisely this shape because of that evolutionary pressure—a right-shifted distribution with stronger drivers would cause sufficiently frequent blood cancers (or other deleterious phenotypes) in reproductively active humans to exert evolutionary pressure, but the

selective shadow means evolutionary pressure for driving the distribution further to the left peters out.

Clonal expansion with abrupt collapse of stem cell diversity may be a feature of ageing that generalizes beyond human blood. Many human organ systems studied to date, including skin[43], bronchus[44], endometrium[45,46] and oesophagus[47,48], show age-related expansions of clones with driver mutations. Spatial organization of these tissues shapes their clonal dynamics, but selection for the same drivers in many independent clones can generate convergent genotypes at similar overall burden to those reported here (Supplementary Note 2). Similarly, oligoclonality occurs in elderly haematopoiesis in other mammalian species, including mice[49] and macaques[50]. With such ubiquity of driver mutations—selected purely for their competitive advantage within the stem cell compartment—and with the wholesale rewiring of cellular pathways they induce, it is feasible that they may contribute to phenotypes of human ageing beyond the risk of cancer.

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

## Methods

### Samples

In order to obtain representative data from across the whole human lifespan, samples were obtained from three sources: (1) Stem Cell Technologies provided frozen mononuclear cells (MNCs) from two cord blood samples that had been collected with informed consent, including for whole-genome sequencing (catalogue no. 70007). (2) Cambridge Blood and Stem Cell Biobank (CBSB) provided fresh peripheral blood samples taken with informed consent from two patients at Addenbrooke's Hospital (NHS Cambridgeshire 4 Research Ethics Committee reference 07/MRE05/44 for samples collected before November 2019 and Cambridge East Ethics Committee reference 18/EE/0199 for samples collected from November 2019 onwards. (3) Cambridge Biorepository for Translational Medicine (CBTM) provided frozen bone marrow with or without peripheral blood MNCs taken with informed consent from six deceased organ donors. Samples were collected at the time of abdominal organ collection (Cambridgeshire 4 Research Ethics Committee reference 15/EE/0152). All samples were collected under the approved studies quoted, with informed consent to use the materials for the research undertaken here and publish the results. Details of the individuals studied and the samples they provided are illustrated in Fig. 1b, with additional information in Supplementary Table 1.

No statistical methods were used to predetermine sample size. The experiments were not randomized and the investigators were not blinded to allocation during experiments and outcome assessment.

### Isolation of MNCs from peripheral blood

MNCs were isolated using Lymphoprep density gradient centrifugation (STEMCELL Technologies), after diluting whole blood 1:1 with PBS. The red blood cell and granulocyte fraction of the blood was then removed. The MNC fraction underwent red cell lysis using 1 incubation at 4 °C for 15 min with RBC lysis buffer (BioLegend). CD34 positive cell selection of peripheral blood and cord blood MNC samples was undertaken using the EasySep human whole blood CD34 positive selection kit (STEMCELL Technologies). The kit was used per the manufacturer's instructions, but with only a single round of magnetic selection. Bone marrow MNCs did not undergo CD34 positive selection prior to cell sorting.

### Fluorescence-activated cell sorting

MNC or CD34 enriched samples were centrifuged and resuspended in PBS/3%FBS containing an antibody panel consisting of: CD3/FITC, CD90/PE, CD49f/PECy5, CD38/PECy7, CD33/APC, CD19/A700, CD34/APCCy7, CD45RA/BV421 and Zombie/Aqua. Cells were stained (30 min at 4 °C) in the dark before washing and resuspension in PBS/3%FBS for cell sorting. For all samples, HSC/MPP pool cells (Lin⁻CD34⁺CD38⁻CD45RA⁻) were sorted using either a BD Aria III or BD Aria Fusion cell sorter (BD Biosciences) at the NIHR Cambridge BRC Cell Phenotyping hub. The gating strategy is illustrated in Extended Data Fig. 1b. The HSC/MPP population was treated as a single entity and not further subclassified in the analysis. The immunophenotypic HSC/MPP population includes both long-term, intermediate-term and short-term HSCs as well as multipotent progenitors, as demonstrated functionally in xenotransplantation assays[51–54].

In a subset of individuals, a small number of HPCs (Lin⁻CD34⁺CD38⁺) were also sorted. The antibody panel used is shown in Supplementary Table 2. The HPC cells were treated as a single entity in the analysis and not further subclassified. The immunophenotypic HPC compartment includes predominantly myeloid and erythroid progenitors.

### Single-cell colony expansion in vitro

Single phenotypic HSC/MPP or HPC cells were index sorted, as above, into Nunc 96 flat-bottomed tissue culture plates (Thermofisher), containing 100 μl supplemented StemPro medium (Stem Cell Technologies) but no mouse cell feeder layer. The following supplements were added to promote proliferation and push differentiation toward granulocyte, monocyte, erythroid and natural killer cell types: StemPro Nutrients (0.035%, Stem Cell Technologies), L-glutamine (1%, ThermoFisher), penicillin-streptomycin (1%, ThermoFisher) and cytokines (SCF, 100 ng ml⁻¹; FLT3, 20 ng ml⁻¹; TPO, 100 ng ml⁻¹; EPO, 3 ng ml⁻¹; IL-6, 50 ng ml⁻¹; IL-3, 10 ng ml⁻¹; IL-11, 50 ng ml⁻¹; GM-CSF, 20 ng ml⁻¹; IL-2, 10 ng ml⁻¹; IL-7, 20 ng ml⁻¹; lipids, 50 ng ml⁻¹). Cells were incubated at 37 °C and the colonies that formed were topped up with 50 μl StemPro medium plus supplements at 14 ± 2 days as necessary. At 21 ± 2 days, a visual size assessment of colonies was undertaken prior to collection of cells for DNA extraction. Larger colonies (≥3,000 cells in size) were transferred to fresh U bottomed 96 well plate (Thermofisher). The U bottomed plates were then centrifuged (500g for 5 min), medium was discarded, and the cells were resuspended in 50 μl PBS prior to freezing at −80 °C. Smaller colonies (less than 3,000 but more than 200 cells in size) were collected into 96-well skirted LoBind plates (Eppendorf) and centrifuged (800g for 5 min). Supernatant was removed to 5–10 μl using an aspirator prior to DNA extraction on the fresh pellet. For larger colonies DNA extraction was performed using the DNeasy 96 blood and tissue plate kit (Qiagen). The Arcturus Picopure DNA Extraction kit (ThermoFisher) was used to extract DNA from the smaller colonies. Both kits were used per the manufacturer's instructions. Overall, 42–89% of the sorted HSC/MPP population in each individual produced a colony. This is much more efficient than previously used methods of colony growth in semi-solid media, where it is has been estimated that 10% of single HSC/MPPs will produce a colony. In addition, analysis of cell surface markers showed there was no immunophenotypic difference between sorted HSC/MPPs that formed colonies and were sequenced, compared to those that did not (Extended Data Fig. 1c, d).

### Whole-genome sequencing of colonies

A recently developed low-input enzymatic fragmentation-based library preparation method[45,55] was used to generate whole-genome sequencing libraries from 1–5 ng extracted DNA from each colony. Whole-genome sequencing was performed at a mean sequencing coverage of 14× (8–35×) on either the Hiseq X or the NovaSeq platforms (Illumina). BWA mem was used to align 150 bp paired end reads generated to the human reference genome (NCBI build 37; GRCh37d5).

### Variant calling

CaVEMan (used for calling SNVs) and Pindel (used for calling small indels) were run against an unmatched synthetic normal genome using in-house pipelines[45,56–58]. Outputs were then filtered using several strategies to exclude library, sequencing and mapping artefacts. Structural variants were called using GRIDDS[59], with all variants confirmed by visual inspection and by checking if they fit the distribution expected based on the SNV-derived phylogenetic tree. Autosomal CNAs and X chromosome CNAs in females were called using allele-specific copy number analysis of tumours[60] (ASCAT), which was run against a single sample selected from each individual. The matched sample was selected to have a coverage > 15×, no loss of Y and to be a singleton in the phylogenetic tree (no coalescences post-birth). The ASCAT output was manually interpreted through visual inspection. ASCAT was unable to accurately call copy number changes on the haploid sex chromosomes in males. Therefore, we also ran the breakpoint analysis algorithm BRASS[61] to generate an intermediate file containing information on binned read counts across 500-bp segments of the genome. A comparison of the mean coverage of the X and Y chromosomes was used to call X or Y CNAs in individual samples, which were then validated by visual inspection of read depth.

Full details of the variant calling strategies and filtering approaches used are described in the Supplementary Information.

## Filtering at the colony level

As outlined in the main text, we removed some colonies from the dataset due to low coverage (17 samples), being technical duplicates (34 samples) and for showing evidence of non-clonality or contamination (7 samples). We used a peak VAF threshold of < 0.4 (after the removal of in vitro variants) as well as visual inspection of the VAF distribution plots to identify colonies with evidence of non-clonality (Extended Data Fig. 2a–c). Visual inspection was particularly important in the cord blood samples where there was greater variability in the distribution of variant allele frequencies due to the lower mutation burden. In these samples the VAF threshold of < 0.4 was therefore less stringently applied.

## Validation of mutation calls

Mutation spectrums were compared between the set of shared mutations (those present in two or more colonies), which are those we have the greatest confidence in, and private mutations (present in only one sample), in which we have lower confidence (Extended Data Fig. 2e). The mutation spectrums are almost identical, providing evidence that the private mutation set does not contain excess artefacts.

## Mutation burden analysis

SNV and indel burden analysis was performed by first correcting the mutation and indel burden to a sequencing depth of 30, by fitting an asymptotic regression to the data (function NLSstAsymptotic, R package stats) (Extended Data Fig. 2d). Subsequently, linear mixed effects models were used to test for a linear relationship between age and number of SNVs or number of indels (function lmer, R package lme4). Number of mutations or indels per colony was regressed using log-likelihood maximization and age as a fixed effect, with the interaction between age and donor as a random effect. Progenitor samples were excluded from this analysis.

We also performed linear regression of age and mutation burden as above broken down by age range. Using data from the two cord blood donors and the youngest adult age 29 gives a rate of mutation accumulation of 17.56 per year (95% confidence interval 17.32–17.78). Using data from just the 3 younger donors aged 29, 38, 48 and 63 gives a rate of mutation accumulation of 17.21 per year (95% confidence interval 16.12–18.3). Linear regression of age and mutation burden using the 5 older donors aged 63, 75, 76, 77 and 81 gives a rate of mutation accumulation of 18.84 per (95% confidence interval 16.82–20.86). These results are very consistent over different phases of life.

## Telomere analysis

Telomere length for each colony sequenced on the Hiseq platform (corresponding to the telomere length in the founding HSC/MPP or HPC) was estimated from the ratio of telomeric to sub-telomeric reads using the algorithm Telomerecat[62]. Colonies sequenced on the Novaseq platform could not be used as telomeric reads were removed by a quality control step prior to bam file creation.

Linear mixed effects models were used to test for a linear relationship between age and telomere length across all the adults. Cord blood samples were excluded due to possible non-linearity of the relationship between age and telomere length in very early life[63].

Normality testing of the telomere length distributions was performed in R using the Shapiro-Wilk normality test (function shapiro.test) and visualized using Q–Q plots and density plots (functions ggqqplot and ggdensity). The percentage of outlying HSC/MPPs per individual was calculated using the interquartile range criterion (all samples outside the interval $[q_{0.25} - 1.5 \times IQR, q_{0.75} + 1.5 \times IQR]$ are considered as outliers; IQR is the interquartile range; $q_{0.25}$ and $q_{0.75}$ are the 25th and 75th centiles respectively) (function boxplot.stats). For all individuals, the only outliers in the data had longer than expected rather than shorter than expected telomeres.

## Construction of phylogenetic trees

The key steps to generate the phylogenies shown in Figs. 2, 3 and Extended Data Fig. 5 are as follows:

1. *Generate a 'genotype matrix' of mutation calls for every colony within a donor.* Our protocol, based on whole-genome sequencing of single cell-derived colonies, generates consistent and even coverage across the genome, leading to very few missing values within this matrix (ranging from 0.005 to 0.034 of mutated sites in a given colony across different donors within our cohort). This generates a high degree of accuracy in the constructed trees.

2. *Reconstruct phylogenetic trees from the genotype matrix.* This is a standard and well-studied problem in phylogenetics. The low fraction of the genome that is mutated in a given colony (<1 per million bases) coupled with the highly complete genotype matrix mean that different phylogenetics methods produce reassuringly concordant trees. We used the MPBoot algorithm[64] for the tree reconstruction, as it proved both accurate and computationally efficient for our dataset.

3. *Correct terminal branch lengths for sensitivity to detect mutations in each colony.* The trees generated in the previous step have branch lengths proportional to the number of mutations assigned to each branch. For the terminal branches, which contain mutations unique to that colony, variable sequencing depth can underestimate the true numbers of unique mutations, so we correct these branch lengths for the estimated sensitivity to detect mutations based on genome coverage.

4. *Make phylogenetic trees ultrametric.* After step 3, there is little more than Poisson variation in corrected mutation burden among colonies from a given donor. Since these colonies all derived from the same timepoint, we can normalize the branch lengths to have the same overall distance from root to tip (known as an ultrametric tree). We used an 'iteratively reweighted means' algorithm for this purpose.

5. *Scale trees to chronological age.* Since mutation rate is constant across the human lifespan, we can use it as a molecular clock to linearly scale the ultrametric tree to chronological age.

6. *Overlay phenotypic and genotypic information on the tree.* The tip of each branch in the resulting phylogenetic tree represents a specific colony in the dataset, meaning that we can depict phenotypic information about each colony underneath its terminal branch (the coloured stripes along the bottom of Figs. 2, 3). Furthermore, every mutation in the dataset is confidently assigned to a specific branch in the phylogenetic tree. This means that we can highlight branches on which specific genetic events occurred (such as *DNMT3A* or other driver mutations).

A complete explanation of the steps undertaken in each of these stages, comparisons of different methods available and notes on the different approaches to validate the phylogenetic trees are available in the Supplementary Information.

## Inferring HSC population size trajectory

The R package phylodyn[65] provides a well-established approach to inferring historic population size trajectories from the pattern of coalescent events (more specifically the density of these events in historic time blocks) in a phylogenetic tree created from a random sample of individuals in the population. Its use has been pioneered in pathogen epidemiology[65] and has also been previously applied to HSPC data from a single individual[66]. Extended Data Figs. 6b, 7b show how phylodyn can accurately recover simulated population trajectories using sample sizes similar to those we have used and illustrates how the number of coalescent events in a given time window in the tree informs on population size through time (assuming a constant rate of HSC symmetric cell division and a neutrally evolving population). We used phylodyn to infer historic changes in LT-HSC $N\tau$ from the ultrametric phylogenies of the four youngest adults in the cohort (Fig. 4a).

Simulations of complete HSC populations from conception to the age of sampling were performed for each individual using the R package rsimpop[67] (https://github.com/NickWilliamsSanger/rsimpop). Rsimpop utilizes a birth-to-death model with specified somatic mutation accumulation rate and symmetric cell division rate, to simulate a complete HSC population. Each cell within the population has a rate of symmetric division and a rate of symmetric differentiation (or death). Asymmetric divisions do not impact on the HSC phylogeny and are not accounted for in the model. Full details are available in the Supplementary Information.

### Approximate Bayesian computation

We evaluated several ABC models of population dynamics, including both models of selectively neutral genetic drift within the HSC compartment (assessing the effects of changes in population size on phylogenetic trees) and models of positive selection among the HSCs. An additional motivation for performing the Bayesian inferences on neutral models was to enable us to perform posterior predictive checks (PPC), with the aim of deciding whether the observed phylogenies are compatible with neutral models. Note that a separate donor-specific posterior distribution was generated (sampled) for each donor (donor-specific ABC), and a separate donor-specific posterior predictive p-value was computed for each donor (donor-specific PPC). Each donor-specific ABC for the neutral model was performed using the ABC rejection method (R package abc)[68–70]. Further details, together with detailed mathematical exposition, are provided in Supplementary Information.

We used the population trajectory from phylodyn to identify the time period prior to the increase related to a ST-HSC/MPP contribution, and the timing of the midlife and late-life fold-change in $N\tau$ (Fig. 4a, Supplementary Fig. 12). We used our data to inform our choices for the time between symmetric cell divisions, which was set at one year (after the initial population growth phase in the first few years of life). We set the rate of mutation accumulation at 15 mutations per year with an additional 1 mutation for every cell division (both of these were drawn from a Poisson distribution centred on the input value).

### Analysis of driver variants

Variants identified were annotated with Variation Annotation Generator (VAGrENT) (https://github.com/cancerit/VAGrENT) to identify protein coding mutations and putative driver mutations in each dataset[71–77]. Supplementary Table 4 lists the 17 genes we have used as our top clonal haematopoiesis genes (those identified by Fabre et al. as being under positive selection in a targeted sequencing dataset of 385 older individuals[78]), whose 'oncogenic' and 'possible oncogenic' mutations (as assessed independently by E.M. and P.J.C.) are shown in Figs. 2, 3 and Extended Data Fig. 5. In the same table, we also list a more extensive list of 92 clonal haematopoiesis genes derived from the literature[37, 71, 72] that were interrogated. In order to explore a wider set of cancer gene mutations we used the 723 genes listed in Cosmic's Cancer Gene Census (https://cancer.sanger.ac.uk/census).

### d$N$/d$S$ analysis

We used the R package dndscv[37] (https://github.com/im3sanger/dndscv) to look for evidence of positive selection in our dataset (https://github.com/emily-mitchell/normal_haematopoiesis/5_dNdS/scripts/all_DNDScv_final.Rmd). The dndscv package compares the observed ratio of missense, truncating and nonsense to synonymous mutations with that expected under a neutral model. It incorporates information on the background mutation rate of each gene and uses trinucleotide-context substitution matrices. The approach provides a global estimate of selection in the coding variant dataset (Supplementary Table 7), from which the number of excess protein coding, or driver mutations can be estimated. In addition, it identifies specific genes that are under significant positive selection. Further details are provided in Supplementary Information.

### Chromosome Y loss analysis

We observe a series of phylogenetic trees from male individuals in which some clades have lost the Y chromosome. By eye, these clades seem to be larger than clades that have not lost Y. To test this formally, we use a randomization or Monte Carlo test to define the null expected distribution of clade size. For each Monte Carlo iteration, we draw branches of the phylogenetic tree at random—one random branch for each observed instance of Y loss. These branches are sampled (with replacement) from the set of all extant branches at the matched time-point in that individual, and the eventual clade size of that draw measured. For each simulation, the geometric mean (to allow for the log-normality of observed clade sizes) of clade sizes is calculated. We can then compare the distribution of geometric means from the Monte Carlo draws with the observed geometric mean.

### Analysis of AML

Mutations in ZNF318 and HIST2H3D were identified in recently described tumour whole-genome sequencing data from 263 patients with AML and myelodysplastic syndromes seen at Washington University School of Medicine in St Louis[38]. Sequencing, initial processing using the hg38 human reference genome, and full variant calling details for this dataset were described previously[38]. Briefly, identification of SNVs and indels in ZNF318 and HIST2H3D was performed with Varscan2 run in SNV and indel mode using custom parameters to enhance sensitivity (--min-reads2 = 3, --min-coverage = 6, --min-var-freq = 0.02, and --p-value = 0.01), along with indel callers Pindel and Manta using default parameters. Variant calls identified via these approaches were merged and harmonized using a custom python script and annotated with VEP using Ensembl version 90. Only one potentially pathogenic variant was found in ZNF318. All identified variants are listed in Supplementary Table 8, the majority are likely to be rare germline variants as the sequencing strategy did not incorporate a matched normal sample.

In addition, there were no variants in ZNF318 or HIST2H3D reported in 200 cases of AML in the TCGA dataset[79], compared to 51 cases with DNMT3A mutations. Similarly of 71 AML cases in the JAMA study[39] 23 cases had DNMT3A mutations but none had ZNF318 or HIST2H3D mutations. Both studies used a tumour/normal WGS approach and reported all somatic variants identified as part of their supplementary datasets.

### Software

Programs and software used in the data analyses: R: version 3.6.1, BWA-MEM: version 0.7.17 (https://sourceforge.net/projects/bio-bwa/), cgpCaVEMan: version 1.11.2/1.13.14/1.14.1 (https://github.com/cancerit/CaVEMan), cgpPindel: version 2.2.5/3.2.0/3.3.0 (https://github.com/cancerit/cgpPindel), Brass: version 6.1.2/6.2.0/6.3.0/6.3.4 (https://github.com/cancerit/BRASS), ASCAT NGS: version 4.2.1/4.3.3 (https://github.com/cancerit/ascatNgs), VAGrENT: version 3.5.2/3.6.0/3.6.1 (https://github.com/cancerit/VAGrENT), GRIDSS: version 2.9.4 (https://github.com/PapenfussLab/gridss), MPBoot: version 1.1.0 (https://github.com/diepthihoang/mpboot), cgpVAF: version 2.4.0 (https://github.com/cancerit/vafCorrect), dNdScv: version 0.0.1 (https://github.com/im3sanger/dndscv), Telomerecat: version 3.4.0 (https://github.com/jhrf/telomerecat), Rsimpop: version 2.0.4 (https://github.com/NickWilliamsSanger/rsimpop), Phylodyn: version 0.9.02 (https://github.com/mdkarcher/phylodyn) and FlowJo: version 10.

### Reporting summary

Further information on research design is available in the Nature Research Reporting Summary linked to this paper.

## Data availability

Raw sequencing data are available on the European Genome–Phenome Archive under accession number EGAD00001007851). The main data

needed to reanalyse or reproduce the results presented are available on Mendeley at https://data.mendeley.com/datasets/np54zjkvxr/1. See Supplementary Information for a guide to the specific file and folder structure.

## Code availability

All scripts, some smaller data matrices and custom code, including step-by-step notebooks detailing the phylogenetic reconstruction for each research subject, are available on github at https://github.com/emily-mitchell/normal_haematopoiesis/.

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

**Acknowledgements** This work was supported by the WBH Foundation. Investigators at the Sanger Institute are supported by a core grant from the Wellcome Trust. P.J.C. was a Wellcome Trust Senior Clinical Fellow (WT088340MA) until 2020. N.M. was supported by a DFG Research Fellowship (ME 5209/1-1). Work in the D.G.K. laboratory is supported by a Bloodwise Bennett Fellowship (15008), a Cancer Research UK Programme Foundation Award (DCRPGF\100008), and a European Research Council Starting Grant (ERC-2016-STG-715371). Work in the A.R.G. laboratory is supported by the Wellcome Trust, Bloodwise, Cancer Research UK, the Kay Kendall Leukaemia Fund, and the Leukemia and Lymphoma Society of America. Work in the E.L. laboratory is supported by a Wellcome Trust Sir Henry Dale Fellowship, BBSRC and a European Haematology Association Non-Clinical Advanced Research Fellowship. The E.L. and A.R.G. laboratories are supported by a core support grant from the Wellcome Trust and Medical Research Council to the Cambridge Stem Cell Institute. K.M. is supported by the Chan-Zuckerberg Initiative. K.C. is supported by a Wellcome Investigator award (210755/Z/18/Z). We thank the Cambridge Blood and Stem Cell Biobank; the Cambridge Biorepository for Translational Medicine for access to human bone marrow and matched peripheral blood; and the Cambridge NIHR BRC Cell Phenotyping Hub for their flow cytometry services and advice. We acknowledge further assistance from the National Institute for Health Research Cambridge Biomedical Research Centre and the Cambridge Experimental Cancer Medicine Centre. We are grateful to the donors, families of donors and the Cambridge Biorepository for Translational Medicine for the gift of tissues. We thank T. Ley for his help with analysis of AML genomes. For the purpose of Open Access, the author has applied a CC-BY public copyright license to any Author Accepted Manuscript version arising from this submission. This research was supported by the NIHR Cambridge Biomedical Research Centre (BRC-1215-20014). The views expressed are those of the authors and not necessarily those of the NIHR or the Department of Health and Social Care.

**Author contributions** P.J.C., E.L. and E.M. designed the experiments. E.L., J.N., A.R.G. and P.J.C. supervised the project. E.M. performed cell sorting and colony growth with advice from N.M., E.F.C., D.G.K. and E.L. and assistance from M.D. and D.H. E.M. performed data curation, data analyses, and modelling with advice and assistance from M.S.C., N.W., K.J.D., T.M., T.H.H.C., H.M., H.L.-S., M.A.F., F.A., A.C., G.S.V., I.M., M.R.S., A.R.G., J.N., E.L. and P.J.C. N.W. and J.N. developed the simulator used for HSC population modelling. K.J.D. developed and performed posterior predictive model checking. H.J. performed structural variant analysis. T.M. performed loss-of-Y analysis; M.S.C. performed phylogeny benchmarking; D.H.S. performed AML genome analysis. K.M., K.C. and K.S.P. collected and processed samples and J.B. assisted with project management. E.M., J.N., E.L. and P.J.C. wrote the manuscript. All authors reviewed and edited the manuscript.

**Competing interests** D.H.S. has received consultancy fees from Wugen. G.S.V. has received consultancy fees from STRM.BIO and is a remunerated member of AstraZeneca's Scientific Advisory Board. D.G.K. has received research funding from STRM.BIO.

**Additional information**
**Correspondence and requests for materials** should be addressed to Jyoti Nangalia, Elisa Laurenti or Peter J. Campbell.

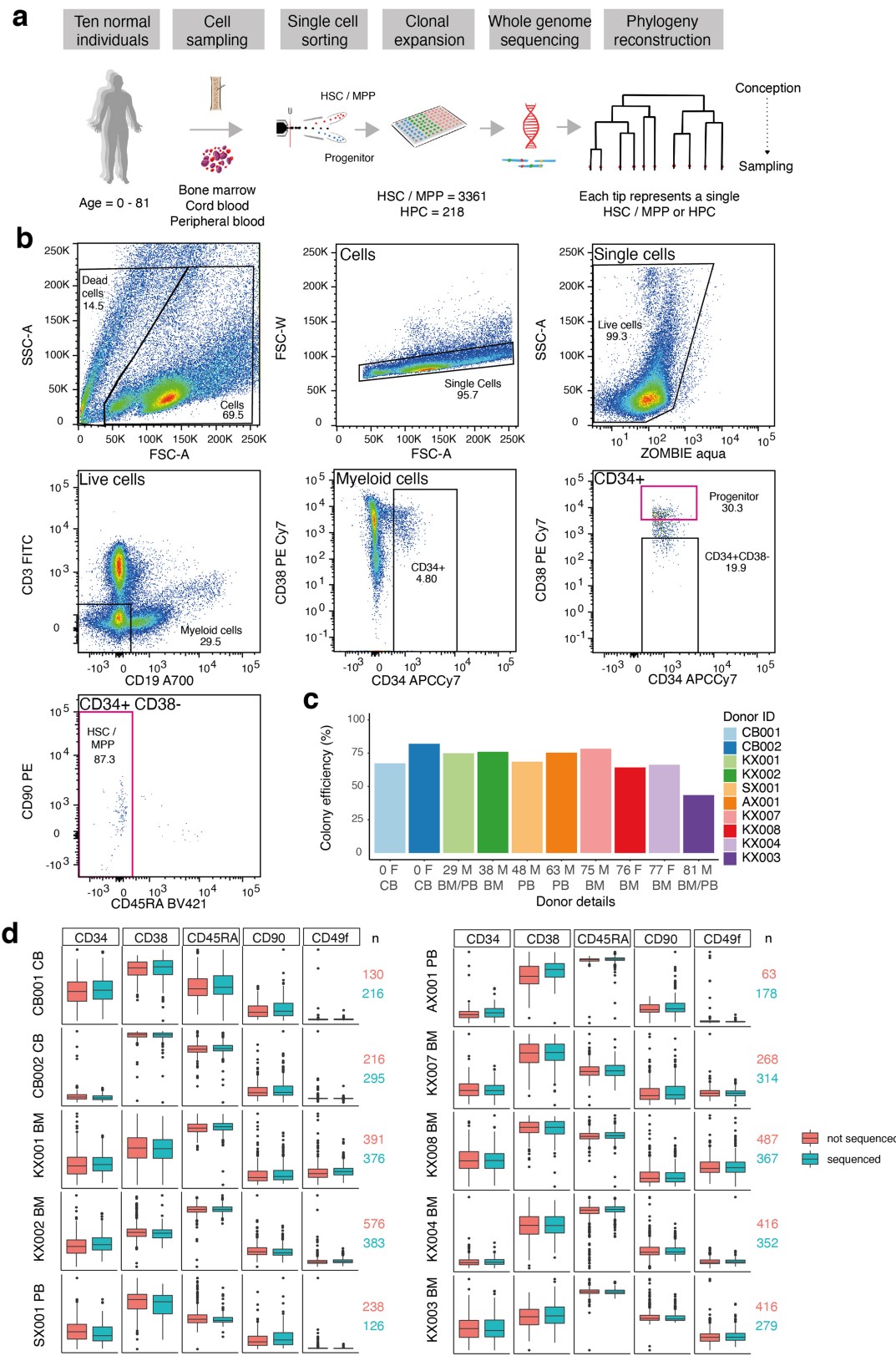

**Extended Data Fig. 1 | Flow-sorting strategy for single HSC/MPP and HPC cells. a**, Experimental approach. **b**, Sorting of single human HSC/MPP and HPCs from cord blood, peripheral blood and bone marrow. Cells were stained with the panel of antibodies in Supplementary Table 1 then single HSC/MPP or HPCs were index sorted according to the strategy depicted into individual wells of 96 well plates. Image created with FlowJo v10. **c**, Colony forming efficiency per individual of all single HSPCs sorted. **d**, Box-and-whisker plots showing fluorescence intensity for different cell surface markers used to

define human HSCs, with different patients in rows. CD90 and CD49f are markers used to define the short and long term HSC subsets, which were included in our panel but were not used in sorting. Cells that produced colonies large enough to sequence are shown in teal; cells that did not form large enough colonies to sequence are in orange. The horizontal lines denote the median, the boxes the interquartile range and the whiskers the range. The number (n) of 'sequenced' (teal) and 'not sequenced' (orange) colonies included for each individual are shown to the right of each panel.

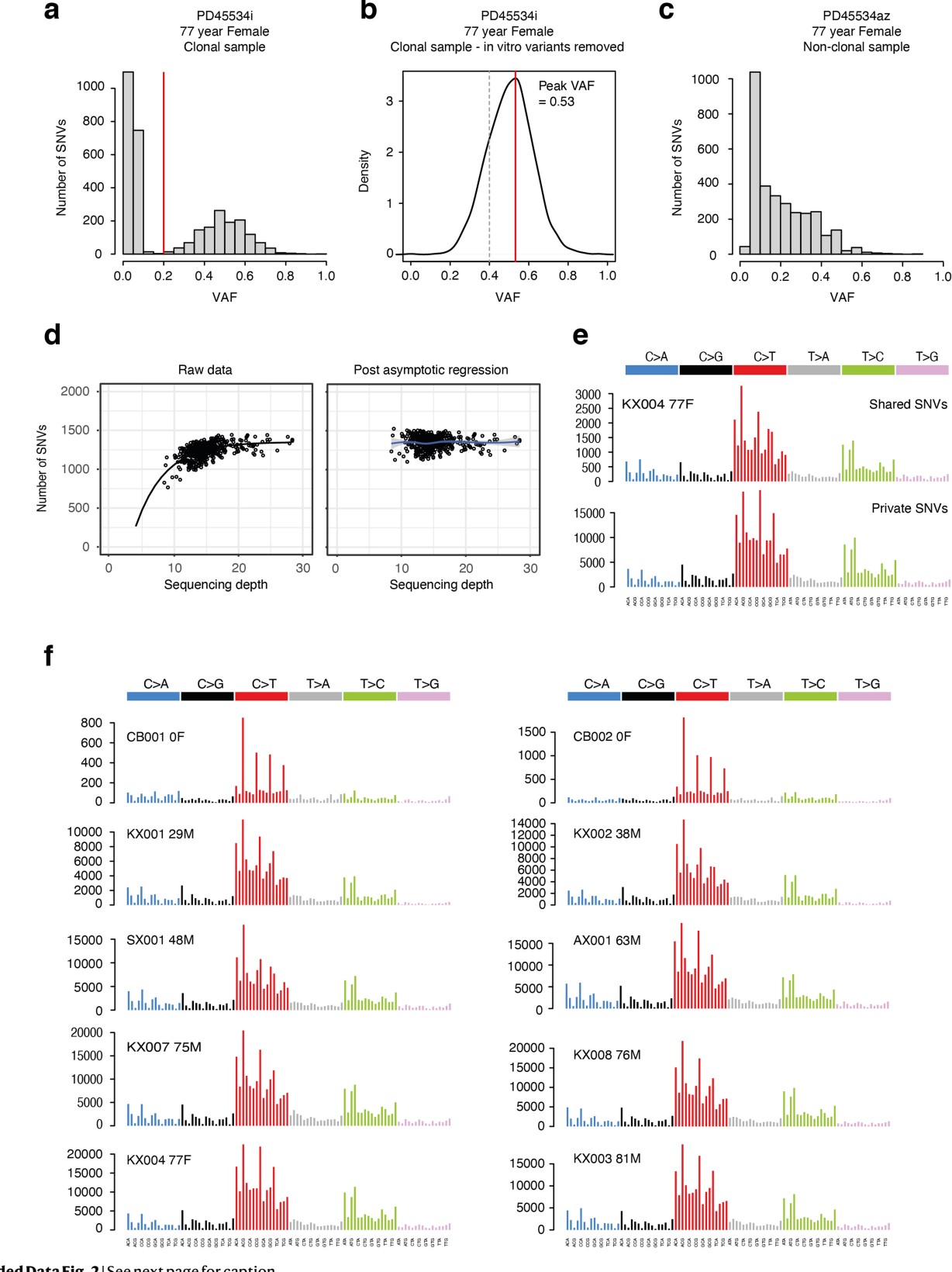

**Extended Data Fig. 2** | See next page for caption.

**Extended Data Fig. 2 | Quality assurance of mutation calls. a**, Histogram of VAFs for a typical sample in the dataset, showing a tight distribution around 50%, as expected for an uncontaminated clonal sample derived from a single cell. The variants with VAFs < 0.2 represent *in vitro* acquired mutations and sequencing artefacts and were removed using a VAF-based filtering strategy with a cut off of 0.2 (red line). **b**, VAF distribution of variants after filtering steps had been applied. The red line shows the peak VAF and the dashed grey line shows the threshold peak VAF for excluding samples as being non-clonal / contaminated. **c**, Histogram of VAFs for a colony that was seeded by 2 cells showing a median VAF around 25%. Colonies showing evidence of non-clonality in this way were excluded from downstream analysis using a peak VAF cut off of 0.4. **d**, Left-hand plot shows the relationship between raw mutation counts per colony for one individual post filtering and sequencing depth. The black line depicts an asymptotic regression line fitted to the raw data. Right-hand plot shows the adjusted mutation burdens per colony after asymptotic regression correction. **e**, Trinucleotide context mutation spectra of private (top plot) and shared variants (bottom plot) for one individual. The spectra are extremely similar, showing the variant filtering strategy used is robust and prevents excess artefacts in the private variant set. **f**, Trinucleotide mutation spectrums for each individual created from all variants post filtering. The results are consistent between the two cord blood donors and all the adult donors.

**a** MPBoot derived phylogeny with unadjusted branch lengths

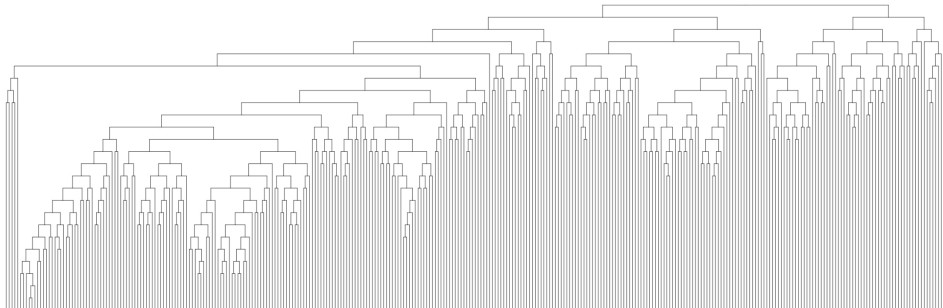

**b** Phylogeny with branch lengths adjusted using raw mutation numbers

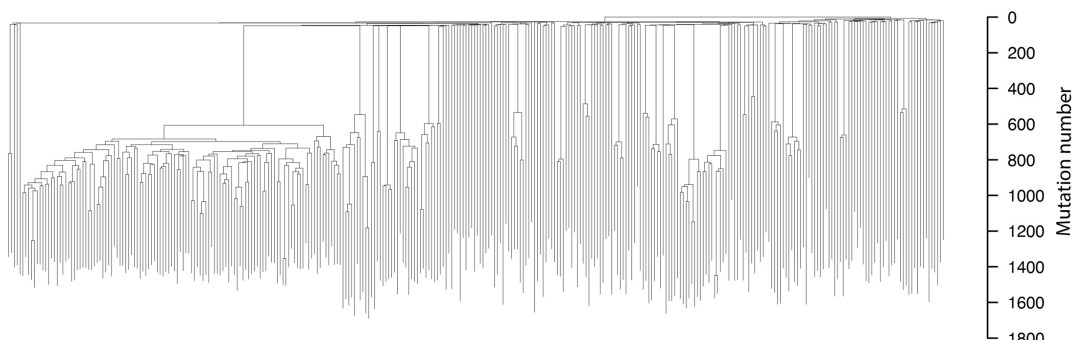

**c** Phylogeny with branch lengths corrected for sequencing depth using sensitivity

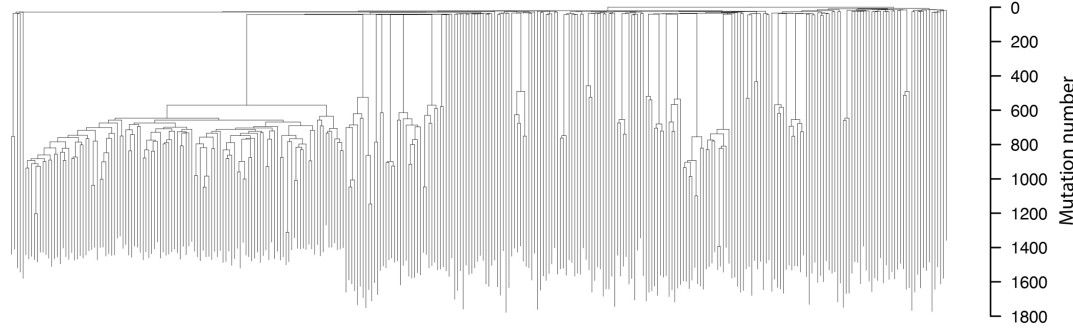

**d** Ultrametric conversion to phylogeny with equal branch lengths

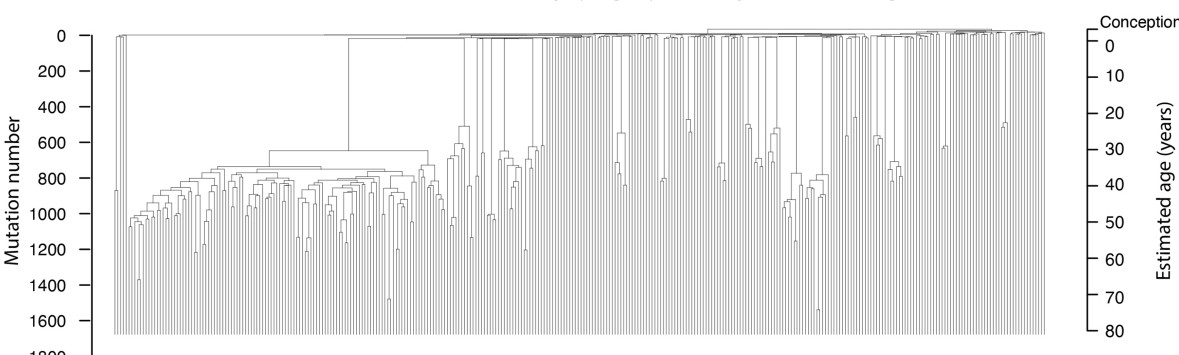

**Extended Data Fig. 3 | Approach to phylogeny construction. a**, Raw phylogeny for KX003 (81-year male) derived directly from *MPBoot*. The input to *MPBoot* is a genotype matrix of all variant calls shared by more than 1 colony from an individual. **b**, Phylogeny with edge lengths proportional to the number of mutations assigned to the branch using original count data and the *tree_mut* package. **c**, Phylogeny with raw mutation count branch lengths adjusted for sequencing depth of the sample using sensitivity for germline variant calling. **d**, Phylogeny with adjusted branch lengths converted to ultrametric form (equal branch lengths). One axis shows mutation number, the other axis shows the equivalent estimated age in years, which is possible due to the linear accumulation of mutations in HSPCs with time. All tips end at age 81, the age at the time of sampling.

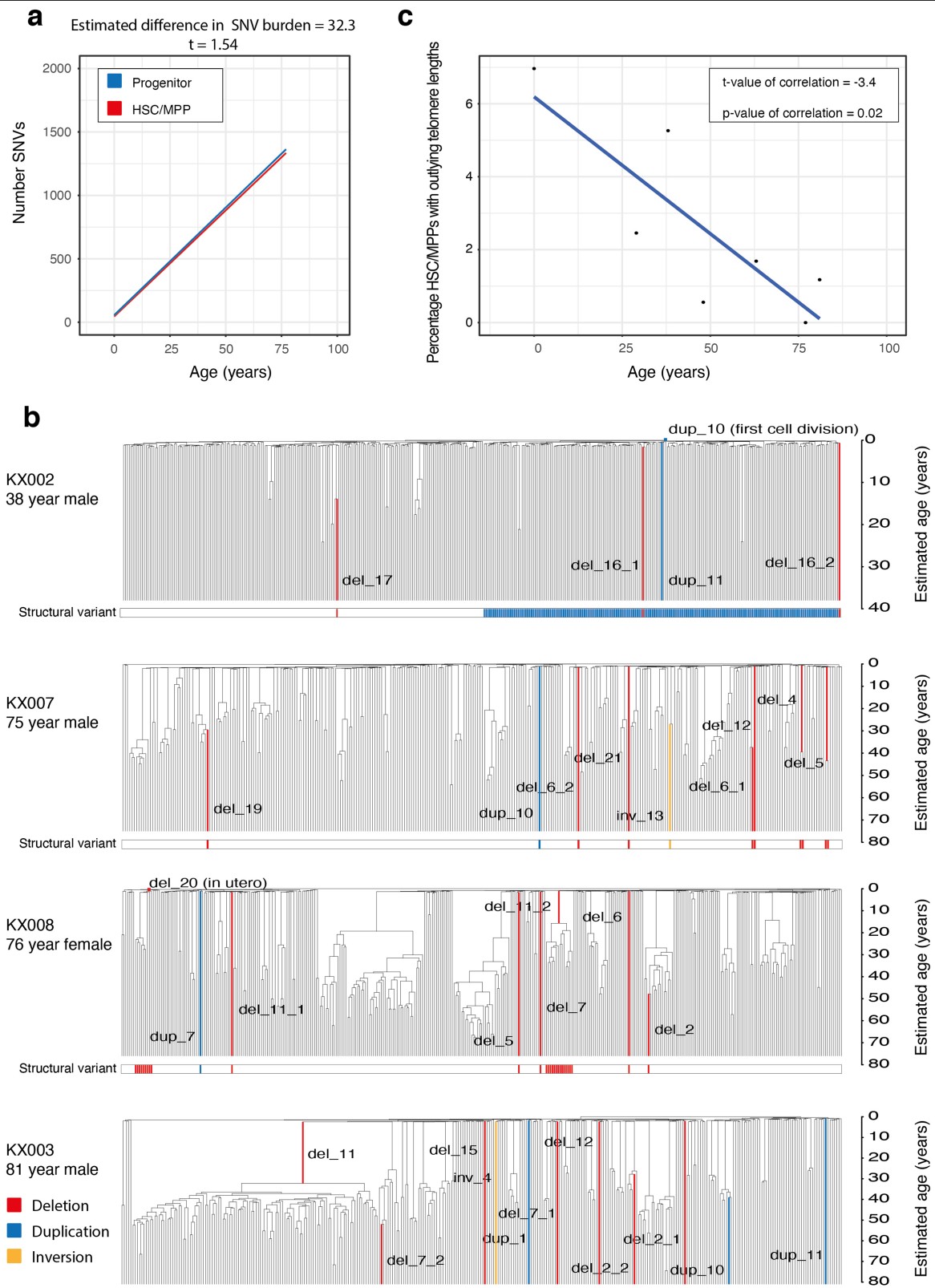

**Extended Data Fig. 4 | Mutational burden. a**, Regression of number of single nucleotide variants (SNVs) in HSCs (red line) compared to HPCs (blue line). Grey shading indicates the 95% CI. The estimated difference in burden, together with the t-value is above the plot. The t-value of 1.54 demonstrates non-significance of the difference. **b**, Phylogenies depicted for the individuals with clonally expanded structural variants (SVs). The bar at the bottom highlights cells with one of the three classes of structural variant. The exact variant breakpoints can be found in Supplementary Table 3. **c**, Plot showing the percentage of HSC/MPP cells that have outlying telomere lengths per individual. Outliers were identified using the Interquartile Range criterion. There were no outliers with shorter than expected telomeres in any individuals, such that this data only reflects the percentage of cells with longer than expected telomeres. The blue line shows a regression of percentage outlying telomere lengths with age. This shows a significant negative correlation (t-value and p-value shown).

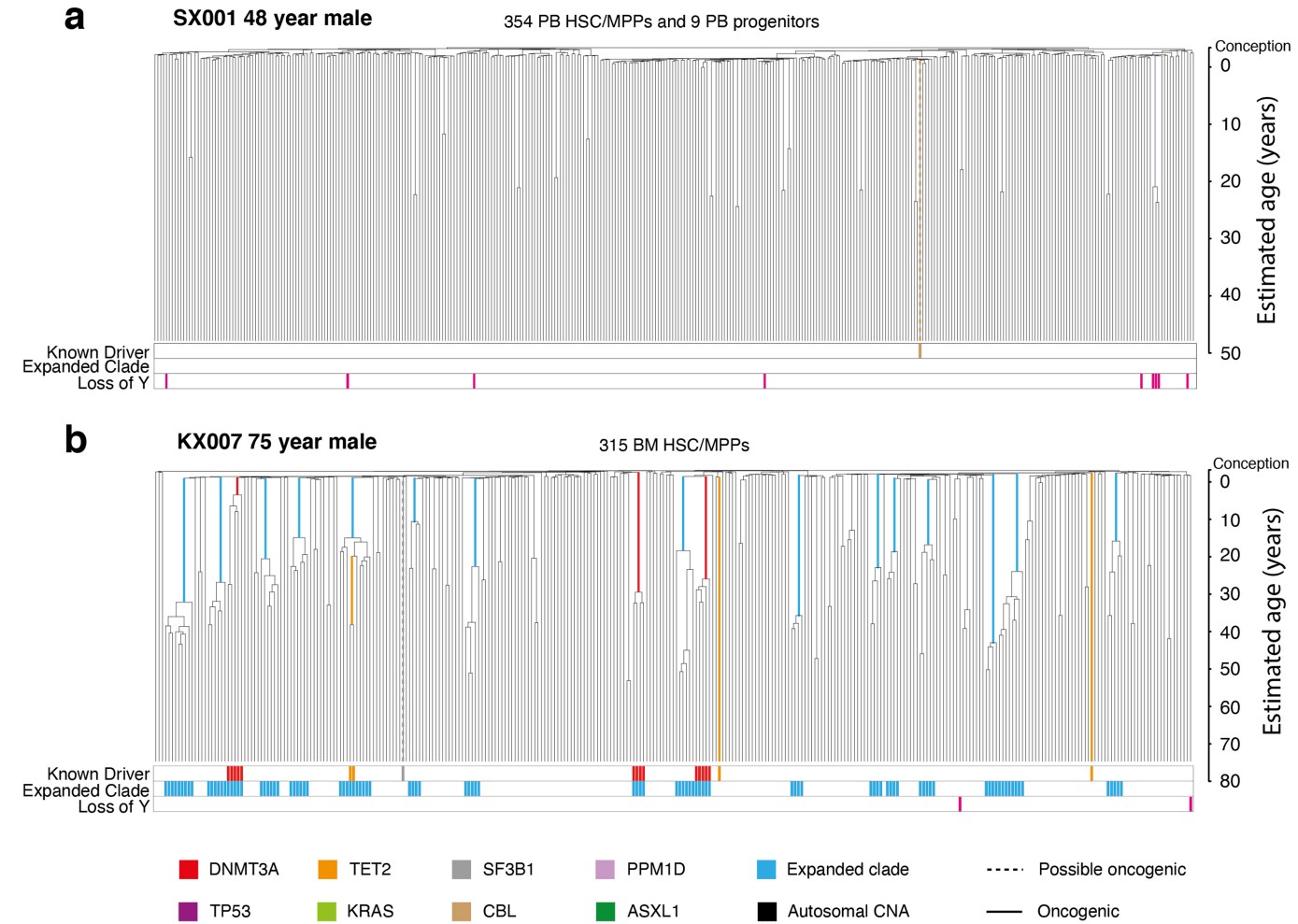

**Extended Data Fig. 5 | Additional HSPC phylogenies for one young and one elderly adult donor.** Phylogenetic trees were constructed and presented as described for Fig. 2.

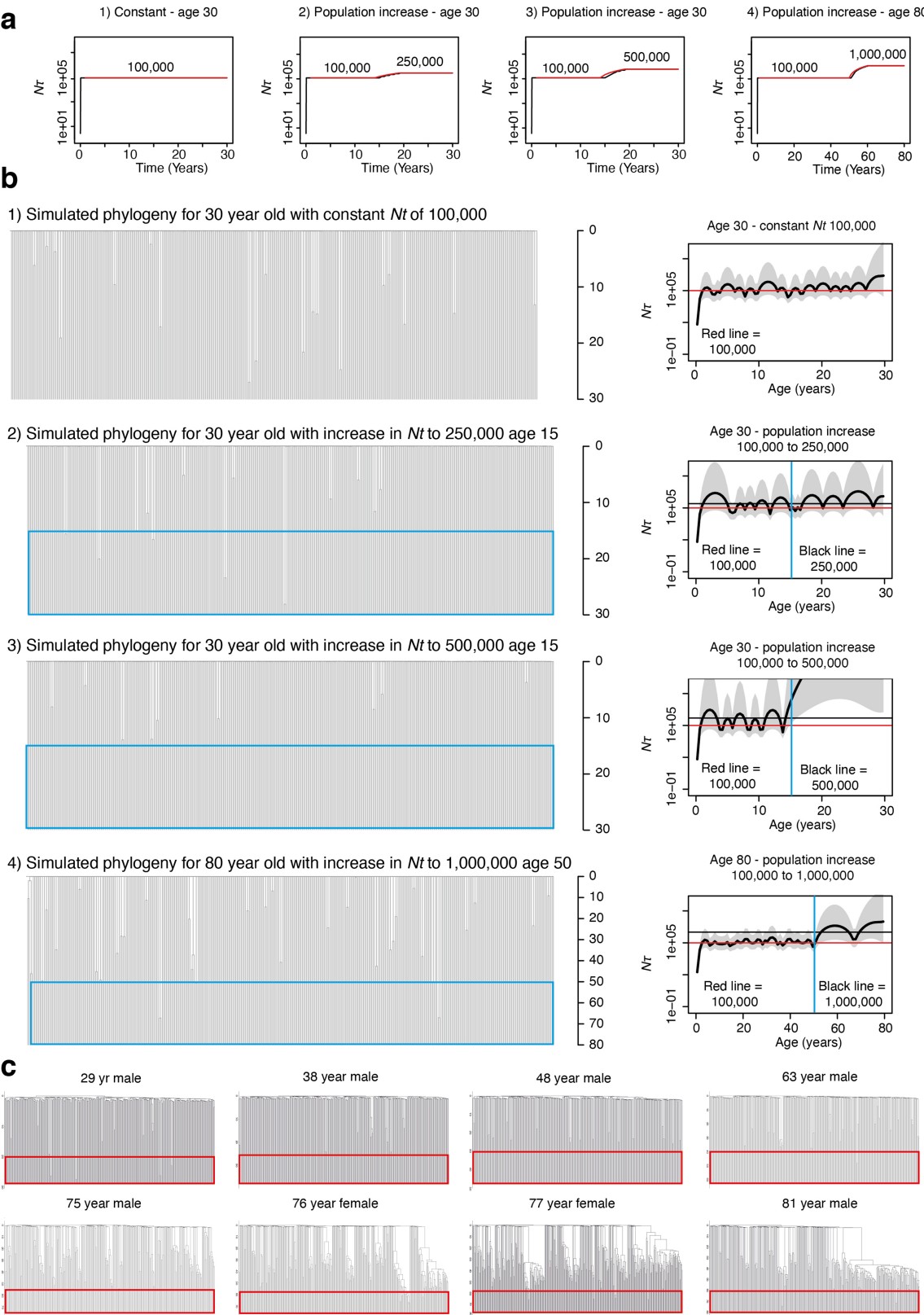

**Extended Data Fig. 6** | See next page for caption.

**Extended Data Fig. 6 | Interpretation of young adult HSPC phylogenies.**
**a**, Trajectories of $N\tau$ used as input to *rsimpop* for the simulations to create phylogenies in b. Note the Y axis depicting $N\tau$ is on a log scale. **b**, Phylogenies created by randomly sampling 380 cells from the final full simulated population of between 100,000 cells (Phylogeny 1) and 1,000,000 cells (Phylogeny 4). Phylogenies 1 to 3 are derived from simulations of the HSC population in a 30-year-old, while phylogeny 4 is derived from a simulation of the HSC population in an 80-year-old. Each simulation has an initial $N\tau$ of 100. In all cases $N\tau$ is the same as the population size ($N$) as the generation time ($\tau$) in all simulations is fixed at 1. The blue boxes indicate the period of time in which the population size is increased. The *phylodyn* trajectories to the right of each simulated phylogeny use the pattern of coalescent events to recover the input trajectories for $N\tau$. The blue line marks the time of change in $N\tau$. In all cases the initial part of the trajectory is able to correctly estimate $N\tau$ at 100,000. However, in Phylogeny 3 where there is a complete absence of coalescent events once the population size is increased, *phylodyn* loses resolution and wildly overestimates the value of $N\tau$. **c**, Real trees with red boxes highlighting the last 10–20 years prior to sampling, where the relative number of coalescent events is decreased (meaning the estimated $N\tau$ is larger).

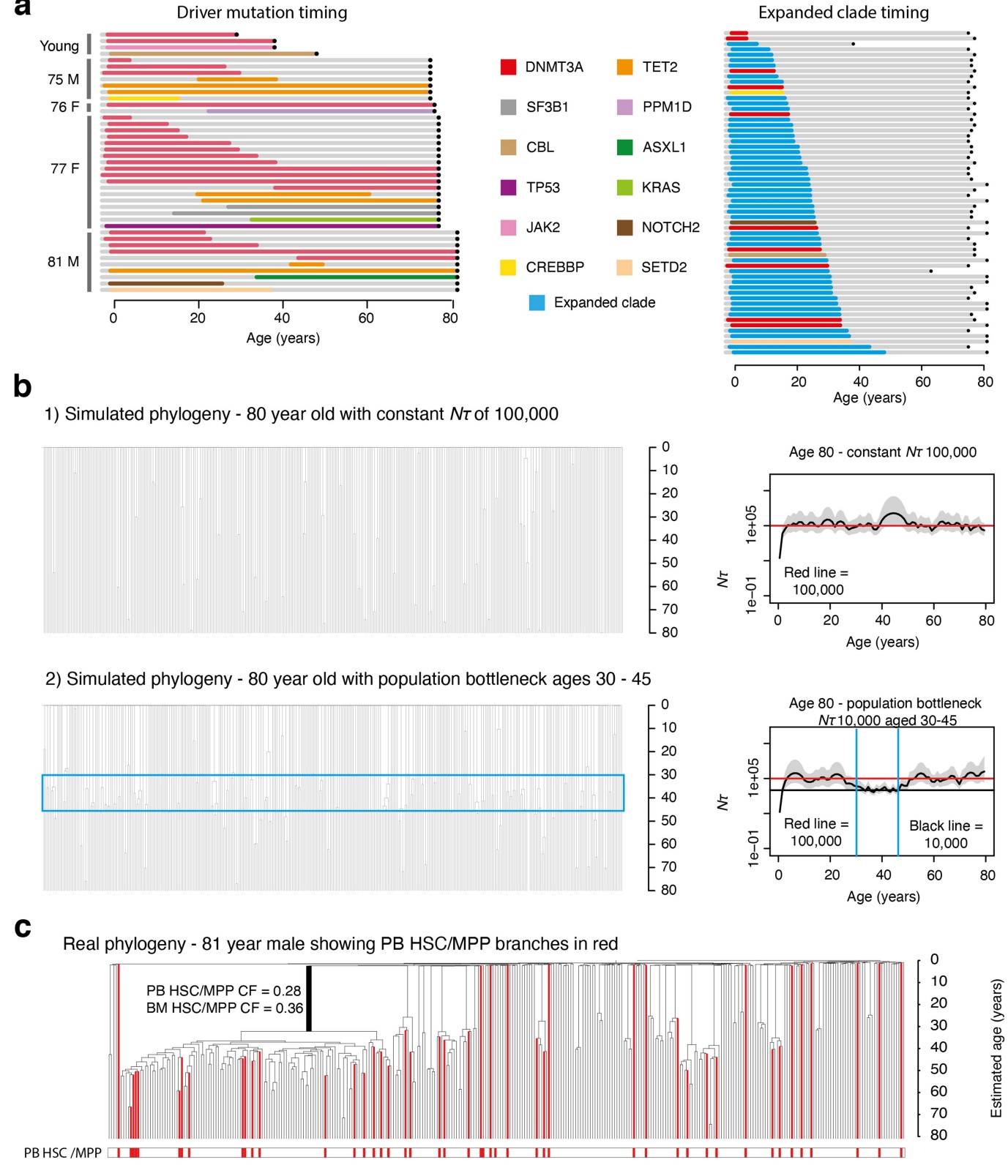

**Extended Data Fig. 7** | See next page for caption.

## a

**Bayesian inference**

### Model: Neutral model allowing varying HSC population size over life
- time between symmetric self renewing cell divisions fixed at a point estimate of 2 per year ($N\tau = N/2$)

**Prior**

Independent prior densities on three parameters:
1. N in 1st 2-3 decades of life (all donors)
   - uniform prior on interval [a,b],
   a = 1000; b = 250,000, 500,000, 750,000
2. Midlife fold-change in N (elderly donors only)
   - uniform prior on interval [ 0.01,1.2]
3. Latelife fold-change in N (all donors)
   - uniform prior on interval [0.5 to 8.0]

**Summary statistics for ABC**

1. Time-weighted mean number of lineages
(calculated at 3 time-points: 0.25, 0.5, 0.75)

### Approximate Bayesian Computation using ABC rejection method
Donor-specific ABC performed separately on data from each donor
Separate Monte Carlo samples generated from the (approximate) donor-specific posterior distribution (one for each donor).
For each donor-specific ABC:
- 100,000 parameter vectors sampled from prior
- From each parameter vector a simulated data set (phylogeny) is generated using (neutral) rsimpop
- From each simulated data set (phylogeny) a vector of summary statistics is computed
- Simulations ranked w.r.t. Euclidean distance between simulated and observed vectors of summary statistics
- The 1000 simulations with shortest Euclidean distances (to the observed data) are accepted (proportion = 1000/100,000 = 0.01)
The parameter vectors for the accepted simulations represent a sample from an approximate posterior distribution.

Simulated phylogeny 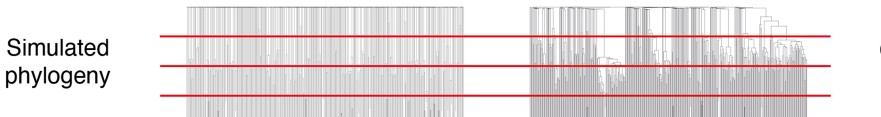 Observed phylogeny

### Posterior predictive checks (PPC)
Donor-specific (posterior predictive) p-value computed. For each donor we do the following:
- We have (from the ABC output) a sample of 1000 parameter vectors from the (approximate) donor-specific posterior distribution.
- For each parameter vector 1000 new (predicted) data sets (phylogenies) are simulated using (neutral) rsimpop;
- From each simulated data set (phylogeny) a vector of summary statistics is computed;
- From each (simulated) vector of summary statistics a (simulated) chi-squared discrepancy value is computed;
The proportion of simulations where the simulated chi-squared discrepancy value exceeds the observed
chi-squared discrepancy value is an estimate of the (posterior predictive) p-value.

**Summary statistics for PPC**

(all calculated at 4 time-points: 0.2, 0.4, 0.6, 0.8)
1. Time-weighted mean number of lineages
2. Size of top 3 largest clades
3. Number of singleton samples (clade size = 1)

**Interpretation of posterior predictive p-values**

- If the (posterior predictive) p-value is close to zero, this is evidence
against the proposed model as an explanation for the features of the
data captured by the summary statistics.
- If the p-value is close to zero, then the observed data is an outlier
compared to the data predicted under the proposed model.

## b

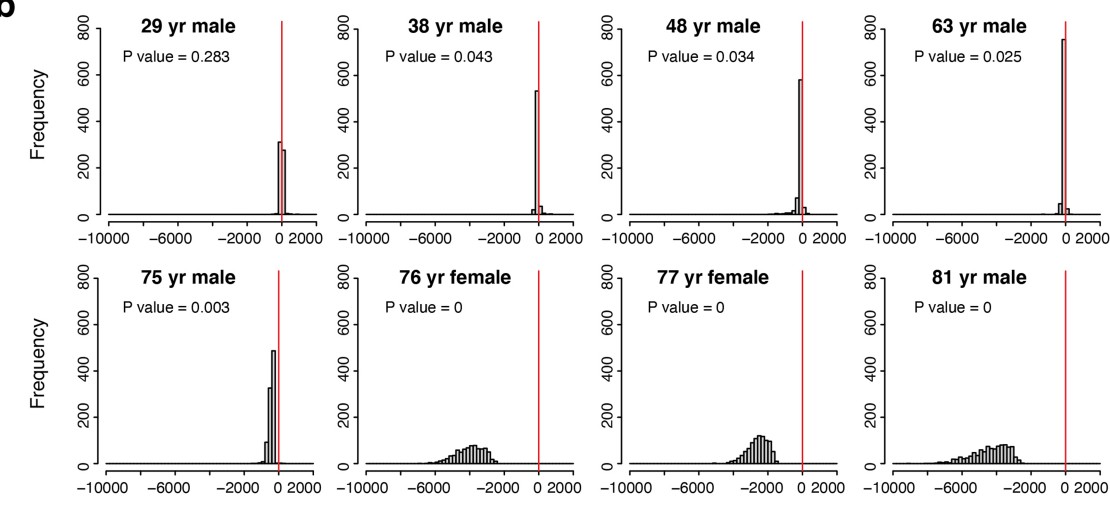

Extended Data Fig. 8 | See next page for caption.

**Extended Data Fig. 8 | Modelling HSC populations incorporating only changes in $N\tau$, without positive selection. a**, Overview of modelling approach used to estimate $N\tau$ alone in the young adult individuals and to investigate whether changes in $N\tau$ could explain the observed clade size distribution in the elderly adult individuals. These simulations were run using a neutral model (that is, no acquisition of driver mutations), with $N\tau$ being the only parameter to change over time. For the young adult individuals $N\tau$ was estimated for two time-blocks (time before and after population increase due to ST-HSC/MPP contribution). For the elderly adult individuals $N\tau$ was estimated for three time-blocks as the *phylodyn* plots predicted a population 'bottleneck' (Supplementary Fig. 12) was the most parsimonious way to recreate the observed change in coalescence density over life. **b**, Plots showing the posterior predictive distribution of the difference between the simulated chi-squared discrepancy and the observed chi-squared discrepancy, for each donor individual under a neutral model incorporating change in population size. For each donor, the posterior predictive distribution of the difference between predictive (simulated) and observed chi-squared discrepancy is represented as a histogram based on a Monte Carlo sample of 1,000,000 simulated phylogenies, drawn from the posterior predictive distribution. The proportion of simulated phylogenies which lie to the right of zero (red line) is a Monte Carlo estimate of the posterior predictive p-value (the probability that the predictive chi-squared discrepancy exceeds the observed chi-squared discrepancy under the neutral model). In the case of the four young adult individuals, the proportion of simulated phylogenies which lie to the right of 0 (red line) is close to 0.5, indicating that the simple neutral models (incorporating changes in $N\tau$ over life) predict trees that have similar clade size distributions to our observed trees. In contrast, for the four elderly adults, the proportion of simulated phylogenies which lie to the right of 0 is very small (Less than 0.05), demonstrating that the neutral models are, on their own, unable to recreate trees with similar clade size distributions to those observed.

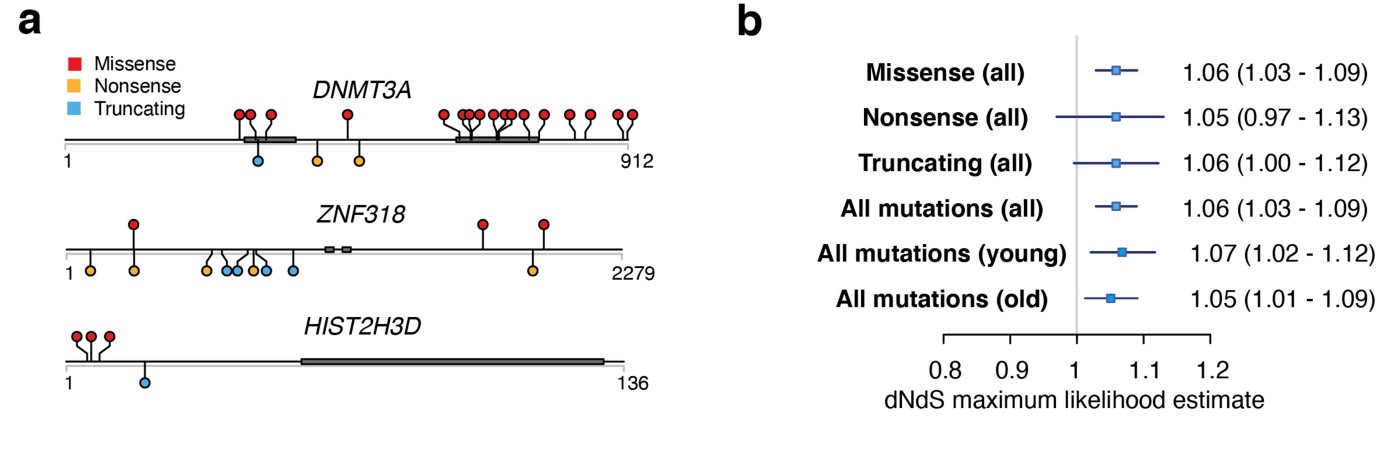

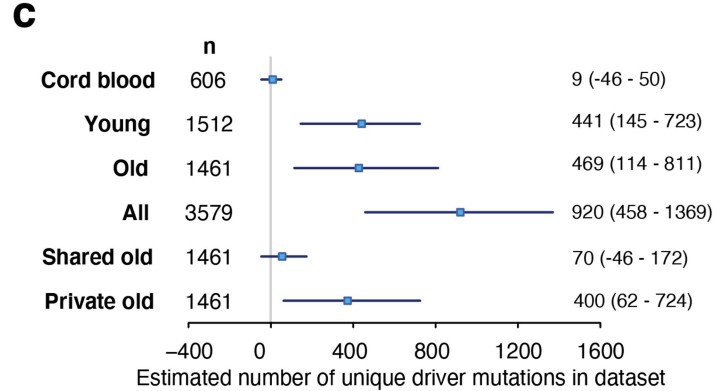

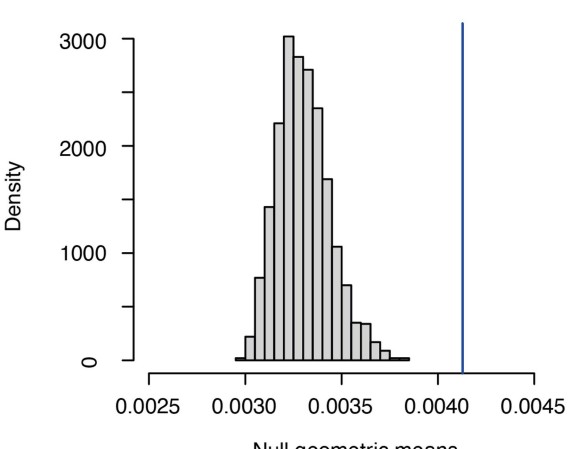

**Extended Data Fig. 9 | Positive selection in blood. a**, Lolliplot plots to show the sites of variants in the dataset in the three genes under significant positive selection according to dN/dS. Thick grey bars denote locations of conserved protein domains. **b**, dN/dS maximum likelihood estimates for missense, nonsense, truncating and all mutations in the complete dataset (n = 25,888 coding mutations) and for all mutations in the young (individuals aged < 65 year) and old (individuals aged > 75 years) datasets analysed separately. The boxes show the estimate with whiskers showing the 95% CI. The numbers to the left give the numeric values for the estimates with 95% CI in brackets. **c**, Estimated number of driver mutations in the different datasets. The boxes

show the estimate with whiskers showing the 95% CI. The numbers to the left give the numeric values for the estimates with 95% CI in brackets. 'n' is the number of cells included in each dataset. **d**, Results of a randomisation/Monte Carlo test to define the null expected distribution of clade size for cells with loss of Y. This null distribution of geometric means from 2000 simulations is shown (histogram) together with the observed geometric mean of clades with Y loss (vertical blue line). The observed value significantly outlies the expected distribution showing that clades with Y loss are significantly larger than would be expected by chance.

**a**
<center>Bayesian inference</center>

**Model incorporating positive selection in the form of 'driver mutations'**
- selection coefficient of 'driver mutation' drawn from gamma distribution;
- lower fitness threshold of selection coefficient set at 0.05 (equivalent to a fitness effect of 5% additional growth per year);
- Nτ fixed at a point estimate of 100,000 (population of N = 100,000 HSCs, dividing symmetrically once per year)

**Prior**

Independent prior densities on three parameters:
1. Shape of gamma distribution of selection coefficients
   - uniform prior on interval [0.1,4.0]
2. Rate of gamma distribution of selection coefficients
   - uniform prior on interval [5,120]
3. Number of drivers entering the population per year
   - uniform prior on interval [1,1000]

**Summary statistics for ABC**

(all calculated at 3 time-points: 0.2, 0.4, 0.6 - apart from stastic 4.)
1. Time-weighted mean number of lineages
2. Size of top 3 largest clades
3. Number of singleton samples (clade size = 1)
4. Number of coalescent events
(calculated at 2 time-intervals: 0.2-0.4, 0.4-0.6)

**Approximate Bayesian Computation using ABC regression method**

Multiple-donor ABC performed on the combined data from the 4 oldest donors.
Single Monte Carlo sample from the (approximate) multiple-donor posterior distribution generated using 4 ABC regression steps.
For each ABC regression step:
- 100,000 parameter vectors sampled from 'prior'
- From each parameter vector a simulated data set (phylogeny) is generated using rsimpop (incorporating driver mutations)
- From each simulated data set (phylogeny) a vector of summary statistics is computed
- Simulations are ranked w.r.t. Euclidean distance between simulated and observed vectors of summary statistics
- The 2000 simulations with the shortest Euclidean distances (to observed data) are accepted (proportion = 2000/100,000 = 0.02)
- For each parameter, the 2000 values (from the accepted parameter vectors) are adjusted by performing
a separate ridge regression on the (re-scaled and logit-transformed) parameter values
The resulting 2000 parameter vectors of adjusted parameter values represent a sample from an approximate posterior distribution.
In the first ABC regression step, 100,000 parameter vectors are sampled from the original prior distribution.
At each subsequent ABC regression step, 100,000 parameter vectors are sampled from the posterior sample generated
by the preceding ABC regression step.
The parameter vectors for the accepted simulations represent a sample from an approximate posterior distribution.
The 2000 parameter vectors of adjusted parameter values obtained from the final ABC regression step represent
a sample from the (approximate) multiple-donor posterior distribution.

Simulated phylogeny 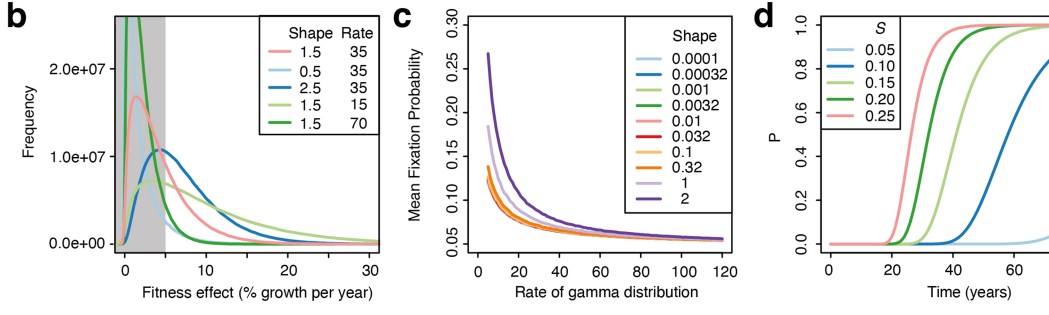 Observed phylogeny

**Posterior predictive checks (PPC)**

Donor-specific (posterior predictive) p-value computed. For each donor we do the following:
- Sample 1000 parameter vectors from the sample (of 2000) from the multiple-donor posterior distribution
- For each parameter vector 1000 new data sets (phylogenies) are simulated using rsimpop (incorporating driver mutations)
resulting in 1,000,000 simulations
- From each (of 1,000,000) simulated data set (phylogeny) a vector of summary statistics is computed
- From each (simulated) vector of summary statistics a (simulated) chi-squared discrepancy value is computed
The proportion of simulations where the simulated chi-squared discrepancy value exceeds the observed chi-squared
discrepancy value is an estimate of the (posterior predictive) p-value.

**Summary statistics for PPC**

For 1st PPC: calculated at 2 time-points (0.2,0.4)
For 2nd PPC: calculated at 3 time-points (0.2, 0.4, 0.6)
1. Time-weighted mean number of lineages
2. Size of top 5 largest clades
3. Number of singleton samples (clade size = 1)
4. Number of coalescent events

**Interpretation of posterior predictive p-values**

- If the (posterior predictive) p-value is close to zero, this is evidence against the proposed model as an explanation for the features of the data captured by the summary statistics.
- If the p-value is close to zero, then the observed data is an outlier compared to the data predicted under the proposed model.

**b**

Shape / Rate
- 1.5 / 35
- 0.5 / 35
- 2.5 / 35
- 1.5 / 15
- 1.5 / 70

Frequency (y-axis): 0.0e+00, 1.0e+07, 2.0e+07
Fitness effect (% growth per year) (x-axis): 0, 10, 20, 30

**c**

Shape
- 0.0001
- 0.00032
- 0.001
- 0.0032
- 0.01
- 0.032
- 0.1
- 0.32
- 1
- 2

Mean Fixation Probability (y-axis): 0.05, 0.10, 0.15, 0.20, 0.25, 0.30
Rate of gamma distribution (x-axis): 0, 20, 40, 60, 80, 100, 120

**d**

S
- 0.05
- 0.10
- 0.15
- 0.20
- 0.25

P (y-axis): 0.0, 0.2, 0.4, 0.6, 0.8, 1.0
Time (years) (x-axis): 0, 20, 40, 60, 80

**Extended Data Fig. 10 |** See next page for caption.

**Extended Data Fig. 10 | Modelling of HSC populations incorporating positive selection. a**, Overview of modelling approach used to estimate the shape and rate of the gamma distribution of selection coefficients from which 'driver mutations' are drawn, and the number of driver mutations drawn from this distribution (using a selection coefficient threshold of > 0.05) that are entering the HSC population per year. For these simulations $N\tau$ was fixed at 100,000 and therefore only summary statistics for the first 3 timepoints were used to assess how well a given simulation for an individual resembled the observed tree. **b**, Plot showing maximum posterior density estimates of the rate and shape parameters of the gamma distribution for selection coefficients (pink line) obtained using Approximate Bayesian computation. Blue/green lines show how altering the rate and shape parameters affect the gamma distribution. **c**, Plot showing how changing the shape of the gamma distribution of selection coefficients (each line has a different shape) alters the probability of a driver gene fixing in the population. Reducing the shape below 0.1 does not affect the probability of driver gene fixation and therefore was the lower limit of the shape prior. **d**, Plot showing how the probability of detecting a clone with CF 2.5% changes over time for different selection coefficients. There is only a probability of 0.1 of being able to identify a driver mutation with a selection coefficient of 0.05 that entered the population at birth. We therefore used a lower threshold of 0.05 for the driver mutation selection coefficients.

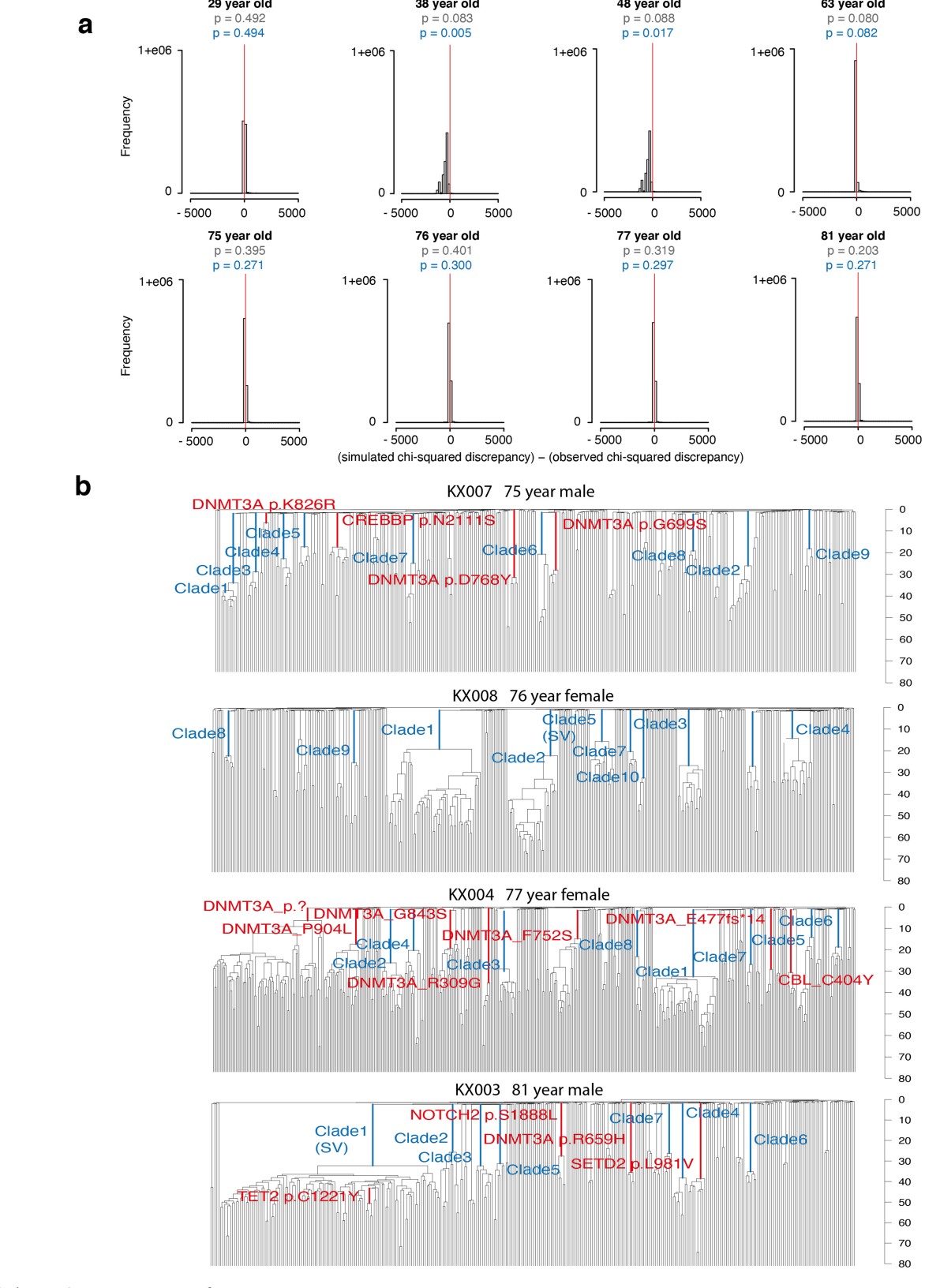

**Extended Data Fig. 11** | See next page for caption.

**Extended Data Fig. 11 | Driver modelling results and expanded clade annotation. a**, Plots showing the posterior predictive distribution of the difference between the predictive (simulated) chi-squared discrepancy and the observed chi-squared discrepancy, for each donor individual under the simple positive selection model. For the definition of the chi-squared discrepancy, and details of how the posterior predictive p-values are estimated, see Supplementary Information "Posterior predictive model checking (PPC) methods which can be applied to Approximate Bayesian Computations (ABC)", Sections 1, 2 and 5. In these plots, the chi-squared discrepancy is computed from summary statistics evaluated at the first 3 (out of 4 equally spaced) timepoints on the phylogeny obtained from the specified donor (Extended Data Fig. 8). For each donor, the posterior predictive distribution of the difference between predictive (simulated) and observed chi-squared discrepancy is represented as a histogram based on a Monte Carlo sample of at least 100,000 simulated phylogenies, drawn from the posterior predictive distribution. The proportion of simulated phylogenies which lie to the right of zero (red line) is a Monte Carlo estimate of the posterior predictive p-value (the probability that the predictive chi-squared discrepancy exceeds the observed chi-squared discrepancy under the positive selection model). Those p-values written in grey text are based on chi-squared discrepancies computed from summary statistics evaluated at the first 2 (out of 4 equally spaced) timepoints. Notice that these p-values are all above the 0.05 threshold, indicating that observed phylogenies (up to the second time point) are compatible with the simple positive selection model. Those p-values written in blue text are based on chi-squared discrepancies computed from summary statistics evaluated at the first 3 (out of 4 equally spaced) timepoints. Notice that all but two observed phylogenies (up to the third time point) are compatible with this positive selection model. These p-values indicate that, once the third time point is included, the phylogenies of two of the younger individuals (38 year-old and 48 year-old) are no longer compatible with the positive selection model. Notice that these two donors also exhibit the most striking increase in population size from the middle part of the population trajectory onwards (Fig. 4a). When all four timepoints are included, the phylogenies of 5 out of 8 donors have become incompatible with the positive selection model (data not shown). Only the phylogenies from the donors of ages 77, 76 and 29, remain compatible with the positive selection model. This suggests that the current positive selection model does not adequately account for the population processes towards the time of sampling. **b**, Phylogenies of the four adults aged > 70 labelled with driver mutations and clade ID annotations as used in Fig. 5d.

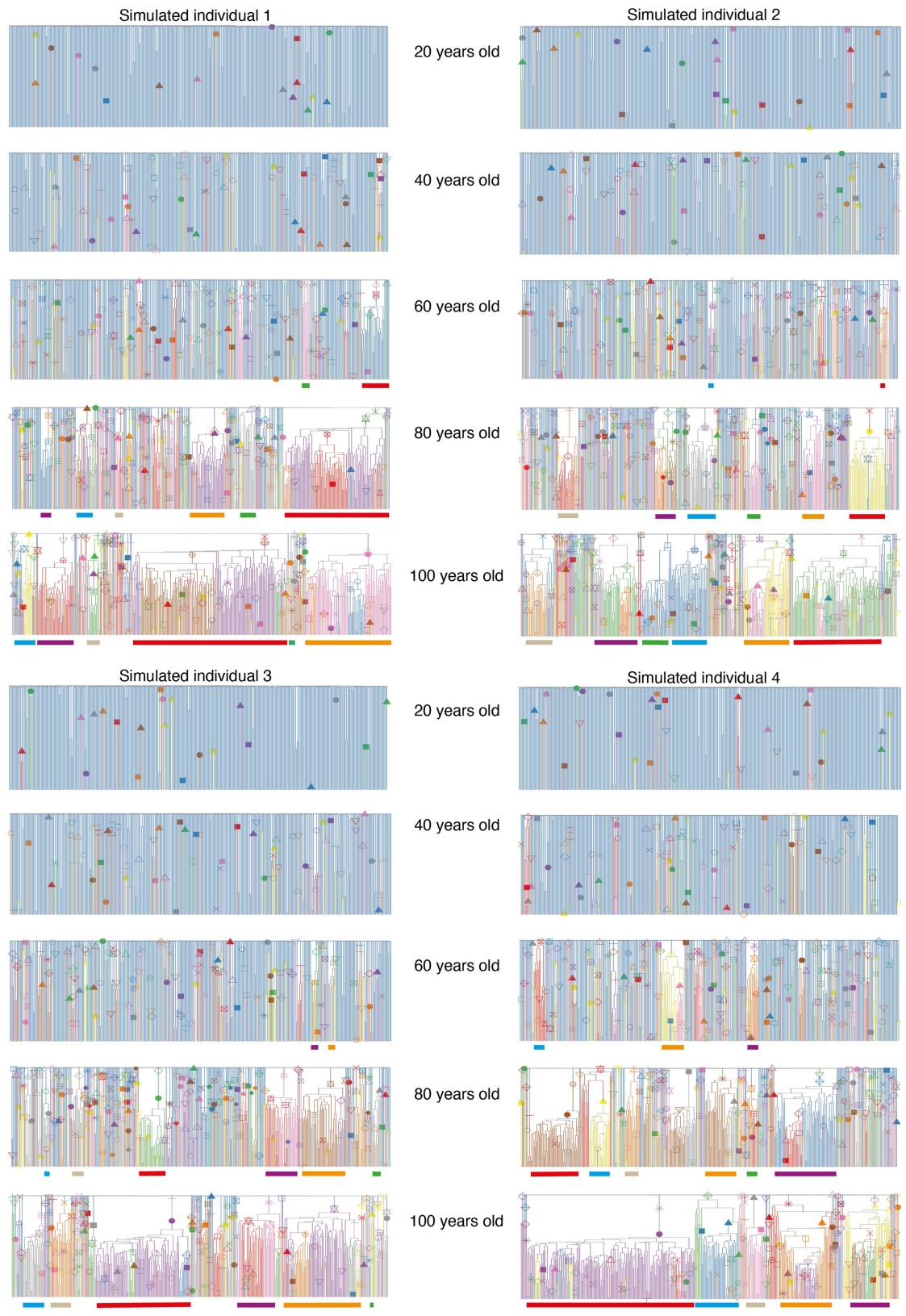

**Extended Data Fig. 12 |** See next page for caption.

**Extended Data Fig. 12 | Positive selection over life.** Four consecutively simulated phylogenies of 380 cells sampled from a population of 100,000 cells that has been maintained at a constant $N\tau$ over life, with incorporation of positively selected 'driver mutations'. The driver mutations have a fitness effect > 5% (drawn from a gamma distribution with shape = 0.47 and rate = 34) and enter the population at a rate of 200 per year. These are the maximum posterior density estimates of the rate and shape parameters obtained using the ABC method. The inclusion of these driver mutations is able to recapitulate a similar clade size distribution to that observed in the real HSPC phylogenies of the observed individuals across the whole age range. However, including driver mutations does not fully recapitulate the observed lack of coalescent events in the last 10–15 years of life, showing that an increase in $N\tau$ over this time is also required to fully recreate the patterns of coalescences in the real phylogenies. Driver mutations are marked with a symbol and their descendent clades are coloured. In all cases $N\tau$ is the same as the population size ($N$) as the generation time ($\tau$) in all simulations is fixed at 1 year. The symbols / colours are not consistent for driver mutations between plots. The largest clades are therefore coloured in a consistent way beneath the plots to show how their size changes over time. The simulated phylogenies illustrate the complex clonal dynamics that can occur in later life as a result of clonal competition. While the majority of clades continue to expand, others stay relatively stable and some reduce in size. The phylogenies also show that by the age of 80 typically > 90% of HSCs in the population carry at least one driver mutation.

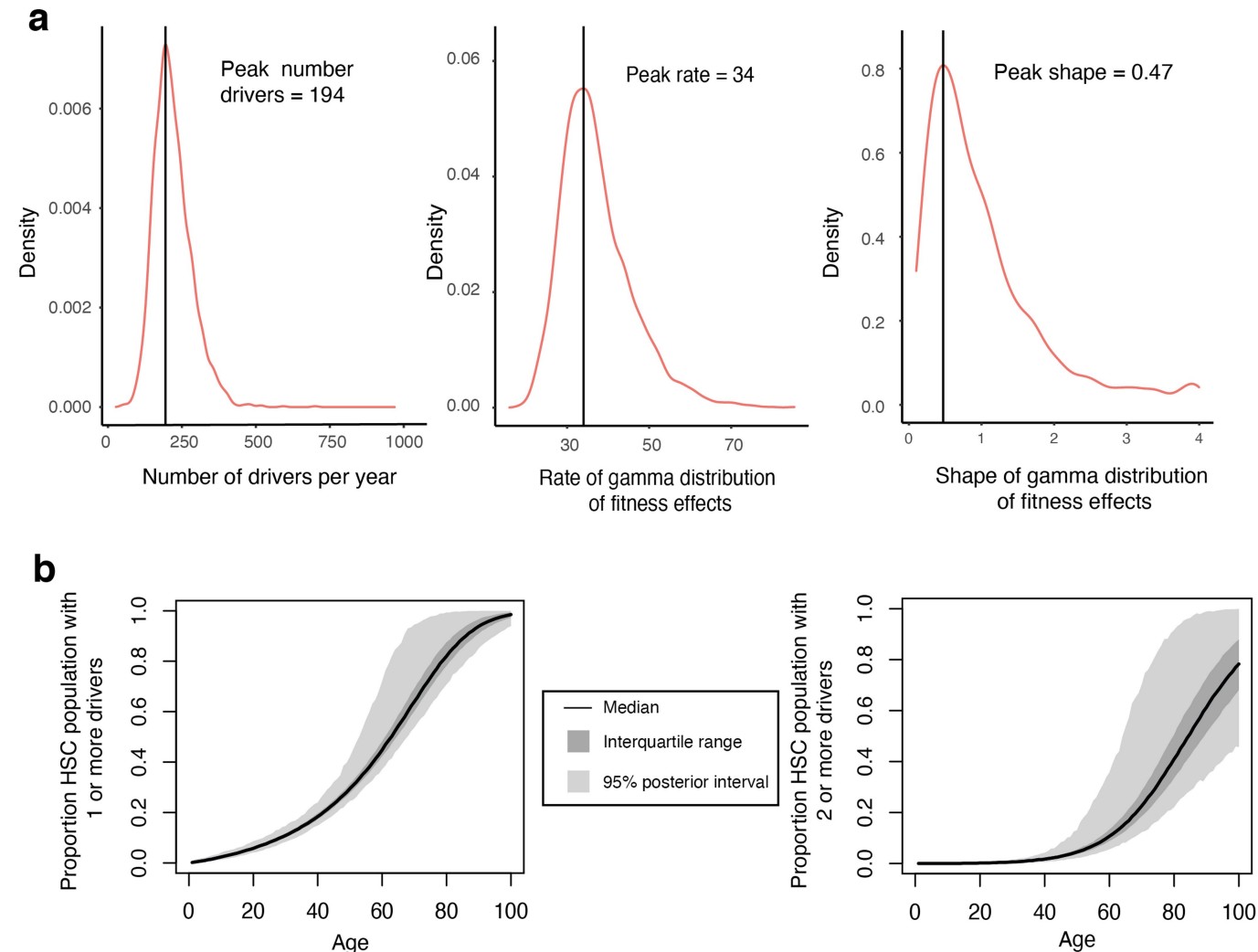

**Extended Data Fig. 13 | Driver modelling parameter and driver acquisition estimates. a**, Posterior distributions for the three driver modelling parameters: 1) Number of 'driver' mutations with a fitness effect > 5% entering HSC population of 100,000 cells per year, 2) Rate of gamma distribution of fitness effects, 3) Shape of gamma distribution of fitness effects. Black lines show peak estimates. **b**, Plot showing the median, interquartile range, and 1st and 99th percentiles for proportion of HSC population with drivers calculated yearly for 10,000 HSC population simulations run utilising the optimal parameter values for driver acquisition rate and fitness effects derived from the ABC modelling approach. The point estimate for the shape and rate parameters of the gamma distribution were shape = 0.47, rate = 34. The point estimate for the number of drivers with $s$>5% entering population per year = 200.

# Reporting Summary

## Statistics

For all statistical analyses, confirm that the following items are present in the figure legend, table legend, main text, or Methods section.

| n/a | Confirmed | |
|---|---|---|
| ☐ | ☒ | The exact sample size (*n*) for each experimental group/condition, given as a discrete number and unit of measurement |
| ☐ | ☒ | A statement on whether measurements were taken from distinct samples or whether the same sample was measured repeatedly |
| ☐ | ☒ | The statistical test(s) used AND whether they are one- or two-sided *Only common tests should be described solely by name; describe more complex techniques in the Methods section.* |
| ☐ | ☒ | A description of all covariates tested |
| ☐ | ☒ | A description of any assumptions or corrections, such as tests of normality and adjustment for multiple comparisons |
| ☐ | ☒ | A full description of the statistical parameters including central tendency (e.g. means) or other basic estimates (e.g. regression coefficient) AND variation (e.g. standard deviation) or associated estimates of uncertainty (e.g. confidence intervals) |
| ☐ | ☒ | For null hypothesis testing, the test statistic (e.g. *F*, *t*, *r*) with confidence intervals, effect sizes, degrees of freedom and *P* value noted *Give P values as exact values whenever suitable.* |
| ☐ | ☒ | For Bayesian analysis, information on the choice of priors and Markov chain Monte Carlo settings |
| ☐ | ☒ | For hierarchical and complex designs, identification of the appropriate level for tests and full reporting of outcomes |
| ☒ | ☐ | Estimates of effect sizes (e.g. Cohen's *d*, Pearson's *r*), indicating how they were calculated |

*Our web collection on statistics for biologists contains articles on many of the points above.*

## Software and code

Policy information about availability of computer code

| Data collection | None |
|---|---|
| Data analysis | List of programs and softwares:<br>• R: version 3.6.1<br>• BWA-MEM: version 0.7.17 (https://sourceforge.net/projects/bio-bwa/)<br>• cgpCaVEMan: version 1.11.2/1.13.14/1.14.1 (https://github.com/cancerit/CaVEMan)<br>• cgpPindel: version 2.2.5/3.2.0/3.3.0 (https://github.com/cancerit/cgpPindel)<br>• Brass: version 6.1.2/6.2.0/6.3.0/6.3.4 (https://github.com/cancerit/BRASS)<br>• ASCAT NGS: version 4.2.1/4.3.3 (https://github.com/cancerit/ascatNgs)<br>• VAGrENT: version 3.5.2/3.6.0/3.6.1 (https://github.com/cancerit/VAGrENT)<br>• GRIDSS: version 2.9.4 (https://github.com/PapenfussLab/gridss)<br>• MPBoot: version 1.1.0 (https://github.com/diepthihoang/mpboot)<br>• cgpVAF: version 2.4.0 (https://github.com/cancerit/vafCorrect)<br>• dNdScv: version 0.0.1 (https://github.com/im3sanger/dndscv)<br>• Telomerecat: version 3.4.0 (https://github.com/jhrf/telomerecat)<br>• Rsimpop: version 2.0.4 (https://github.com/NickWilliamsSanger/rsimpop)<br>• Phylodyn: version 0.9.02 (https://github.com/mdkarcher/phylodyn)<br>• FlowJo: version 10<br>Custom code made available (also stated in manuscript): https://github.com/emily-mitchell/normal_haematopoiesis<br>No commercial software used. |

For manuscripts utilizing custom algorithms or software that are central to the research but not yet described in published literature, software must be made available to editors and reviewers. We strongly encourage code deposition in a community repository (e.g. GitHub). See the Nature Portfolio guidelines for submitting code & software for further information.

## Data

Policy information about availability of data

All manuscripts must include a data availability statement. This statement should provide the following information, where applicable:

- Accession codes, unique identifiers, or web links for publicly available datasets
- A description of any restrictions on data availability
- For clinical datasets or third party data, please ensure that the statement adheres to our policy

Sequence data that support the findings of this study have been deposited in the European Genome-Phenome Archive (https://www.ebi.ac.uk/ega/home; accession number EGAD00001007851).
The main data needed to reanalyse / reproduce the results presented is available on Mendeley Data (https://data.mendeley.com/datasets/np54zjkvxr/1).
All scripts and some smaller data matrices are available on github (https://github.com/emily-mitchell/normal _haematopoiesis).
hg37 human reference genome has been used in this study.

# Field-specific reporting

Please select the one below that is the best fit for your research. If you are not sure, read the appropriate sections before making your selection.

☒ Life sciences          ☐ Behavioural & social sciences          ☐ Ecological, evolutionary & environmental sciences

For a reference copy of the document with all sections, see nature.com/documents/nr-reporting-summary-flat.pdf

# Life sciences study design

All studies must disclose on these points even when the disclosure is negative.

| | |
|---|---|
| Sample size | We optimised the number of individuals (10) and number of haematopoietic stem cells sequenced per individual (average of 358 cells per individual) to describe the mutation burden, telomere lengths and clonal structure of normal haematopoietic stem cell populations across a range of ages. No power calculation was performed, and there was no target effect size. |
| Data exclusions | Per pre-established criteria, genomes with a sequencing depth of less than 7x (17 samples) or with a VAF distribution showing evidence of non-clonality or contamination (peak VAF < 40%) (7 samples) were excluded from the analysis. |
| Replication | While the specific donor samples used have been exhausted, the results from this study should be generally reproducible in separate healthy individuals of the same age, using the protocols and code included in this manuscript. |
| Randomization | This is not relevant to our study. All individuals were haematopoietically normal, and there was no test versus control groups. |
| Blinding | Blinding was not relevant to our study. There was no test performed that required blinding. |

# Reporting for specific materials, systems and methods

We require information from authors about some types of materials, experimental systems and methods used in many studies. Here, indicate whether each material, system or method listed is relevant to your study. If you are not sure if a list item applies to your research, read the appropriate section before selecting a response.

### Materials & experimental systems

| n/a | Involved in the study |
|---|---|
| ☐ | ☒ Antibodies |
| ☒ | ☐ Eukaryotic cell lines |
| ☒ | ☐ Palaeontology and archaeology |
| ☒ | ☐ Animals and other organisms |
| ☐ | ☒ Human research participants |
| ☒ | ☐ Clinical data |
| ☒ | ☐ Dual use research of concern |

### Methods

| n/a | Involved in the study |
|---|---|
| ☒ | ☐ ChIP-seq |
| ☐ | ☒ Flow cytometry |
| ☒ | ☐ MRI-based neuroimaging |

## Antibodies

| | |
|---|---|
| Antibodies used | Marker ; Fluorochrome ; Manufacter ; Catalogue Number ; Clone ; Dilution<br>CD3 ; FITC ; BD ; 555339 ; HIT3a ; 1 in 500<br>CD90 ; PE ; Biolegend ; 328110 ; 5E10 ; 1 in 50<br>CD49f ; PECy5 ; BD ; 551129 ; GoH3 ; 1 in 100<br>CD19 ; A700 ; Biolegend ; 302226 ; HIB19 ; 1 in 300 |

CD34 ; APCCy7 ; Biolegend ; 343514 ; 581 ; 1 in 100
Zombie  ; Aqua ; Biolegend ; 423101 ; NA ; 1 in 2000
CD38 ; PECy7 ; Biolegend ; 303516 ; HIT2 ; 1 in 100
CD45RA ; BV421 ; Biolegend ; 304130 ; HI100 ; 1 in 100
CD33; APC; BD; 571817; WM53; 1 in 200

Validation

These were all previously validated commercially available antibodies.
CD3 FITC: Validated by supplier with the following notes - species reactivity: human; application - flow cytometry
CD90 PE: Validated by supplier with the following notes - species reactivity: human, rhesus, baboon, macaque, pig; application - flow cytometry
CD49f PECy5: Validated by the supplier with the following notes - species reactivity: human, mouse, pig; application: flow cytometry
CD19 A700: Validated by the supplier with the following notes - species reactivity: human, chimpanzee, rhesus; application: flow cytometry
CD34 APCCy7: Validated by the supplier with the following notes - species reactivity: human, cynomolgus; application: flow cytometry
Zombie aqua: Validated by the supplier with the following notes - species reactivity: NA; application: flow cytometry and immunofluorescence microscopy
CD38 PECy7: Validated by the supplier with the following notes - species reactivity: human, chimpanzee, horse, cow; application: flow cytometry
CD45RA BV421: Validated by the supplier with the following notes - species reactivity: human, chimpanzee; application: flow cytometry and  immunohistochemistry
CD33 APC: Validated by the supplier with the following notes - species reactivity: human, chimpanzee; application: flow cytometry

# Human research participants

Policy information about studies involving human research participants

Population characteristics

The dataset comprised 3759 whole genomes from the bone marrow, blood or cord blood of 10 individuals aged 0 years to 81 years. All individuals were haematologically normal. One individual (38 year male) had inflammatory bowel disease (Crohn's disease) and one individual (48 year male) had selenoprotein deficiency, a rare genetic disorder not known to impact haematopoietic stem cell population dynamics. In total 4 females and 6 males were included in the study.

Recruitment

Cord blood mononuclear cell samples were obtained from Stem Cell Technologies. These had been collected with informed consent including for whole genome sequencing (catalog #70007). Cambridge Blood and Stem Cell Biobank (CBSB) provided fresh peripheral blood samples taken with informed consent from two patients at Addenbrooke's Hospital (NHS Cambridgeshire 4 Research Ethics Committee reference 07/MRE05/44 for samples collected pre-November 2019 and Cambridge East Ethics Committee reference 18/EE/0199 for samples collected from November 2019 onwards. Cambridge Biorepository for Translational Medicine (CBTM) provided frozen bone marrow +/- peripheral blood MNCs taken with informed consent from six deceased organ donors. Samples were collected at the time of abdominal organ harvest (Cambridgeshire 4 Research Ethics Committee reference 15/EE/0152).

Ethics oversight

Cambridgeshire 4 Research Ethics Committee reference 07/MRE05/44; Cambridge East Ethics Committee reference 18/EE/0199; Cambridgeshire 4 Research Ethics Committee reference 15/EE/0152.

Note that full information on the approval of the study protocol must also be provided in the manuscript.

# Flow Cytometry

## Plots

Confirm that:

☒ The axis labels state the marker and fluorochrome used (e.g. CD4-FITC).

☒ The axis scales are clearly visible. Include numbers along axes only for bottom left plot of group (a 'group' is an analysis of identical markers).

☒ All plots are contour plots with outliers or pseudocolor plots.

☒ A numerical value for number of cells or percentage (with statistics) is provided.

## Methodology

Sample preparation

Mononuclear cells (MNCs) were isolated using lymphoprepTM density gradient centrifugation (STEMCELL Technologies), after diluting whole blood 1:1 with PBS. The red blood cell and granulocyte fraction of the blood was then removed. The MNC fraction underwent red cell lysis using 1 incubation at 4C for 15 mins with RBC lysis buffer (BioLegend). CD34 positive cell selection of peripheral blood and cord blood MNC samples was undertaken using the EasySep human whole blood CD34 positive selection kit (STEMCELL Technologies). The kit was used as per the manufacturer's instructions, but with only a single round of magnetic selection. Bone marrow MNCs did not undergo CD34 positive selection prior to cell sorting.

Instrument

Stained MNC or CD34+ enriched cell fractions were single cell index sorted on either a BD Aria III or BD Aria fusion cell sorter.

Software

No analysis of flow cytometry data is presented in this manuscript. FlowJo v10 was used to generate the gating strategy image.

Cell population abundance

Typical cell population abundances are shown in the sorting strategy ED Fig. 1a.  In BM samples and PB CD34+ selected

Cell population abundance

samples, myeloid cells were 20-50% of live cells and CD34+ cells were 1-5% of myeloid cells. In CB CD34+ selected samples, myeloid cells were 10-30% of live cells and CD34+ cells were 5-15% of myeloid cells. Approximately 10 x 96 well plates of HSC/MPP cells could be sorted from 5+E07 BM MNCs and 2-3 x 96 well plates of HSC/MPP cells could be sorted from 10+E07 PB MNCs.

Gating strategy

The gating strategy used for cell sorting is illustrated in ED Fig. 1a.
To summarise:
1. FSC-A vs SSC-A showing all events: gate on cell population (to exclude dead cells and debris)
2. FSC-A vs FSC-W showing cell population: gate on singlets (to exclude doublets)
3. ZOMBIE aqua vs SSC-A showing singlets: gate on live cells (to exclude dead cells)
4. CD19 A700 vs CD3 FITC showing live cells: gate on CD19-CD3- myeloid cells (to exclude T and B cells)
5. CD34 APCCy7 vs CD38 PECy7 showing myeloid cells: gate on CD34+
6. CD34 APCCy7 vs CD38 PECy7 showing CD34+ cells: gate on CD34+CD38- (approx 20% of population)
7. CD45RA BV421 vs CD90 PE showing CD34+CD38- cells: gate on CD45RA neg cells = sorted "HSC/MPP" population

☒ Tick this box to confirm that a figure exemplifying the gating strategy is provided in the Supplementary Information.

