## [Peer Review File · Nature]

Manuscript Title: Clonal dynamics of haematopoiesis across the human lifespan

Reviewer Comments & Author Rebuttals

Reviewer Reports on the Initial Version:

Referees' comments:

Referee #1 (Remarks to the Author):

Mitchell et al performed whole genome sequencing of 3579 single-cell-derived colonies from 10 healthy individuals to study the clonal dynamics haematopoiesis. They carefully analyzed these data to confirm and refine what has been known about clonal haematopoiesis such as the linear accumulation of point mutations and indels over time and the shortening of telomeres with increasing age. Additionally, the authors inferred phylogenetic trees and analyzed the subclonal composition. They found that in adults aged <65 haematopoiesis is highly polyclonal while in adults aged >70 expanded subclones made up 32% to 61% of the sequenced colonies. Moreover, they could estimate the timing of branching points in the phylogenies, the occurrence of mutations, and multiple independent losses of chromosome Y. To investigate potential explanations for the different subclonal compositions, the authors developed a simple mathematical model

Overall, I have to admit that I struggled with the text. Mostly because of the long introduction and the repetition of known results. It's not really clear which of these results are new and novel. That actually starts in the abstract which does say "We sequenced..", but then continues with rather generic statements and it is unclear which of these statements are new or novel findings of this particular study. Most findings seem rather incremental refinements. I also think that the computational methods should be described in much greater detail. For example, it's not clear to me how the phylogenies in Figs. 2 and 3 were generated. Although there is the "Construction of phylogenetic trees" section in the supplement, it only references the phylogenies in the Extended Data Figures section.

Fig 1c,d: It would be great if there would be marks for means/median (perhaps even quantiles such as in Fig. 1h). Due to the overlapping clouds of points, it is hard to see where most of these points lie.

Figs. 2 and 3: These figures are very interesting. How were these phylogenies inferred? The figure captions could provide some more details. Additional summary statistics about number of driver mutations, expanded clones, fraction of expanded clones, average onset times of these clones/drivers would be helpful. The information in Fig. 5a and b could be integrated directly to better put this information in context.

Line 210: Doubled "in mice"

Line 302: It is not clear to me how the authors converted the dN/dS ratios into estimated numbers of

driver mutations. Can you please elaborate?

Line 386: Time of oligoclonal haematopoiesis was also already predicted by the models presented in refs. 44 and 61. Similarly line 390, isn't the universal loss of clonal diversity in haematopoiesis an already known fact? It is also stated in that way in the first paragraph of the related manuscript by Fabre et al.

One final remark: I have no idea why the authors choose to have the figure captions not next to the figures and instead even put them in different files for the extended data figures. There is so much material to review and this makes it unnecessarily hard.

Referee #2 (Remarks to the Author):

In this manuscript, Mitchell et al present an elegant analysis of whole-genome sequences from single-cell derived colonies of HSC/MPPs across haematologically normal subjects across a wide age range. The conclusions are well-supported, are certain to be of interest to the readership of Nature, and have vast significance for several reasons:

- (i) Only a $\sim 1/5$ of clones had known driver mutations, which has important implications for the clonal hematopoiesis field, in which the majority of studies to date have primarily focused on CH with known drivers
- (ii) The authors find that the landscape of CH drivers may be much larger than previously appreciated (and as compared to the landscape of events that drive malignancy), that $1/34$ to $1/12$ non-synonymous mutations may be drivers
- (iii) Their findings refine estimates of the number of long-term HSCs in humans
- (iv) The authors report a striking difference between the polyclonal nature of hematopoiesis in younger individuals as compared to the oligoclonal landscape in older individuals, and compellingly, find that these age-related dynamics and abrupt emergence of oligoclonality appear to be explainable by simple, predictable processes

Specific points

- In EDF 1b, the data reveal that not all sorted cells were able to form colonies. Could the authors comment on whether this may represent a source of bias? Could the ability to form colonies track with some of the measurements in their paper such as telomere length, number of somatic mutations, which would in turn influence some of their quantitative inferences?
- Given that X-chromosome mosaic events are common, why do the authors primarily only focus on loss of Y chromosome and not X-chromosome events as well?
- Stem cell biologists have focused on the role of the bone marrow microenvironment with age. Could the authors speculate on the known changes to the microenvironment at older age and how that might influence the dramatic changes in clonal diversity with age?

- How do the authors think that the clones without known driver mutations contribute to the development of myeloid malignancies?

Referee #3 (Remarks to the Author):

This is an excellent research article in which authors report single-cell sequences of hematopoietic stem cells and multipotent progenitors (HSC/MPPs). Several interesting observations and claims are made based on molecular evolutionary analysis with dating. I am very supportive of its publication. But, I do have many major/minor comments and concerns that will help clarify important findings reported and claims made.

(a) Single-cell sequences reveal that hematopoiesis is massively polyclonal, and there is high clonal diversity. It will be useful to include a comparison with polyclonality and diversity of stem cells in other tissues or organs.

(b) Using parameters estimated from the data, empirical simulations show concordance with the observations under a simple model. While it is exciting, one wonders how the simulation consistently achieves an inflection point in the change in diversity in the last few years of the simulated data. Is it a deterministic outcome by design or by model properties? An intuitive explanation will be useful.

(c) The finding that older individuals have decreased clonal diversity is interesting, especially that it became pronounced only after the age of 70. It will be great to know if there is something special about this timing from a functional genomics point of view?

(d) An interesting observation is made about the association between the loss of the Y chromosome and the increase in the representation of the descending lineages. This is interpreted as a selective benefit. But, I wonder how the alternative explanation of the relaxation of negative selection is excluded. Intuitively, loss of the Y chromosome will reduce the number of proteins encoded, reducing the degree of purifying selection, especially if any of the protein was functionally used (e.g., dN/dS was smaller than others). Of course, if it is not possible to exclude the relaxation of purifying selection, alternatives need to be mentioned together.

(e) Pervasive positive selection is claimed based on dN/dS ratio being slightly larger than 1. First, the value is barely larger than 1, which could easily be due to the bias of the estimate. In somatic evolution, overall dN/dS close to 1 is universally observed. A slight deviation from this ratio can occur because there is no way to eliminate bias in estimation methods. Since the effect size is small and could be due to bias, the estimate's variance for a large dataset becomes so small that the P values become nonsensical. Therefore, the authors have an unenviable task of proving that their dN/dS estimates are unbiased if they used dN/dS ratios slightly greater than 1 to indicate positive selection. Of course, this does not exclude the possibility that some mutations will likely confer an advantage for some cells to grow larger in numbers, but it is not the same as pervasive positive selection. Anyway, in Figure 5d, 95% CIs are rather close to 1 (even without considering bias in generating the estimate and its CI), so one needs to be very careful in claiming pervasive positive selection.

(f) Was the neutrality of amino acid variants tested by using a prediction tool? Anyway, it will be good to know what fraction of non-driver AA variants were likely non-neutral.

(g) How were the phylogenetic trees linearized? Since the phylogenies are not strictly clock-like, it is

best to use a relaxed clock method that can convert a phylogeny into a tree (e.g., a Bayesian approach, if possible, or a rapid method such as RelTime for many sequences).

(h) Based on the phylogeny included in the supplementary information (for an older individual), the accumulation of mutations seems to be relatively constant. But, the mutation rate is likely different in the initial life stages (first few years) in which cell divisions seem to produce two stem cells as compared to the rest of the lifetime in which cell divisions appear to be asymmetric such that they produce one stem cell and another progenitor/specialized cell. Wouldn't this change the mutation rate between two phases, making it difficult to convert a phylogeny into a clonal timetree? Is it possible to estimate the early and late mutation rates and compare them to mutation rates in other somatic contexts?

(i) Also, the authors need to test whether the mutation rate is constant during all phases of the lifetime. This may be checked by examining the estimates of mutation rates (or diversity) from younger individuals compared to older individuals (but maybe not the oldest). This could reveal a mutational rate shift in different segments of long legs in the phylogeny.

(j) For such data, SCITE is the most appropriate method for lower-quality data (many unknown and incorrect base assignments), and the authors have used it already (as reported in the Supplementary information). SCITE produces a mutation tree, and thus a clone phylogeny and branch lengths. Those are expected to be most appropriate for the current application. So, why MPBoot is being applied and why the ML method estimates branch lengths are not clear.

(k) Also, in Supplementary Figure S3, the method used for S3b needs to be mentioned. MPBoot phylogenies in Figures S3e and S3f have different bootstrap values?

(l) The article also does not present enough information on how the completeness of sequence alignments. What was the percentage of positions with missing data, and was there a need to use methods to impute missing data and correct sequencing errors (e.g., SCITE, BEAM, etc.). It would also be useful to know how often cells were not heterozygous (e.g., LOH).

(m) As for standard bootstrapping, it is expected to be low for short branch lengths, i.e., those with only one or a few mutations, due to standard properties of the site-sampling bootstrap (e.g., a branch must contain at least 3 mutations before the expected bootstrap support can ever reach 95%). So, bootstrap support for early branching events is expected to be very low, but this uncertainty is unlikely to affect the downstream inference.

(n) It will be useful to be more explicit about how the trees were rooted. I gleaned that somewhat from the protocol noted, but was an actual sequence of (e.g., 0's) used in MPBoot analysis to root the tree? SCITE will automatically assume that but was that done in MPBoot, or is midpoint rooting used? This clarification is important to know if the early phase was symmetric cell division, leading to a balanced tree or a highly asymmetric one.

(o) In the phylogenies presented, HSCs and MPPs are not immediately clear. Maybe I missed it. It will be useful to mark those lineages if they are both included. Are the dynamics different for HSC and MPPs?

(p) Data sharing. While the raw data and the associated source code to summarize them are available, the availability of alignments (1/0/? states and base calls) used for MPBoot (and other evolutionary analysis) is extremely important for others to be able to reanalyze/reproduce/advance the results presented. In particular, providing those in a form that can allow one to use standard methods for dN/dS analyses will be important to reproducing results and scrutinizing patterns.

Author Rebuttals to Initial Comments:

“Clonal dynamics of haematopoiesis across the human lifespan”

Thank you for the opportunity to provide a point-by-point outline response to the reviewer comments. We were pleased that overall the comments were positive. The major changes to the manuscript that we have made include: (1) testing additional methods for reconstruction, linearisation and scaling of phylogenetic trees; (2) rewriting of the introduction and discussion to more clearly delineate the novelty of our data, and their relevance to ageing; and (3) considerably more detail on the computational methods used, together with release of scripts (Github) and intermediate datasets (Mendeley Data – doi: 10.17632/np54zjkvvr.1) that will enable readers to reproduce all findings in the manuscript.

Below, the reviewer comments are in blue, with our response in black and actions that we have undertaken for the revision in red and bold.

Referees' comments:

Referee #1 (Remarks to the Author):

Mitchell et al performed whole genome sequencing of 3579 single-cell-derived colonies from 10 healthy individuals to study the clonal dynamics haematopoiesis. They carefully analyzed these data to confirm and refine what has been known about clonal haematopoiesis such as the linear accumulation of point mutations and indels over time and the shortening of telomeres with increasing age. Additionally, the authors inferred phylogenetic trees and analyzed the subclonal composition. They found that in adults aged <65 haematopoiesis is highly polyclonal while in adults aged >70 expanded subclones made up 32% to 61% of the sequenced colonies. Moreover, they could estimate the timing of branching points in the phylogenies, the occurrence of mutations, and multiple independent losses of chromosome Y. To investigate potential explanations for the different subclonal compositions, the authors developed a simple mathematical model.

Overall, I have to admit that I struggled with the text. Mostly because of the long introduction and the repetition of known results. It's not really clear which of these results are new and novel. That actually starts in the abstract which does say “We sequenced..”, but then continues with rather generic statements and it is unclear which of these statements are new or novel findings of this particular study. Most findings seem rather incremental refinements.

We accept that our original submission did not clearly distinguish between the past literature and the key advances in understanding offered by our work. The major novelties in our paper rest on critical methodological differences from past publications which, when deployed at scale, have generated high-resolution and unbiased insights into clonal dynamics over the human lifespan. We summarise the key methodological differences first, and follow this with a summary of the novel biological insights that emerge.

Key methodological features

1. *Clone-specific resolution* – The critical difference between this work and the majority of the clonal haematopoiesis literature is that we have sequenced single HSC/MPP-derived colonies. Clonal haematopoiesis was discovered through sequencing DNA from bulk blood cell populations¹⁻⁵. This approach has the advantage that large numbers of patient samples can be assessed – indeed, several of the key initial papers leveraged existing exome data from large biobanks sequenced for germline association studies. However, bulk sequencing is really only capable of detecting mutations in clones that contribute more than 5% of blood production, so is strongly biased towards the one or two most numerically dominant clones that might be present. Our design provides mutations at the resolution of individual HSC/MPPs. First, we observe many ‘singleton’ colonies in all patients – HSC/MPPs unrelated to any others sequenced, for which the clonal fraction must be considerably lower than what could be detected with bulk sequencing. Second, we directly report clonal dynamics in long-term blood-producing HSC/MPPs and do not infer these from sequencing of relatively short-lived mature cells, eliminating any biases due to selection during differentiation.
2. *Genome-wide coverage at scale* – Most studies to date have used targeted sequencing of either the exome or the few hundred known cancer genes. Furthermore, the data analyses have focused on known drivers of myeloid leukaemia, which means that selection acting outside of these ~200 genes has proved difficult to assess. Our study used whole genome sequencing, undertaken on over 3,500 colonies. This provides considerable statistical power for quantifying the patterns of positive selection, and is able to discover clonal expansions that occur in the absence of known drivers.
3. *Accurate reconstruction of lineage trees* – With bulk sequencing, it is impossible to determine which mutations travel together in the same clone, or their temporal relationships to the clone’s expansion. With single-cell colonies, we can reconstruct highly accurate phylogenetic trees, which we can couple with the predictable mutational clock of blood cells to infer the distribution of clonal expansions across the HSC/MPP compartment and their age of onset.

Novel biological insights

1. *Universal, rapid decrease in clonal diversity after age 70 years* – Based on identifying mutations in bulk sequencing data, papers have typically reported 10-20% prevalence of clonal haematopoiesis after the age of 70 years¹⁻⁵. With error-corrected sequencing for known drivers, the prevalence increases⁶, but these studies cannot estimate the number of expanded clones in a given individual. In our paper, we show that **all** elderly individuals sequenced have decreased clonal diversity. We demonstrate that this loss of diversity is inevitable – the rate at which driver mutations enter the HSC population together with the large population size mean that many clones, typically 10-20, will have undergone sizable expansions by the age of 70. The accurate phylogenetic trees inform us that these expansions are entirely independent of one another, sharing no clonal relationships, something that could not be confidently inferred from bulk sequencing. Together, these clones account for 30-60% of all blood production in elderly individuals – this is a considerably larger fraction than reported in the bulk sequencing studies.
2. *Clonal expansions in the absence of known driver mutations* – An unexpected finding in our study is that only a quarter of the clonal expansions in elderly haematopoiesis have known driver mutations. Indirect evidence for the existence of clones without known drivers has been previously reported from patterns of synonymous variants⁷, but the low sensitivity for mutation-calling when variant allele fractions are in the 1-5% range makes these inferences challenging. Here, with multiple colonies from an expanded clone, each sequenced at genome-wide coverage, we have

very high sensitivity for calling individual mutations, and therefore have high confidence that these clones do not carry known drivers. Furthermore, our observation of an elevated dN/dS ratio across the whole genome provides independent genetic analysis for positive selection acting on mutations in genes yet to be discovered.

3. *Clonal expansions start decades earlier* – Clonal haematopoiesis is generally assumed to be an age-related phenomenon, but we show that it has its origins in mutations acquired in childhood and early adulthood. The phylogenetic trees provide accurate insights into age-of-onset of driver mutations: for example, of 13 expanded *DNMT3A* mutations, 2 were acquired before the age of 10 years, 3 before the age of 20, and the remainder before the age of 40. Likewise, the distribution of branch-points on the phylogenetic tree for clones without known drivers also shows that they began in childhood and early adulthood. These insights could not be achieved with bulk sequencing studies – the exponential nature of these expansions means that for the vast majority of those intervening decades, each clone is well beneath the detection limits of bulk sequencing.

We would argue that these observations have important implications for our understanding of ageing; considerably more substantial than “incremental refinements”. First, they provide an intriguing model for how lifelong, constant accumulation of molecular damage (in this case, somatic mutations) can lead to abrupt, organ-wide deterioration after the age of 70 years – a fundamental conundrum of ageing⁸. Second, the wide repertoire of available driver mutations means it is statistically inevitable that a few HSCs will prevail in their Darwinian struggle, but the properties of each successful clone will be shaped by its specific driver – this suggests a hypothesis for how the phenotypes of ageing can be both universal but also variable between individuals. Third, the progressive expansion of these clones over many decades provides a plausible mechanistic link to other lifelong disease processes such as atherosclerosis – an established but unexplained clinical correlate of clonal haematopoiesis^{3,9}.

We accept the reviewer’s criticism that the manuscript as originally submitted did not sufficiently distinguish the novel insights of our study from the existing knowledge. **We have therefore rewritten the Introduction to emphasise what was known and establish why our experimental design enables both higher resolution and less bias in inferring clonal dynamics (‘Introduction’, p. 3). We have also rewritten the Discussion to draw out the key biological insights that have emerged (‘Discussion’, pp. 9-10).**

I also think that the computational methods should be described in much greater detail. For example, it’s not clear to me how the phylogenies in Figs. 2 and 3 were generated. Although there is the “Construction of phylogenetic trees” section in the supplement, it only references the phylogenies in the Extended Data Figures section.

We agree that the computational methods for constructing the phylogenetic trees need more detailed explanation. The key steps in the algorithm to generate the trees shown in **Figures 2-3** are as follows:

1. Generate a ‘genotype matrix’ of mutation calls for every colony within a donor – Our protocol, based on whole genome sequencing of single-cell-derived colonies, generates consistent and even coverage across the genome, leading to very few missing values within this matrix (ranging from 0.005 – 0.034 of mutated sites in a given colony across different donors within our cohort). This generates a high degree of accuracy in the constructed trees.
2. Reconstruct phylogenetic trees from the genotype matrix – This is a standard and well-studied problem in phylogenetics. The low fraction of the genome that is mutated in a given colony (<1/million bases) coupled with the highly complete genotype matrix mean that different phylogenetics methods produce reassuringly concordant trees. We used the *MPBoot* algorithm for the tree reconstruction, as it proved both accurate and computationally efficient for our dataset.

3. Correct terminal branch lengths for sensitivity to detect mutations in each colony – The trees generated in the previous step have branch lengths proportional to the number of mutations assigned to each branch. For the terminal branches, which contain mutations unique to that colony, variable sequencing depth can underestimate the true numbers of unique mutations, so we correct these branch lengths for the estimated sensitivity to detect mutations based on genome coverage.
4. Make phylogenetic trees ultrametric – After step 3, there is little more than Poisson variation in corrected mutation burden among colonies from a given donor. Since these colonies all derived from the same timepoint, we can normalise the branch lengths to have the same overall distance from root to tip (known as an ultrametric tree). We used an ‘iteratively reweighted means’ algorithm for this purpose.
5. Scale trees to chronological age – Since mutation rate is constant across the human lifespan, we can use it as a ‘molecular clock’ to scale the y axis of the ultrametric tree to chronological age.
6. Overlay phenotypic and genotypic information on the tree – The tip of each branch in the resulting phylogenetic tree represents a specific colony in the dataset, meaning that we can depict phenotypic information about each colony underneath its terminal branch (the coloured stripes along the bottom of **Figures 2-3**). Furthermore, every mutation in the dataset is confidently assigned to a specific branch in the phylogenetic tree. This means that we can highlight branches on which specific genetic events occurred (such as *DNMT3A* or other driver mutations).

The critical steps in this process are steps 2 and 4. We have now performed more extensive comparison of different methods for these two stages. For step 2 (tree building), we have compared two methods – SCITE and MPBoot for 7 of the 8 phylogenies (we were unable to get the 8th phylogeny, KX004, to complete its SCITE analysis within the 4-week time limit set on our compute farm). Reassuringly, in all cases there was high concordance in the phylogenies produced by the two approaches (Robinsons-Foulds distance < 0.07). Most differences that did emerge affected the precise arrangement of some early embryonic branch points – these differences would not have an impact on the key downstream analyses in the manuscript. We have also formally compared the summary statistics used in the Approximate Bayesian Calculation models from the phylogenies inferred using MPBoot versus SCITE. We found that when the range of summary statistics we utilise for the ABC modelling were assessed at 4 timepoints, in 5 out of 7 individuals, the statistics for the MPBoot and SCITE phylogenies were identical. For the other 2 individuals, 6 of 32 (KX007) and 4 of 32 (KX003) summary statistics calculated for each phylogeny were discordant but by a negligible amount (2 or less). These overall highly concordant findings further confirm that the choice of tree-building approach would not have altered the conclusions of the downstream modelling analyses.

For step 4, making the phylogenetic trees ultrametric (normalising them to identical total branch lengths), we have now tested a Bayesian algorithm, Rtreefit, developed by Williams *et al*¹⁰ for somatic mutations in blood cancers in addition to the iterative reweighted means algorithm used in the first submission. In brief, Rtreefit is a Bayesian model for converting trees based on mutation counts into time-based trees. The mean branch timings are directly sampled from the posterior distribution and, by construction, the resulting trees are guaranteed to have a root to tip distance that matches the sampling age of the colony. Reassuringly, we found extremely high concordance between our custom ‘iteratively reweighted means’ approach and the Bayesian approach. In all phylogenies the R² for branch length comparisons between the two approaches was > 0.99.

We have now included more information about how the phylogenies were inferred and presented in the figure captions (Fig. 2) and referenced the Materials and Methods where additional details have been provided as above. We have also given more information regarding how MPBoot was run and what percentage of sites in our genotype matrix of shared mutations had ‘missing data’ in

the Materials and Methods. The code used in linearisation of phylogenies (to make them 'ultrametric' with equal branch lengths) has also been released on Github.

In terms of providing additional information for other computational methods, we have included the formula for calculating driver mutation number in the "Driver mutation acquisition rate estimation" section of the Materials and Methods. Further details on the ABC methods and PPC (posterior predictive checks) calculations have been added to the technical appendix on these methods.

All code used is available on github and data for running scripts to fully reproduce the analyses is available on Mendeley data (doi: 10.17632/np54zjkvrxr.1). **We have included a guide to available data in the Materials and Methods section "Data availability".**

Fig 1c,d: It would be great if there would be marks for means/median (perhaps even quantiles such as in Fig. 1h). Due to the overlapping clouds of points, it is hard to see where most of these points lie.

We have added box-plots showing median and interquartile range of mutation burden for each donor in Figure 1c,d as suggested.

Figs. 2 and 3: These figures are very interesting. How were these phylogenies inferred? The figure captions could provide some more details. Additional summary statistics about number of driver mutations, expanded clones, fraction of expanded clones, average onset times of these clones/drivers would be helpful. The information in Fig. 5a and b could be integrated directly to better put this information in context.

See response above regarding additional information on how **Figures 2-3** were generated. **We have added more detail on the steps involved in generating the trees to the main text ('Whole genome sequencing of HSC-derived colonies', p. 4). We have added more clarification on the construction and interpretation of the trees in the Figure legends, as suggested.**

We see the main purpose of **Figures 2-3** being to display within-donor information, such as phylogenetic structure and timing of driver mutations and expanded clades. In contrast, we see the main purpose of **Figure 5a-b** being to display comparisons of clonal diversity and clone size distribution across different donors. Because **Figures 2-3** are already very information-dense, we would prefer to avoid overlaying summary statistics such as shown in **Figure 5a-b** on the trees. **However, we have provided more clarity in the main text for interpretation ('Qualitative change in HSC population structure after 70 years of age', p. 5). We have also created a new figure summarising the inferred timing of acquisition of both driver mutations and expansion of clades with no known drivers (Extended Fig. 6a, shown below as Reviewer Figure 1).**

Reviewer Figure 1 (included in manuscript as Extended Figure 6a). Plots illustrating the timing of acquisition of driver mutations and onset of clonal expansions respectively. These timings are inferred from the timing of the corresponding branches in the phylogenies depicted in Figs. 2 and 3. Bars are colours by gene mutation or blue for expanded clades with no known driver. Age 0 denotes the time of birth and black dots illustrate the age at sampling.

Line 210: Doubled “in mice”

Many thanks, we have amended this.

Line 302: It is not clear to me how the authors converted the dN/dS ratios into estimated numbers of driver mutations. Can you please elaborate?

The dN/dS parameter is widely used in evolutionary genetics to infer patterns of selection^{11,12}, recently adapted for cancer and somatic mutations¹³. It is essentially a measure of how far the observed number of non-synonymous mutations diverges from the number that would be expected from the synonymous mutation rate, after correction for mutation spectrum¹³. It is underpinned by the assumption that synonymous mutations evolve neutrally, and selection only acts on non-synonymous mutations. For example, a dN/dS ratio of 1 means that we observed exactly the same number of non-synonymous mutations as we would have expected for the number of synonymous variants. A dN/dS ratio of 2 means we observed twice as many non-synonymous mutations as expected, implying that half of the observed non-synonymous mutations occurred as expected for the background mutational processes, while the other half have accumulated through positive selection. From this, with a total number of observed non-synonymous mutations, we can estimate the number of driver mutations in excess of the background expectation (noting that this is an underestimate of the true number in the presence of any negative selection).

This, then, is the intuition for the formal mathematical exposition. Given an observed number of non-synonymous mutations, n_{NS} , and an estimated dN/dS ratio, ω_{NS} , the formula for the expected number of drivers, n_D , is as follows:

$$n_D = \frac{(\omega_{NS} - 1)}{\omega_{NS}} n_{NS}$$

To give a worked example using missense substitutions in our dataset, we estimated the overall dN/dS ratio to be 1.06 with the 95% confidence interval to be 1.03 – 1.09 (**Figure 5d**). We observed a total of 16,536 non-synonymous mutations. The number of excess missense mutations, then, is calculated as $(1.06 - 1)/1.06 * 16536$, which works out at 936, with the lower bound on the confidence interval as $(1.03 - 1)/1.03 * 16536$, which equals 482. This then equates to the estimation in the manuscript (p. 8) that 1 mutation in every 18 ($16536/936=17.7$) missense mutations is under positive selection.

We have included this intuition, formal equation and worked example in the Materials and Methods ('Driver mutation acquisition rate estimation').

Line 386: Time of oligoclonal haematopoiesis was also already predicted by the models presented in refs. 44 and 61. Similarly line 390, isn't the universal loss of clonal diversity in haematopoiesis an already known fact? It is also stated in that way in the first paragraph of the related manuscript by Fabre et al.

We agree that this particular paragraph of the Discussion was clumsily worded – as the reviewer states, previous mathematical models have indeed shown that increased cell turnover accelerates clonal advantage, bringing forward the time of their emergence. We apologise for this oversight.

We do not, however, believe that universal loss of clonal diversity in haematopoiesis is a known fact. As argued above, studies using sequencing data from bulk blood samples have estimated prevalence rates of known driver mutations of 10-20% of individuals aged over 70 years, with drivers present at very small clonal fractions, typically 1-5%¹⁻⁵ only – the corollary being, of course, that 80-90% of elderly individuals did not have drivers detected. A study using error-corrected sequencing for known drivers showed this prevalence does increase if considering mutations at clonal fractions less than 1%⁶ – this study found 1-6 variants per person on average at a median clonal fraction of 0.002, with no ability to determine whether those variants were carried by a single clone or more than one clone. Either way, such clone sizes would not be sufficient to meaningfully impact on global measures of clonal diversity such as the Shannon Diversity Index.

Our data suggest that age-related loss of clonal diversity is orders of magnitude more pervasive than estimated from these studies: (1) the prevalence of clones at >1% VAF will be universal after the age of 70 years, not 10-20% prevalent; (2) the number of expanded clones per individual will be 10-20, not 1-2; and (3) the fraction of overall haematopoiesis accounted for by expanded clones is 30-60%, not 1-5%. To this, three further insights not accessible with published methods can be added: (4) the age of onset of driver mutations and clonal expansions is decades earlier, typically before the age of 40; (5) the loss of clonal diversity emerges abruptly after the age of 70 years; and (6) the repertoire of genes conferring selective advantage on HSCs is considerably broader than those known to cause blood cancers. This is, of course, not a criticism of the previous studies – we are merely emphasising that using single-cell-derived, high-resolution methods for characterising haematopoiesis leads to an unbiased assessment of its clonal diversity.

We have rewritten the Discussion to remove the rather weak point about lifestyle and disease risk factors affecting the timing of clonal changes, replacing it with a clearer delineation of the novel insights from this study and its broader implications for ageing ('Discussion', pp. 9-10).

One final remark: I have no idea why the authors choose to have the figure captions not next to the figures and instead even put them in different files for the extended data figures. There is so much material to review and this makes it unnecessarily hard.

Many apologies for this. **We have addressed this in the resubmitted version, placing each legend immediately before the corresponding figure.**

Referee #2 (Remarks to the Author):

In this manuscript, Mitchell et al present an elegant analysis of whole-genome sequences from single-cell derived colonies of HSC/MPPs across haematologically normal subjects across a wide age range. The conclusions are well-supported, are certain to be of interest to the readership of Nature, and have vast significance for several reasons:

- (i) Only a ~1/5 of clones had known driver mutations, which has important implications for the clonal hematopoiesis field, in which the majority of studies to date have primarily focused on CH with known drivers
- (ii) The authors find that the landscape of CH drivers may be much larger than previously appreciated (and as compared to the landscape of events that drive malignancy), that 1/34 to 1/12 non-synonymous mutations may be drivers
- (iii) Their findings refine estimates of the number of long-term HSCs in humans
- (iv) The authors report a striking difference between the polyclonal nature of hematopoiesis in younger individuals as compared to the oligoclonal landscape in older individuals, and compellingly, find that these age-related dynamics and abrupt emergence of oligoclonality appear to be explainable by simple, predictable processes

Specific points

- In EDF 1b, the data reveal that not all sorted cells were able to form colonies. Could the authors comment on whether this may represent a source of bias? Could the ability to form colonies track with some of the measurements in their paper such as telomere length, number of somatic mutations, which would in turn influence some of their quantitative inferences?

The reviewer is correct that not all sorted cells were able to form colonies, and in theory, this could represent a source of bias if there were a specific cell subtype, driver mutation or telomere length with reduced colony-forming capacity *in vitro*.

Several points argue that the colonies we generated are broadly representative of the historic HSC dynamics we infer:

- First, the protocol we followed to generate single-cell-derived colonies is considerably more efficient than methods used in previous studies where colonies were grown in semi-solid medium. In the latter conditions and using published protocols, our group has estimated that fewer than 10% of single HSC/MPPs will produce a colony, whereas our liquid medium single cell flow-sorted method is at least 5 to 8-fold more efficient.
- Second, colony-forming efficiency with our method was broadly equivalent across all donor ages and tissues examined (**Extended Figure 2b**), a finding that we have seen repeatedly in our laboratory^{14,15}. This means that any potential bias would apply across the lifespan and should therefore not affect the main conclusions of our study.
- Third, the rates of colony-formation efficiency were in the range 40-75% for each adult in the study, meaning that the cells we do access with this protocol represent a majority population of HSC/MPPs in each individual.
- Fourth, the genomic data we observe from the multiple colonies per individual in aggregate closely match data published from bulk sequencing studies – telomere lengths of a few kilobases

shortening at $\sim 30\text{bp}/\text{cell}/\text{year}$ are close to published estimates¹⁶; mutation burdens of colonies are identical to those obtained by single-molecule sequencing of bulk blood cells¹⁷; and the distributions of driver mutations and loss-of-Y across donor ages follow similar trajectories as those from bulk studies¹⁻⁵ after allowing for the greater sensitivity of our approach.

- Fifth, our inferences of clonal dynamics depend upon the genomic records of historic HSC divisions encoded in the phylogenetic trees, rather than the specific current properties of the cell whose colony we have actually sequenced. That is, even if we sequence a colony arising from a differentiated progenitor cell in a 70-year old, that cell will only have been a progenitor for the most recent year or two – branch-points in that cell’s lineage tree from 30-40 years earlier will reflect historic stem cell divisions, not its current status.

We have undertaken further analysis to address the question of bias between cells that successfully seeded colonies and those that did not, leveraging the index flow-sorting data collected from the donor samples. For each cell that was flow-sorted into the culture plates, we collected the data on fluorescence intensity of cell surface markers – from this, we compared the expression level of these markers between HSC/MPPs that successfully generated colonies and cells that did not. We have included the additional HSC markers CD90 and CD49f in this analysis¹⁸, although these markers were not used in the sort gating strategy. This analysis shows there was no immunophenotypic difference in cells that were sequenced compared to those that were not in surface expression of CD34, CD38, CD45RA, CD90 or CD49f, across any of the donor samples (see **Reviewer Figure 2** below). This provides further reassurance that the colonies that we sequenced are broadly representative of the HSC/MPP compartment we model.

We have added a comment on the efficiency of colony formation with our liquid culture method compared to previously used methods to the Supplementary Methods. We have included Reviewer Figure 2 below as a new panel in Extended Figure 2 (panel c), and signposted this in the main text ('Whole genome sequencing of HSC-derived colonies', p. 3).

Reviewer Figure 2 (included as Extended Figure 1c). Box and whisker plots showing the range of fluorescence intensity levels for cells that successfully grew colonies (teal-green/sequenced) versus

those that did not (salmon-pink/not-sequenced), broken down by research subject (rows) and cell surface marker (columns). Boxes show the interquartile range and the centre horizontal lines show the median. Whiskers extend to the minimum of either the range or 1.5× the interquartile range, with outlier cells beyond the whiskers shown as individual points.

- Given that X-chromosome mosaic events are common, why do the authors primarily only focus on loss of Y chromosome and not X-chromosome events as well?

The copy number algorithm we used, ASCAT, does call chromosomal aberrations and focal copy number changes genome-wide. In fact, on re-reviewing these data, we did not identify any X chromosome changes, either gains or losses. We believe this has interesting implications – after all, the inactive X chromosome and the Y chromosome have similar gene content and expression from the pseudoautosomal regions. Outside these regions, with a few exceptional genes, the inactivation of the X chromosome in females is strongly maintained. As a consequence, only a small handful of active genes would be different between a male cell losing the Y chromosome and a female cell losing the inactive X chromosome – and yet we see 94 independent occurrences of the former and 0 of the latter across the dataset. This suggests that either the rate of mitotic loss is vastly different between the two chromosomes (a hypothesis we have not found any data to support) or there is a growth-suppressor gene on chromosome Y that results in a growth advantage to cells upon loss-of-Y. Our observation that clones with loss-of-Y are more expanded than cells with retained Y, the vast majority of which are euploid, provides additional evidence for the hypothesis of selective advantage.

We have rephrased the main text to clarify this point ('Mutation burden and telomere lengths', p. 5) and expanded the Materials and Methods to make it clear that the copy number analysis covers the whole genome ('Structural variant and copy-number calling').

- Stem cell biologists have focused on the role of the bone marrow microenvironment with age. Could the authors speculate on the known changes to the microenvironment at older age and how that might influence the dramatic changes in clonal diversity with age?

This is an intriguing question – as the reviewer notes, it is widely accepted that ageing results in microenvironmental changes in the bone marrow. For example, ageing is associated with changing cytokine concentrations in the bone marrow. Increased signalling of specific cytokines could synergise with specific driver mutations (as has been shown experimentally for *TET2* and *IL6*¹⁹ as well as *DNMT3A* and *IFN γ* ²⁰), conferring an increased proliferative advantage on these clades, or cause a more general increase in cell division rates across the whole population. Other microenvironmental changes include the accumulation of adipocytes as well as remodelling of structural components such as the vasculature, extracellular matrix and bone. These are likely to alter the niche for HSCs and potentially shape the selective landscape the clones are competing in, which may alter clonal dynamics. As a further complication, it is entirely feasible that clones with driver mutations actively remodel their own niche – this has been documented in mouse models of *Tet2* mutations, for example²¹.

Our data report on *relative fitness* of different clones within the same individual; namely, HSCs competing with one another in the same haematopoietic microenvironment. For microenvironmental ageing to promote the loss of clonal diversity we observe, it would still have to act in a clone-specific manner. Interestingly, our data show that all of the expanded clades began their expansion in the first half of life – this suggests that their selective advantage is lifelong, emerging before the local marrow microenvironment has itself undergone age-related remodelling.

We have incorporated further comment on the possible role of microenvironmental changes with ageing in the Discussion (p. 11).

• How do the authors think that the clones without known driver mutations contribute to the development of myeloid malignancies?

This is a fascinating question. Most previous studies of clonal haematopoiesis have focused on mutations in known blood cancer genes, and have shown a consistently elevated risk of subsequent blood cancer in those individuals carrying such known drivers. The risk arising from clones without known driver mutations has been much less comprehensively studied. Probably the most informative data come from a reanalysis of 11,262 bulk whole genome sequences from the deCODE cohort²². Here, the authors distinguished three groups of subjects: (1) those without detectable clonal haematopoiesis (acknowledging the low sensitivity of bulk WGS); (2) those with clonal haematopoiesis carrying known drivers; and (3) those with evidence for clonal haematopoiesis but no known drivers. Interestingly, the latter two groups both had elevated risk of all-cause mortality and future haematologic malignancy, after correction for confounders – furthermore, this elevated risk was broadly equivalent whether or not known drivers were present²².

These data, while suggestive, do not prove that clones without known driver mutations are themselves at greater risk of leukaemic transformation – it could be that their presence denotes a microenvironment of increased cell turnover and selective pressure conducive to clonal outgrowth, with future malignancies arising from other, independent clones. We have observed this in other settings, such as chronic liver disease, which represents a clear field effect for tumour evolution, but where the diseased microenvironment selects for clones with mutations in metabolism genes that do not themselves transform to malignancy²³.

At present we do not have any evidence that the clones without known driver mutations have themselves an increased risk of malignant transformation. For example, screening 534 published AML genomes^{24–26} for variants in the two relatively new genes described, we identified only one possible oncogenic mutation in *ZNF318* and no mutations in *HIST2H3D* (**Table S8**).

We have included these points in the revised Discussion (pp. 10-11).

Referee #3 (Remarks to the Author):

This is an excellent research article in which authors report single-cell sequences of hematopoietic stem cells and multipotent progenitors (HSC/MPPs). Several interesting observations and claims are made based on molecular evolutionary analysis with dating. I am very supportive of its publication. But, I do have many major/minor comments and concerns that will help clarify important findings reported and claims made.

(a) Single-cell sequences reveal that hematopoiesis is massively polyclonal, and there is high clonal diversity. It will be useful to include a comparison with polyclonality and diversity of stem cells in other tissues or organs.

The haematopoietic system is distinctive among organ systems for being well-mixed – when we compare the variant allele fraction of mutations in a single bone marrow draw with that from peripheral blood, we find strong correlation²⁷, confirming that recirculation of stem cells is sufficiently frequent that spatial biases are negligible over the timescales we are interested in here.

In contrast, studies in solid tissues have revealed that stem cell clones exhibit considerable spatial organisation²⁸. For example, human colonic epithelium is organised by crypt, with 5-15 independent stem cells at the base of each crypt dividing neutrally to generate clonal sweeps every 1-2 years^{29,30} – with ~10 million crypts per colon³¹, which undergo crypt fission events only every 2-3 decades³⁰, this implies many tens of millions of independent colonic stem cells per adult human. Likewise, detailed lineage tracing of human prostate has revealed that each of the 24-30 independent glandular subunits are laid down *in utero* by 5-10 embryonic cells – these then proliferate to seed stem cells throughout the ductal tree which, following a wave of further proliferation and duct formation during puberty, enter a relatively quiescent phase of local stem/progenitor cell tissue maintenance in adulthood³². In squamous tissues, such as oesophagus^{33,34}, bronchial epithelium^{35,36} and skin³⁷, driver mutations accumulate steadily with ageing, causing exponential clonal expansions³⁸, with competition occurring predominantly at clone boundaries³⁹.

The effects of ageing on stem cell clonal dynamics in solid tissues have not been extensively studied to date. However, the effects of disease and of toxicity have had some initial evaluation. In intestine, it is clear that inflammatory bowel disease leads to considerably larger clonal expansions than seen in normal individuals, partially driven by selection for driver mutations that protect against the inflammatory process and partially driven by the regenerative pressure of a relapsing-remitting disease course⁴⁰⁻⁴². Likewise, while normal liver is a tightly knit patchwork of clones as small as 100-1000 hepatocytes, chronic liver disease is characterised by considerably larger clones, millimetres to centimetres in size, often accounting for entire cirrhotic nodules^{23,43-45} – again, these clonal expansions are driven by a combination of selection for protective driver mutations and the regenerative milieu arising from sustained hepatocyte toxicity. Interestingly, cellular toxicity, such as that arising from tobacco smoke in bronchus³⁵ or ultraviolet light in skin⁴⁶, also increases the rate of driver mutations and clonal expansion in solid tissues.

Taken together, these data show that tissue maintenance in adults in solid organs is typically a hugely polyclonal process, strongly shaped by the spatial organisation of the tissue. Clonal competition therefore remains local, and the opportunities for massive clonal expansion remain limited under normal physiological conditions. However, with disease or toxicity, when selective pressures are more pronounced and coupled with increased regenerative pressure, clonal expansions can be sizable, encompassing (square or cubic) millimetres to centimetres of tissue. Furthermore, convergent evolution, where the same genes are recurrently mutated and positively selected in independent clones, can lead to an analogous situation to that we have observed here in blood, where 20-80% of all epithelial cells within, say, skin^{37,46}, oesophagus^{33,34}, endometrium⁴⁷ or bronchus³⁵ carry mutations in specific driver genes.

We have summarised these points in the main text (Discussion, p. 10), and included the text above as Supplementary Note 2.

(b) Using parameters estimated from the data, empirical simulations show concordance with the observations under a simple model. While it is exciting, one wonders how the simulation consistently achieves an inflection point in the change in diversity in the last few years of the simulated data. Is it a deterministic outcome by design or by model properties? An intuitive explanation will be useful.

We agree that it is important to explain the basis for the Approximate Bayesian Computation framework for the general reader; thankfully, it is relatively intuitive. In the initial phase, we generate hundreds of thousands of different simulations of haematopoietic stem cell compartments. Each

simulation follows exactly the same assumptions – constant population size of HSCs during adulthood; linear entry of driver mutations into the HSC compartment across life; fitness coefficients of drivers drawn from a gamma distribution; fitness coefficient is constant with time. We do not know *a priori* the values for several of these key parameters (distribution of fitness coefficients, rate of driver mutation entry), so each simulation takes a draw for these parameters from relatively uninformative prior distributions.

We then take the huge number of simulations and the phylogenetic trees that they produce, and compare informative summary statistics from simulated trees against the same summary statistics generated from our real phylogeny data. Clearly, many of the simulations will generate trees that are very different to those observed – for example, low driver mutation rates generate many trees with no clonal expansions; fitness coefficients that are too high generate single, massive clonal expansions rather than the oligoclonal patterns we observe.

From the small fraction of simulated trees that best match the observed data, then, we can extract posterior distributions of the parameters we are most interested in. The formal mathematics for this is well-established^{48,49} – reassuringly, the posterior distributions we extract using these methods are a well-defined subspace of the prior distribution, suggesting that the observed phylogenetic trees contain considerable information about, and constraints upon, the underlying distribution of the key parameters.

With this intuition for how the modelling works, then, we can see that the inflection point in clonal diversity from the age of 70 years observed in the simulations is not an outcome by design – we do not build such an inflection point into the models as an explicit feature. Rather, it is a data-driven outcome of the simulations. That is, only a relatively narrow window in estimates for the rate of driver mutation acquisition and distribution of fitness coefficients are compatible with the observed phylogenies (**Extended Fig. 9b; Supplementary Fig. 12**) – pleasingly, this narrow window of parameter estimates generates simulations that match the inflection point of sharply reduced clonal diversity after the age of 70 years that we observe in the real data (see **Reviewer Figure 3** below).

To better illustrate the inflection point in the simulated data, we have generated a new figure (Fig. 5g, shown below as Reviewer Figure 3) that illustrates the median, interquartile range and 95% posterior intervals for Shannon Diversity Indices calculated yearly from age 10 – 100 for 10,000 HSC population simulations run utilising the optimal parameter values for driver acquisition rate and fitness effects derived from our driver ABC modelling work. The figure helps demonstrate that the inflection point is determined by the model properties, specifically the optimal parameters we derived from the phylogenies, and is not built into the model design. We have also included the intuitive explanation of the Approximate Bayesian Computation framework above as Supplementary Note 1, signposted in the main text ('Modelling positive selection in HSCs', p. 8).

Reviewer Figure 3 (included as Figure 5g). Median, interquartile range, and 95% posterior intervals for Shannon Diversity Indices calculated yearly for 10,000 HSC population simulations run utilising the optimal parameter values for driver acquisition rate and fitness effects derived from the ABC modelling approach. The point estimate for the shape and rate parameters of the gamma distribution were shape = 0.47, rate = 34. The point estimate for the number of drivers with $s > 5\%$ entering population per year = 200.

(c) The finding that older individuals have decreased clonal diversity is interesting, especially that it became pronounced only after the age of 70. It will be great to know if there is something special about this timing from a functional genomics point of view?

We also found the observation that clonal diversity diminishes sharply after the age of 70 interesting. We hypothesise that this particular timing arises as a consequence of our species' historic survival-fecundity curves – our past demographic structure has exerted evolutionary pressure on the germline genome to find strategies to reduce blood cancer risk up till the ages of 50-65 years. However, the relatively low historic contribution of individuals over this age to successful reproductive output substantially reduces that evolutionary pressure. This combination of progressive mortality and declining fecundity with age generates what has been termed a 'selective shadow', in which the force of natural selection rapidly reduces to zero at ages beyond which further reproductive output is negligible⁵⁰.

With this evolutionary perspective, it is then interesting, as the reviewer poses, to speculate on the functional basis of the evolutionary strategies that maintain clonal diversity until the age of 70. From our modelling, the parameters that clearly shape the age at which clonal diversity declines are:

- The population size of HSCs and generation time (average time between symmetric self-renewals);
- Rate of entry of driver mutations into the HSC compartment (a function of both the overall mutation rate and the fraction of those mutations that confer selective benefit);
- The magnitude of the fitness benefit conferred by the driver mutations.

We believe, therefore, that each of these parameters has had sustained (germline) evolutionary pressure to maintain a high degree of clonal diversity until the age of 60-70 years. Evolutionary pressure on germline variants in genes controlling these dynamics – HSC population size and turnover, somatic mutation rates, distribution of fitness benefits – has historically acted to maintain robust, polyclonal haematopoiesis until age 60-70, but no further. Thus, the distribution of fitness effects we infer (**Figure 5e**) is precisely this shape because of that evolutionary pressure – a right-shifted distribution with stronger drivers would cause sufficiently frequent blood cancers (or other deleterious

phenotypes) in reproductively active humans to exert evolutionary pressure; but the selective shadow means evolutionary pressure for driving the distribution further to the left peters out.

Supportive evidence for this comes from considering other mammalian species with very different lifespans. For example, somatic mutation rates have evolved such that the end-of-life mutation burden is remarkably similar across different mammals, despite there being 50-fold variation in lifespan⁵¹. Considering HSC-specific parameters, shorter lived species such as mice have a smaller HSC population (~5,000-10,000)^{52,53}, with a shorter generation time (~5 weeks)⁵⁴⁻⁵⁶ – smaller population size and more rapid generations both lead to accelerated opportunity for clonal expansions, but presumably a maximal lifespan of 2-3 years in a mouse limits the risk that these will result in decreased reproductive output. Some degree of decreased clonality has already been observed in elderly mice⁵⁷ and macaques⁵⁸. It would be a fascinating follow-up experiment to assess whether loss of clonal diversity towards the end of the natural lifespan is seen systematically across many mammalian species.

We have now elaborated these ideas in the Discussion (p. 10).

(d) An interesting observation is made about the association between the loss of the Y chromosome and the increase in the representation of the descending lineages. This is interpreted as a selective benefit. But, I wonder how the alternative explanation of the relaxation of negative selection is excluded. Intuitively, loss of the Y chromosome will reduce the number of proteins encoded, reducing the degree of purifying selection, especially if any of the protein was functionally used (e.g., dN/dS was smaller than others). Of course, if it is not possible to exclude the relaxation of purifying selection, alternatives need to be mentioned together.

The hypothesis of relaxed purifying selection acting on the Y chromosome is an interesting one – perhaps there is a moderate rate of chromosomal aneuploidy across all chromosomes, but the only cells that survive are those that lose Y because the gene content on Y is so low. Under this hypothesis, loss of any other chromosome would severely disadvantage the HSC, and it would therefore not survive to be sequenced.

Two observations from our data argue against this hypothesis. The first is that clones with loss-of-Y are larger than age-matched clones from the same individual. The vast majority (>99%) of these age-matched clones are euploid, which suggests that the clones with loss-of-Y have expanded to a greater degree than their wild-type counterparts; something more suggestive of positive selection than diminished negative selection. The second is that we do not observe equivalent frequency of losses of the inactive X chromosome in the elderly females in the study. The inactive X chromosome and the Y chromosome have similar gene content and expression within the pseudoautosomal regions, and only a few exceptional genes remain expressed outside these regions. Thus, the content of expressed genes for a male cell losing the Y chromosome and a female cell losing the inactive X chromosome would be broadly similar, and yet we see 94 independent occurrences of the former and 0 of the latter across the dataset.

We accept that these arguments, while suggestive, are not definitive, and have therefore acknowledged the possibility of relaxed purifying selection in the manuscript ('Genetic evidence for pervasive positive selection', p. 8).

(e) Pervasive positive selection is claimed based on dN/dS ratio being slightly larger than 1. First, the value is barely larger than 1, which could easily be due to the bias of the estimate. In somatic evolution, overall dN/dS close to 1 is universally observed. A slight deviation from this ratio can

occur because there is no way to eliminate bias in estimation methods. Since the effect size is small and could be due to bias, the estimate's variance for a large dataset becomes so small that the P values become nonsensical. Therefore, the authors have an unenviable task of proving that their dN/dS estimates are unbiased if they used dN/dS ratios slightly greater than 1 to indicate positive selection. Of course, this does not exclude the possibility that some mutations will likely confer an advantage for some cells to grow larger in numbers, but it is not the same as pervasive positive selection. Anyway, in Figure 5d, 95% CIs are rather close to 1 (even without considering bias in generating the estimate and its CI), so one needs to be very careful in claiming pervasive positive selection.

The reviewer is correct that a small bias in estimated dN/dS ratio could lead to an apparently significant excess when the dataset contains large numbers of mutations, as our does. In defence of the claim, we proffer four lines of argument and new analyses:

1. Correction for confounders in the dN/dS algorithm

The dN/dS algorithm¹³ is one of the best-in-class algorithms for quantifying somatic selection, as demonstrated by a recent pan-cancer comparison of different methods⁵⁹. One of the reasons for this is the rigorous approach it takes to correcting for the known variables influencing mutation rate across the genome, including replication timing, chromatin state and DNase accessibility⁶⁰. In addition, the model balances the predicted mutation rates from these global covariates with the observed synonymous mutation rate within a gene – this latter correction captures many of the unknown variables affecting mutation rates acting at a local level.

Furthermore, the algorithm corrects for the observed mutational spectrum¹³ – this is important because, for example, transitions are more likely to generate a synonymous mutation than transversions. The model parameterises all 192 rates representing the 6 different types of base substitution, the 16 combinations of bases 3' and 5' to the mutated base, and transcribed versus non-transcribed strand. This means that trinucleotide mutational signatures do not bias the overall dN/dS estimate.

2. Running dN/dS algorithm with greater stringency

In addition to running the dN/dS algorithm in its standard implementation, we have checked whether the overall estimates are materially altered if we run it using two adaptations to impose greater stringency.

The first adaptation was to run the algorithm excluding sites that are masked by our variant caller in both the numerator and the denominator (a total of 175 million sites genome-wide). Essentially, most somatic mutation callers, including ours⁶¹, have a 'normal panel' where sites that are frequently non-reference because of sequencing artefact or germline polymorphism are masked. Since germline polymorphisms have a dN/dS ratio $\ll 1$, this can lead to under-calling of synonymous somatic mutations relative to non-synonymous mutations. Running our algorithm with sites in this normal panel excluded from both numerator and denominator had minimal impact on the estimated overall value of dN/dS (1.0548, $CI_{95\%}=1.02488-1.0856$; versus 1.0586, $CI_{95\%}=1.02861-1.0895$ for the standard implementation). This argues that there is no bias arising from masking of true somatic mutations at germline polymorphisms.

The second adaptation was to run the dN/dS algorithm using correction for pentanucleotide sequence context. While a trinucleotide context captures virtually all of the effects of mutational signatures⁶², there remains the theoretical possibility that any signature extending beyond that may affect

synonymous mutations differently to non-synonymous mutations. To test this, we repeated the analysis using rates for the 6 mutation classes and 256 different combinations of 2 bases each side of the mutated base – the pentanucleotide context. This also had minimal impact on the estimated value of dN/dS for missense variants (1.0472, CI_{95%}=1.0155-1.0799; versus 1.0589, CI_{95%}=1.02852-1.0902 for the standard implementation) or the dN/dS for truncating variants (1.0788, CI_{95%}=1.0106-1.1516; versus 1.0569, CI_{95%}=0.99558-1.1220 for the standard implementation). Importantly, a pentanucleotide context covers the whole of the codon, no matter which base in the codon is mutated (whereas a trinucleotide context only covers the whole codon if the middle base is mutated) – this means that even if there were residual effects of mutational signatures beyond the pentanucleotide, they would not affect the mutated codon, and therefore the amino acid change.

3. Measuring dN/dS on simulated mutations

As a further check, we have now generated simulated mutations in the sequencing data and run the dN/dS algorithm. We took 19 BAM files from cord blood HSC/MPPs in our dataset for which zero coding mutations were identified by our variant caller. For each BAM file, we randomly chose 2000 sites in the exome to have simulated mutations, with the mutations following the same mutational spectrum as observed in the whole dataset. At each position with a mutation, we then extracted the reads reporting that base, and changed the base-call recorded at that base with 0.5 probability (to get average VAF of 50%), according to the following rules: change to mutant base if read reported reference base; change to reference base if read reported mutant base; change to the other non-reference, non-mutant base if read reported non-reference, non-mutant base.

The modified BAM files then underwent exactly the same process of variant calling as our real data. We verified that the majority of the simulated mutations were correctly called (the proportion dependent on sequencing coverage), and that the mutation spectrum was the same as that observed in the real data. In total, we called 29,008, which was close to our real dataset of 25,888 coding mutations. We ran the dN/dS algorithm over the simulated dataset and found no bias in the results, with a dN/dS ratio for all randomly simulated variants of 1.00. For the simulated missense mutations, the estimated dN/dS was 1.001 (CI_{95%}=0.974-1.028); and for simulated truncating mutations, it was 1.001 (0.956-1.067).

These simulations would have captured any biases in the estimation of dN/dS that arose from, for example, differential sequencing coverage across the genome, variant calling, variant filtering, variant annotation or the dN/dS algorithm. Instead, the estimates of dN/dS are almost exactly 1.00, as expected, with confidence intervals that do not overlap with those for our real data.

4. Frequency of clonal expansions concurs with estimates of numbers of driver mutations

Our argument that there is pervasive positive selection is not based exclusively on the genetic analysis. We also observe a number of clonal expansions in our dataset. We tested Approximate Bayesian Computation (ABC) models to see whether these expansions could be explained by neutral drift (for example, programmed changes in HSC population size). However, these neutral models were unable to capture the asymmetry of branching across the clades in the elderly subjects (many singleton branches interspersed with 10-20 considerably expanded clones – see **Extended Figure 7**).

Instead, to capture this asymmetry, we had to use models which included positive selection. The estimates of driver mutation rates that were required to generate trees that matched those we observed were about 2.0×10^{-3} /HSC/year. These models considered only driver mutations with fitness coefficient $s > 5\%$. Note that the ABC modelling uses no genetic data to arrive at this estimate – it is

purely based on how many drivers are required to generate the observed branching patterns in the real phylogenies of the older individuals.

Reassuringly, this estimate of the rate of drivers from the ABC modelling is broadly comparable with the estimate obtained from the dN/dS analysis. Overall, non-synonymous mutations accumulated in HSC/MPPs at a rate of 0.12/HSC/year ($CI_{95\%}=0.11-0.13$), with dN/dS estimates suggesting that 1/34 to 1/12 non-synonymous mutations were drivers (approximately 5%). This computes to a driver rate of $3.6-10 \times 10^{-3}$ /HSC/year estimated from direct genetic analysis, an estimate which would include drivers with $s < 5\%$ present in sequenced colonies.

We have included a more detailed discussion of potential biases in the dN/dS analysis in the Materials and Methods ('dN/dS analysis'), including these new analyses, and have signposted this in the main text ('Genetic evidence for pervasive positive selection', p. 8).

(f) Was the neutrality of amino acid variants tested by using a prediction tool? Anyway, it will be good to know what fraction of non-driver AA variants were likely non-neutral.

We have now performed this analysis using:

- SIFT4G (<https://sift.bii.a-star.edu.sg/sift4g/AnnotateVariants.html>)⁶³ and
- Polyphen2 (<http://genetics.bwh.harvard.edu/pph2/bgi.shtml>)⁶⁴.

Of a total 16536 missense mutations in our dataset, 5088 could be annotated by SIFT4G (38 in myeloid driver genes) and 4551 could be annotated by Polyphen2 (35 in myeloid driver genes). Approximately 42% and 45% of the annotated mutations were deemed to be 'deleterious' respectively (see Table below). If the same proportion of missense mutations is present in the dataset as a whole we would predict approximately 7000 'deleterious mutations', equating to around 1000 per adult individual.

	SIFT4G	Polyphen2
Total missense mutations in dataset	16536	16536
Number annotated	5088	4551
Number in known driver (excluded)	38	35
Deleterious	1998	2049
Possibly deleterious	319	762
Tolerated	2495	1709
Fraction deleterious	0.42	0.45
Predicted deleterious in whole dataset	6945	7441

We have added this analysis to the Supplementary Methods and also included a Supplementary Table 7 of annotated coding mutations used in the dN/dS analysis incorporating their SIFT4G and Polyphen2 annotations.

(g) How were the phylogenetic trees linearized? Since the phylogenies are not strictly clock-like, it is best to use a relaxed clock method that can convert a phylogeny into a tree (e.g., a Bayesian approach, if possible, or a rapid method such as RelTime for many sequences).

We would argue that somatic mutations within the haematopoietic system from the same individual represent an excellent and linear molecular clock. This is based on two properties of mutation burdens

in HSCs, which not only are conclusively demonstrated in our current dataset, but have also been reproducibly demonstrated by other researchers previously^{65,66}. First, across individuals, mutations increase linearly with age throughout adulthood, suggesting that the mutation rate is constant and clock-like. For our purposes, at a mutation every 2-3 weeks per cell on average, this rate is sufficiently high that over the time scales we are interested in (decades of life), the stochastics of the mutational processes average out. Second, within an individual, we see remarkably consistent estimates of the overall mutation burden per cell. As can be seen in the table below, the standard deviation of mutation burden across colonies is remarkably low compared to the mean, suggesting that there is only limited overdispersion in the observed burdens. Therefore, we do not need to account for material differences in mutation rates across different lineages, as we might need to do for inter-species phylogenetic trees, say.

Sample_ID	Mean mutation burden	Standard deviation
CB001	56.69	8.47
CB002	53.30	8.81
KX001	562.76	40.73
KX002	677.59	41.37
SX001	833.54	59.39
AX001	1103.43	69.83
KX007	1335.59	70.40
KX008	1332.96	93.94
KX004	1346.60	82.52
KX003	1456.33	82.08

Another important feature of our data is that we have more confidence in the branch lengths of early branches than late branches. Early branches have effectively been sequenced in many individual colonies, to a high combined coverage, meaning that our confidence in their lengths is high. Conversely, private (terminal) branches have higher uncertainty due to the lower effective coverage from sequencing only a single colony containing its variants.

To reflect this, we developed a custom method for making our phylogenies ultrametric that gives more weight to the lengths of early branches. This method, which we term an ‘iteratively reweighted means approach’, starts from the root of the tree and moves progressively towards each tip. For each coalescence, the fraction of overall time at that branch-point is calculated as the fraction of remaining time multiplied by the number of mutations on the given shared branch divided by the mean number of mutations of all descendants from that shared branch. The function is called recursively, updating the fraction of remaining time, as the algorithm moves from root to tip. This algorithm therefore has the property that the most confident timings (nodes near the root) are defined first, anchoring the timings of subsequent, less confident nodes.

In response to the reviewer’s suggestion to test other approaches for tree linearisation, we have used the Bayesian algorithm, Rtreefit, developed by Williams *et al*¹⁰ for somatic mutations in blood cancers. In brief, Rtreefit is a Bayesian model for converting trees based on mutation counts into time-based trees. The method jointly fits a global constant mutation acquisition rate and absolute time branch lengths under the assumption that the observed mutation count based branch lengths are Poisson distributed with $Mean = Duration \times Sensitivity \times Mutation\ Rate$ and subject to the constraint that the root to tip duration is the age at colony sampling. The mean branch timings are directly

sampled from the posterior distribution and by construction the resulting trees are guaranteed to have a root to tip distance that matches the sampling age of the colony. The model is coded in R and Rstan and inferred using the Rstan implementation of Stan's No-U-Turn sampler variant of Hamiltonian Monte Carlo method. For each patient tree, the model was fitted across four chains each with 20,000 iterations including 10,000 burn-in iterations. The code is available as an R package "Rtreefit" at <https://github.com/NickWilliamsSanger/rtreefit>.

Reassuringly, we found extremely high concordance between our custom 'iteratively re-weighted means' approach and the Bayesian approach described above. In all phylogenies the R^2 for branch length comparisons between the two approaches was > 0.99 (see **Reviewer Figure 4** below).

We have included results from the comparison of the iteratively re-weighted means approach we used for phylogeny linearisation and the alternative Bayesian approach described above in the Materials and Methods. Reviewer Figure 4 below has been included as Supplementary Figure 3. We have also provided more computational details on the iteratively reweighted means approach in the Methods.

Reviewer Figure 4 (included as Supplementary Figure 3). Scatter plots showing the estimates of branch lengths for each phylogeny from the iteratively reweighted means algorithm (x axis) versus the branch lengths estimated using the Bayesian algorithm, Rtreefit (y axis). The lines show the equality $x=y$.

(h) Based on the phylogeny included in the supplementary information (for an older individual), the accumulation of mutations seems to be relatively constant. But, the mutation rate is likely different in the initial life stages (first few years) in which cell divisions seem to produce two stem cells as compared to the rest of the lifetime in which cell divisions appear to be asymmetric such that they produce one stem cell and another progenitor/specialized cell. Wouldn't this change the mutation rate between two phases, making it difficult to convert a phylogeny into a clonal timetree? Is it possible to estimate the early and late mutation rates and compare them to mutation rates in other somatic contexts?

We agree this is an important point to address. Thankfully, we have reasonably accurate estimates of the mutation rates during foetal development, which enables us to calibrate the early molecular clock on the phylogenies to chronological age.

The mean mutation burden across HSC/MPPs from our two cord blood samples was 55 SNVs/cell. Our published data on whole genome sequencing from two human foetuses allow us to refine this rate further⁶⁷, where we observed a mean mutation burden in HSPCs at 8 weeks' gestation of 25 SNVs/cell, and at 18 weeks of 42 SNVs/cell. These data show that the rate of mutation acquisition slows considerably during development, from an average rate of 3.2 mutations per week in the first 8 weeks to an average rate of 1.6 mutations per week between weeks 8 and 18, and likely also slows considerably in the latter half of gestation when compared to our cord blood mutation burden of 55/cell. The study on foetuses is also helpful in demonstrating there is a rate of mutation acquisition per cell division of < 0.9 after the first 3 cell divisions in the zygote⁶⁷. Although this rate of mutation accumulation per cell division is low, due to the very much higher rates of cell division in early development, we believe it accounts for the overall higher rates of mutation acquisition observed.

Bulk blood telomere data show that telomere loss is higher in the first 6 months of life after birth, before stabilising to adult levels over the rest of childhood⁶⁸. It is therefore likely that from the age of 1 year onwards mutation rate is close to constant due to the fact that cell division rate (and therefore telomere loss) stabilises after this period. Telomere loss data are helpful as they allow us to assess total cell division rates (both symmetric and asymmetric) over life. Although the reviewer makes the statement that HSC division over adult life is asymmetric, recent experimental work in mice supports the view that over 90% of HSC divisions in adult life remain symmetric^{69,70}.

Despite the hint from telomere data that the rate of mutation burden accumulation may be higher in the first year of postnatal life, we do not have any direct evidence of a higher mutation rate in very early childhood from our data. A linear regression on age and mutation burden using data from the two cord blood donors and the youngest adult age 29 gives a rate of mutation accumulation of 17.56 per year (CI_{95%}= 17.32-17.78) over this younger period, which is very close to the rate calculated over the whole lifespan (16.8). In addition, the analysis we made of population dynamics (**Fig. 4a**) suggests that the HSC/MPP population reaches its adult size around the time of birth, which would fit with a switch from a population growth phase requiring more frequent cell division to steady state adult cell division frequency within the first year after birth. To further support the linearity of mutation accumulation in early life, recently published work has looked at HPC mutation burden in paediatric haematopoietic stem cell donors and recipients aged between 2 and 20 years⁷¹. The mutation burdens across this age range fit exactly with the model of linear acquisition that we have documented in adult life. Although we cannot exclude a slightly higher HSC mutation rate in the first 1-2 years of life, this would have very minimal impact on the validity of our conversion of the phylogenies to time trees from birth onwards.

Importantly, as discussed in the previous point, the within-individual mutation burden for the two foetuses was remarkably consistent, with little more than Poisson variation. This again argues that while the mutational clock ticks faster during development than in adulthood, this is constant across different HSC clades, and therefore the linearisation methods across clades are appropriate.

We had already accounted for the higher mutation rate during *in utero* development in Fig. 4a, but in response to this comment, we have now also accounted for it in the age axis of the phylogenies in Figs 2 and 3. The adjustment is performed by assigning the first 55 mutations of molecular time to the 'in utero' period (such that 'birth' or age zero on the axis occurs at 55 mutations) and scaling the remaining postnatal mutation time by chronological age in years. We have also referenced the recent work on paediatric HPC mutation burdens in the Introduction.

(i) Also, the authors need to test whether the mutation rate is constant during all phases of the lifetime. This may be checked by examining the estimates of mutation rates (or diversity) from

younger individuals compared to older individuals (but maybe not the oldest). This could reveal a mutational rate shift in different segments of long legs in the phylogeny.

There is now considerable evidence that the major influence on HSC mutation rate is time, rather than cell division, meaning that small changes in numbers of cell division over life or in response to perturbations will have minimal impact on mutation burden^{17,71,72}.

As stated in the response to the previous point, we have evidence for an elevated HSC/MPP mutation rate during *in utero* development. There is little precise evidence for HSC/MPP mutation rate in the first year of postnatal life as discussed above, but in general, all the evidence we have points towards a relatively constant relationship from very early childhood onward. We have now performed linear regression of age and mutation burden in age ranges. Using data from the two cord blood donors and the youngest adult age 29 gives a rate of mutation accumulation of 17.56 per year (CI_{95%}= 17.32-17.78). Using data from just the 3 younger donors aged 29, 38, and 48 gives a rate of mutation accumulation of 17.21 per year (CI_{95%}= 16.12-18.3). Linear regression of age and mutation burden using the 5 older donors aged 63, 75, 76, 77 and 81 gives a rate of mutation accumulation of 18.84 per year (CI_{95%}= 16.82-20.86). These results are very consistent over different phases of life. We do, however, agree that due to the relatively small number of individuals in our study, this analysis cannot be wholly conclusive. Considerable further work would need to be performed to identify any small variation in HSC/MPP mutation rate over life that we are unable to detect with this study.

Whilst this analysis on variable mutation rate with age still leaves some uncertainty in the exact timing of historical coalescent events, we believe the range of uncertainty is small and does not have any impact on the main analyses we present or conclusions we draw.

We have included the results of the linear regression broken down into age ranges in the Materials and Methods ('Mutation burden analysis').

(j) For such data, SCITE is the most appropriate method for lower-quality data (many unknown and incorrect base assignments), and the authors have used it already (as reported in the Supplementary information). SCITE produces a mutation tree, and thus a clone phylogeny and branch lengths. Those are expected to be most appropriate for the current application. So, why MPBoot is being applied and why the ML method estimates branch lengths are not clear.

The major advantage of our approach of growing single cells into colonies, rather than directly sequencing the single cells themselves after whole genome amplification, is that the genomes we get are considerably higher quality. First, we have very few false positive mutation calls because the DNA amplification from the single cell happens *in cellulo*, where all the DNA repair processes remain active, rather than *in vitro*, where only a single, rather error-prone processive bacteriophage polymerase is used. Second, the genome coverage is much more even in the colonies, meaning that the average depth we achieved here, 14x, is generally consistent across every mutated base. As a result, the fraction of sites where we are missing data is generally no more than 0.5-3% (see table below). For example, in KX003, only 1.17% of genotypes were 'unknown', and, as demonstrated by the disagreement scores, the vast majority of allocated genotypes were concordant with phylogenetic assumptions.

Sample_ID	Genotype 0 'absent'	Genotype 0.5 'unknown'	Genotype 1 'present'	% sites 'missing data'
KX001	3,068,118	42,512	22,049	1.38

KX002	2,719,616	28,599	20,085	1.05
SX001	4,003,266	127,878	32,218	3.17
AX001	5,086,607	173,657	39,577	3.39
KX007	9,081,044	48,041	97,580	0.52
KX008	10,248,698	78,247	157,878	0.75
KX004	21,179,417	197,625	249,761	0.92
KX003	8,993,397	107,236	187015	1.17

These features mean that MPBoot is an appropriate method for phylogeny inference – maximum parsimony methods work well when the data are relatively complete across sites, as evidenced here, and the infinite sites assumption holds (which it does since overall mutation burdens are very low, less than 1 in a million bases genome-wide per cell, with relatively even distribution across the genome).

We do agree with the reviewer that SCITE is also an appropriate method, with several advantages, as pointed out. We have now run SCITE over all but one of the phylogenies for this revision. However, we have found that the computational burden of SCITE is challenging – taking about 2-3 weeks to complete on average. The largest phylogeny (KX004) cannot complete within the timeframe required for our compute farm ‘basement queue’, which terminates jobs after 4 weeks. For this reason, we have stayed with the MPBoot trees in the figures for consistency across the dataset.

To be confident that the two methods give concordant trees, we have measured the similarity of trees estimated with MPBoot and SCITE. Reassuringly, in all cases there was high concordance in the phylogenies produced by the two approaches (Robinsons-Foulds distance < 0.07) as shown in the table below.

Individual	Robinsons-Foulds Similarity of SCITE tree	Quartet Similarity of SCITE tree	Comparison of 32 summary statistics
KX001	0.934	1.000	Unchanged
KX002	0.949	0.999	Unchanged
SX001	0.945	0.999	Unchanged
AX001	0.947	1.000	Unchanged
KX007	0.977	0.999	Subtle changes (see below)
KX008	0.960	0.998	Unchanged
KX003	0.954	1.000	Subtle changes (see below)

Most differences that did emerge affected the precise arrangement of some early embryonic branch points – these differences would not be anticipated to have an impact on any of the key downstream analyses in the manuscript. See **Reviewer Figure 5** below for 4 examples of the comparison, in which discrepant nodes between the phylogenies inferred using MPBoot vs. SCITE are highlighted.

We have also formally compared the summary statistics obtained from the phylogenies inferred using MPBoot vs SCITE. We found that when the range of summary statistics we utilise for the driver ABC modelling are assessed at 4 timepoints, in 5 out of 7 individuals the statistics for the MPBoot and SCITE phylogenies are identical. For KX007, 6 of 32, and for KX003, 4 of 32 summary statistics calculated for each phylogeny are discordant but by a negligible amount (2 or less). See the tables below for the discordant statistics for KX007 and KX003, showing the marginal differences between the approaches.

These overall highly concordant findings further confirm that the choice of tree-building approach would not have altered the conclusions of the downstream modelling analyses.

KX007: discrepant summary statistics between MPBoot and SCITE phylogenies

Tree building	n_singletons_2	n_singletons_3	ltt_2	ltt_3	coals_1	coals_3
MPBoot	224	290	253	300	43	15
SCITE	226	292	254	301	44	14

KX003: discrepant summary statistics between MPBoot and SCITE phylogenies

Tree building	n_singletons_3	clade_size_b_3	ltt_3	coals_3
MPBoot	241	4	271	54
SCITE	240	5	269	56

n_singletons_2 = number singleton samples at timepoint 2

n_singletons_3 = number singleton samples at timepoint 3

ltt_2 = number lineages at timepoint 2

ltt_3 = number lineages at timepoint 3

coals_1 = number coalescences in time period 1

coals_3 = number coalescences in time period 2

clade_size_b3 = size of the second largest clade at timepoint 3

We have incorporated the percentage of sites with missing data and a concise version of comparison of the summary statistics between the MPBoot and SCITE phylogenies in the Materials and Methods ('Validation of the phylogeny').

Reviewer Figure 5. Comparison of phylogenetic trees between MPBoot and SCITE methods for four individuals in that study. Red bars highlight branches that are discrepant between the two methods.

(k) Also, in Supplementary Figure S3, the method used for S3b needs to be mentioned. MPBoot phylogenies in Figures S3e and S3f have different bootstrap values?

Supplementary Fig. 3b – the same raw read count bootstrapping approach was used as in Supplementary Fig. 3a. **We have updated the figure legend to clarify this.**

Supplementary Fig. 3e and 3f – the annotated nodes do not relate to bootstrap values, but highlight differences in the phylogenies obtained between MPBoot and alternative phylogeny-inference approaches. **We have changed the figures and legends to clarify this.**

(l) The article also does not present enough information on how the completeness of sequence alignments. What was the percentage of positions with missing data, and was there a need to use methods to impute missing data and correct sequencing errors (e.g., SCITE, BEAM, etc.). It would also be useful to know how often cells were not heterozygous (e.g., LOH).

We have included information on the completeness of sequence alignments in response to comment (j) and the Materials and Methods. No imputation of missing data or correction of sequencing errors was performed. LOH was rare in our data as shown in **Fig. 1e-f**, and therefore the overwhelming majority of somatic mutations were heterozygous, with a VAF of ~50%.

(m) As for standard bootstrapping, it is expected to be low for short branch lengths, i.e., those with only one or a few mutations, due to standard properties of the site-sampling bootstrap (e.g., a branch must contain at least 3 mutations before the expected bootstrap support can ever reach 95%). So, bootstrap support for early branching events is expected to be very low, but this uncertainty is unlikely to affect the downstream inference.

As the reviewer points out, the shape of our tree, with multiple short branches that are ancestral to many cells, makes bootstrapping rather conservative. As some of the earliest splits in the tree are supported by a single mutation which may be omitted in a bootstrap replicate (indeed, a given mutation will be present on average in 630 out of 1000 replicates), some of the earliest splits may not be supported by a particularly high proportion of replicates, even though the mutation calls themselves are highly confident (based on multiple colonies carrying 5-10 reads reporting the variant, while being completely absent from other colonies).

We have extended this point in the relevant section of the Materials and Methods ('Validation of the phylogeny').

(n) It will be useful to be more explicit about how the trees were rooted. I gleaned that somewhat from the protocol noted, but was an actual sequence of (e.g., 0's) used in MPBoot analysis to root the tree? SCITE will automatically assume that but was that done in MPBoot, or is midpoint rooting used? This clarification is important to know if the early phase was symmetric cell division, leading to a balanced tree or a highly asymmetric one.

The trees were rooted using an 'ancestral' branch in the MPBoot analysis (coded as a string of W's for wildtype). Variants were coded as 'V' and missing data as '?'. See example below:

Ancestral:
"XXXXXXXXXXXXXXXXXXXXXXXXXXXXXXXXXXXX"
PD40521aa:
"XXXXXXXXVXXXXXXXXXXXXXXXXXXXXX?"
PD40521ab:
"XXXXXXXXXXXXXXXXXXXXVXXXXXXXXXXXX"

This information has been added to the Materials and Methods ('Construction of phylogenetic trees').

(o) In the phylogenies presented, HSCs and MPPs are not immediately clear. Maybe I missed it. It will be useful to mark those lineages if they are both included. Are the dynamics different for HSC and MPPs?

Apologies for the confusion surrounding the HSC nomenclature used in the manuscript. The term 'HSC/MPP' refers to a single population of sorted cells (as per **Extended Fig. 1**). Xenograft transplantation experiments have demonstrated that this population is the best described to date which contains both HSCs, defined as providing long-term human blood repopulation in the transplanted animals, as well as MPPs, which have short-term repopulation capacity in the same assays^{18,73–75}. We have not further subclassified the "HSC/MPP" population in this analysis.

In response to this comment we have improved our description of the HSC/MPP population in the Materials and Methods ('Fluorescence activated cell sorting').

(p) Data sharing. While the raw data and the associated source code to summarize them are available, the availability of alignments (1/0/? states and base calls) used for MPBoot (and other evolutionary analysis) is extremely important for others to be able to reanalyze/reproduce/advance the results presented. In particular, providing those in a form that can allow one to use standard methods for dN/dS analyses will be important to reproducing results and scrutinizing patterns.

We have uploaded all the dN/dS input data to Mendeley data (doi: 10.17632/np54zjkvxr.1) along with all the data necessary to repeat the mutation burden and tree building analysis (including running MPBoot). See below for a guide on how to locate different data matrices in the same format as used in the analyses presented here.

dNdS_input folder

Contains all raw input files for the dN/dS analysis.

Filtering_output_XXXX folders (one for each individual)

Contains three files:

a) annotated_mut_set_XXXX_01_standard_rho01

This is an R data object and is uploaded into an R workspace using load()

The genotype matrix used for MPBoot tree building is available in the matrix: filtered_muts\$Genotype_shared_bin

The dna strings used as input for MPboot are available in the vector: filtered_muts\$dna_strings

The annotated variant calls with tree node information are available in the matrix: filtered_muts\$COMB_mats.tree.build\$mat

The genotype matrix of mutations calls per sample is available in: filtered_muts\$COMB_mats.tree.build\$Genotype_bin

Information on whether the variant is an SNV or indel is available in: filtered_muts\$COMB_mats.tree.build\$mat\$Mut_type

A summary of total numbers of shared and private SNVs and indels is available in:

filtered_muts\$summary

b) XXXX_sensitivity

This file contains information on the sensitivity of SNV and Indel calls per sample.

c) tree_XXXX_01_standard_rho01.tree

The raw tree with branch lengths equal to number of mutations assigned (without adjustment for sequencing coverage).

metadata_matrix folder

Contains file “Summary_cut.csv” which records metadata on each sample in the dataset including cell_type sorted, sequencing depth, sequencing_platform, SNV burdens, indel burdens and telomere length.

The above guide to available data has been included in the Materials and Methods section ‘Data Availability’.

References

1. Busque, L. *et al.* Recurrent somatic TET2 mutations in normal elderly individuals with clonal hematopoiesis. *Nat. Genet.* **44**, 1179–1181 (2012).
2. Genovese, G. *et al.* Clonal hematopoiesis and blood-cancer risk inferred from blood DNA sequence. *N. Engl. J. Med.* **371**, 2477–87 (2014).
3. Jaiswal, S. *et al.* Age-related clonal hematopoiesis associated with adverse outcomes. *N. Engl. J. Med.* **371**, 2488–98 (2014).
4. Xie, M. *et al.* Age-related mutations associated with clonal hematopoietic expansion and malignancies. *Nat. Med.* **20**, 1472–8 (2014).
5. Mckerrell, T. *et al.* Leukemia-Associated Somatic Mutations Drive Distinct Patterns of Age-Related Clonal Hemopoiesis. *Cell Rep.* **10**, 1239–1245 (2015).
6. Young, A. L., Challen, G. A., Birman, B. M. & Druley, T. E. Clonal haematopoiesis harbouring AML-associated mutations is ubiquitous in healthy adults. *Nat. Commun.* **7**, 1–7 (2016).
7. Poon, G. Y. P., Watson, C. J., Fisher, D. S. & Blundell, J. R. Synonymous mutations reveal genome-wide levels of positive selection in healthy tissues. *Nat. Genet.* **53**, 1597–1605 (2021).
8. López-Otín, C., Blasco, M. A., Partridge, L., Serrano, M. & Kroemer, G. The hallmarks of aging. *Cell* **153**, (2013).
9. Jaiswal, S. *et al.* Clonal Hematopoiesis and Risk of Atherosclerotic Cardiovascular Disease. *N. Engl. J. Med.* NEJMoa1701719 (2017). doi:10.1056/NEJMoa1701719
10. Williams, N. *et al.* Phylogenetic reconstruction of myeloproliferative neoplasm reveals very early origins and lifelong evolution. doi:10.1101/2020.11.09.374710
11. Nei, M. & Gojobori, T. Simple methods for estimating the numbers of synonymous and nonsynonymous nucleotide substitutions. *Mol. Biol. Evol.* **3**, 418–426 (1986).
12. Greenman, C., Wooster, R., Futreal, P. A., Stratton, M. R. & Easton, D. F. Statistical analysis of pathogenicity of somatic mutations in cancer. *Genetics* **173**, 2187–2198 (2006).
13. Martincorena, I. *et al.* Universal Patterns of Selection in Cancer and Somatic Tissues. *Cell* **171**, 1029–1041.e21 (2017).
14. Belluschi, S. *et al.* Myelo-lymphoid lineage restriction occurs in the human haematopoietic stem cell compartment before lymphoid-primed multipotent progenitors. doi:10.1038/s41467-018-06442-4
15. Mende, N. *et al.* Quantitative and molecular differences distinguish adult human medullary and extramedullary haematopoietic stem and progenitor cell landscapes. *bioRxiv* 2020.01.26.919753 (2020). doi:10.1101/2020.01.26.919753
16. Werner, B. *et al.* Reconstructing the in vivo dynamics of hematopoietic stem cells from telomere length distributions. *Elife* **4**, 1–23 (2015).
17. Abascal, F. *et al.* Somatic mutation landscapes at single-molecule resolution. *Nature* **593**, 405–410 (2021).

18. Notta, F. *et al.* Isolation of single human hematopoietic stem cells capable of long-term multilineage engraftment. *Science* **333**, 218–21 (2011).
19. Meisel, M. *et al.* Microbial signals drive pre-leukaemic myeloproliferation in a Tet2-deficient host. *Nat.* 2018 5577706 **557**, 580–584 (2018).
20. Hormaechea-Agulla, D. *et al.* Chronic infection drives Dnmt3a-loss-of-function clonal hematopoiesis via IFN γ signaling. *Cell Stem Cell* **28**, 1428–1442.e6 (2021).
21. Ramdas, B. *et al.* Driver Mutations in Leukemia Promote Disease Pathogenesis through a Combination of Cell-Autonomous and Niche Modulation. *Stem Cell Reports* **15**, 95–109 (2020).
22. Zink, F. *et al.* Clonal hematopoiesis, with and without candidate driver mutations, is common in the elderly. *Blood* **130**, 742–752 (2017).
23. Ng, S. W. K. *et al.* Convergent somatic mutations in metabolism genes in chronic liver disease. *Nature* **598**, 473–478 (2021).
24. Duncavage, E. J. *et al.* Genome Sequencing as an Alternative to Cytogenetic Analysis in Myeloid Cancers. *N. Engl. J. Med.* **384**, 924–935 (2021).
25. The Cancer Genome Atlas Research Network. Genomic and epigenomic landscapes of adult de novo acute myeloid leukemia. *N Engl J Med* **368**, 2059–2074 (2013).
26. Klco, J. M. *et al.* Association between mutation clearance after induction therapy and outcomes in acute myeloid leukemia. *JAMA - J. Am. Med. Assoc.* **314**, 811–822 (2015).
27. Lee-Six, H. *et al.* Population dynamics of normal human blood inferred from somatic mutations. *Nature* **561**, 473–478 (2018).
28. Li, R. *et al.* A body map of somatic mutagenesis in morphologically normal human tissues. *Nature* **597**, 398–403 (2021).
29. Lopez-Garcia, C., Klein, A. M., Simons, B. D. & Winton, D. J. Intestinal stem cell replacement follows a pattern of neutral drift. *Science (80-.).* **330**, 822–825 (2010).
30. Lee-Six, H. *et al.* The landscape of somatic mutation in normal colorectal epithelial cells. *Nature* **574**, 532–537 (2019).
31. Nguyen, H. *et al.* Deficient Pms2, ERCC1, Ku86, CcOI in field defects during progression to colon cancer. *J. Vis. Exp.* 2–6 (2010). doi:10.3791/1931
32. Grossmann, S. *et al.* Development, maturation, and maintenance of human prostate inferred from somatic mutations. *Cell Stem Cell* **28**, 1262–1274.e5 (2021).
33. Yokoyama, A. *et al.* Age-related remodelling of oesophageal epithelia by mutated cancer drivers. *Nature* **565**, 312–317 (2019).
34. Martincorena, I. *et al.* Somatic mutant clones colonize the human esophagus with age. *Science (80-.).* **917**, 911–917 (2018).
35. Yoshida, K. *et al.* Tobacco smoking and somatic mutations in human bronchial epithelium. *Nature* **578**, 266–272 (2020).
36. Teixeira, V. H. *et al.* Stochastic homeostasis in human airway epithelium is achieved by neutral competition of basal cell progenitors. *Elife* **2**, e00966 (2013).
37. Martincorena, I. *et al.* High burden and pervasive positive selection of somatic mutations in normal human skin. *Science (80-.).* **348**, 880–886 (2015).
38. Williams, M. J. *et al.* Measuring the distribution of fitness effects in somatic evolution by combining clonal dynamics with dN/dS ratios. *Elife* **9**, 1–19 (2020).
39. Colom, B. *et al.* Spatial competition shapes the dynamic mutational landscape of normal esophageal epithelium. *Nat. Genet.* **52**, 604–614 (2020).
40. Kakiuchi, N. *et al.* Frequent mutations that converge on the NFKBIZ pathway in ulcerative colitis. *Nature* **577**, 260–265 (2020).
41. Nanki, K. *et al.* Somatic inflammatory gene mutations in human ulcerative colitis epithelium. *Nature* **577**, 254–259 (2020).
42. Olafsson, S. *et al.* Somatic Evolution in Non-neoplastic IBD-Affected Colon. *Cell* **182**, 672–684.e11 (2020).

43. Kim, S. K. *et al.* Comprehensive analysis of genetic aberrations linked to tumorigenesis in regenerative nodules of liver cirrhosis. *J. Gastroenterol.* (2019). doi:10.1007/s00535-019-01555-z
44. Zhu, M. *et al.* Somatic Mutations Increase Hepatic Clonal Fitness and Regeneration in Chronic Liver Disease. *Cell* **177**, 608–621 (2019).
45. Brunner, S. F. *et al.* Somatic mutations and clonal dynamics in healthy and cirrhotic human liver. *Nature* **574**, 538–542 (2019).
46. Fowler, J. C. *et al.* Selection of oncogenic mutant clones in normal human skin varies with body site. *Cancer Discov.* **11**, 340–361 (2021).
47. Moore, L. *et al.* The mutational landscape of normal human endometrial epithelium. *Nature* **580**, (2020).
48. Bertorelle, G., Benazzo, A. & Mona, S. ABC as a flexible framework to estimate demography over space and time: Some cons, many pros. *Mol. Ecol.* **19**, 2609–2625 (2010).
49. Beaumont, M. A., Zhang, W. & Balding, D. J. Approximate Bayesian Computation in Population Genetics. *Genetics* **162**, 2025–2035 (2002).
50. Kirkwood, T. B. L. & Melov, S. On the programmed/non-programmed nature of ageing within the life history. *Curr. Biol.* **21**, R701–R707 (2011).
51. Cagan, A. *et al.* Somatic mutation rates scale with lifespan across mammals. *bioRxiv* (2021). doi:10.1101/2021.08.19.456982
52. Abkowitz, J. L., Catlin, S. N., McCallie, M. T. & Guttorp, P. Evidence that the number of hematopoietic stem cells per animal is conserved in mammals. *Blood* **100**, 2665–2667 (2002).
53. Cosgrove, J., Hustin, L. S. P., de Boer, R. J. & Perié, L. Hematopoiesis in numbers. *Trends Immunol.* **42**, 1100–1112 (2021).
54. Foudi, A. *et al.* Analysis of histone 2B-GFP retention reveals slowly cycling hematopoietic stem cells. *Nat. Biotechnol.* **27**, 84–90 (2009).
55. Kaschutnig, P. *et al.* The Fanconi anemia pathway is required for efficient repair of stress-induced DNA damage in haematopoietic stem cells. *Cell Cycle* **14**, 2734–2742 (2015).
56. Wilson, A. *et al.* Hematopoietic stem cells reversibly switch from dormancy to self-renewal during homeostasis and repair. *Cell* **135**, 1118–29 (2008).
57. Ganuza, M. *et al.* The global clonal complexity of the murine blood system declines throughout life and after serial transplantation Short title (48 characters including spaces): The clonal complexity of blood declines with age. *Blood First Ed. Pap.* (2019). doi:10.1182/blood-2018-09-873059
58. Yu, K. R. *et al.* The impact of aging on primate hematopoiesis as interrogated by clonal tracking. *Blood* **131**, 1195–1205 (2018).
59. Rheinbay, E. *et al.* Analyses of non-coding somatic drivers in 2,658 cancer whole genomes. *Nature* **578**, 102–111 (2020).
60. Lawrence, M. S. *et al.* Mutational heterogeneity in cancer and the search for new cancer-associated genes. *Nature* **499**, 214–8 (2013).
61. Jones, D. *et al.* cgpCaVEManWrapper: Simple Execution of CaVEMan in Order to Detect Somatic Single Nucleotide Variants in NGS Data. in *Current Protocols in Bioinformatics* **2016**, 15.10.1-15.10.18 (John Wiley & Sons, Inc., 2016).
62. Alexandrov, L. B. *et al.* The repertoire of mutational signatures in human cancer. *Nature* **578**, 94–101 (2020).
63. Vaser, R., Adusumalli, S., Ngak Leng, S., Sikic, M. & Ng, P. C. SIFT missense predictions for genomes. *Nat. Protoc.* (2015). doi:10.1038/nprot.2015.123
64. Adzhubei, I., Jordan, D. M. & Sunyaev, S. R. Predicting Functional Effect of Human Missense Mutations Using PolyPhen-2. doi:10.1002/0471142905.hg0720s76
65. Welch, J. S. *et al.* The Origin and Evolution of Mutations in Acute Myeloid Leukemia. *Cell* **150**, 264–278 (2012).
66. Osorio, F. G. *et al.* Somatic Mutations Reveal Lineage Relationships and Age-Related

- Mutagenesis in Human Hematopoiesis. *Cell Rep.* **25**, 2308-2316.e4 (2018).
67. Spencer Chapman, M. *et al.* Lineage tracing of human development through somatic mutations. *Nature* **595**, 85–90 (2021).
 68. Rufer, N. *et al.* *Telomere Fluorescence Measurements in Granulocytes and T Lymphocyte Subsets Point to a High Turnover of Hematopoietic Stem Cells and Memory T Cells in Early Childhood.* *J. Exp. Med* **190**, (1999).
 69. Barile, M. *et al.* Hematopoietic stem cells self-renew symmetrically or gradually proceed to differentiation. doi:10.1101/2020.08.06.239186
 70. Ito, K. *et al.* Self-renewal of a purified Tie2+ hematopoietic stem cell population relies on mitochondrial clearance. *Science (80-.).* **354**, 1156–1160 (2016).
 71. de Kanter, J. K. *et al.* Antiviral treatment causes a unique mutational signature in cancers of transplantation recipients. *Cell Stem Cell* **28**, 1726-1739.e6 (2021).
 72. M, S. C. & J, N. Caught in the antiviral crossfire: Ganciclovir-associated mutagenesis in HSC transplant recipients. *Cell Stem Cell* **28**, 1683–1685 (2021).
 73. Huntsman, H. D. *et al.* Human hematopoietic stem cells from mobilized peripheral blood can be purified based on CD49f integrin expression. *Blood* **126**, 1631–1633 (2015).
 74. Laurenti, E. *et al.* CDK6 Levels Regulate Quiescence Exit in Human Hematopoietic Stem Cells. *Cell Stem Cell* **16**, 302 (2015).
 75. Notta, F. *et al.* Distinct routes of lineage development reshape the human blood hierarchy across ontogeny. *Science* **351**, 1–16 (2015).

Reviewer Reports on the First Revision:

Referees' comments:

Referee #2 (Remarks to the Author):

This is one of the best manuscripts I've read in a while, and one of the most thoughtful response to reviewers. I have no further comments.

Referee #3 (Remarks to the Author):

Authors have responded to my comments satisfactorily.